# Information-theoretic Generalization Analysis for VQ-VAEs: A Role of Latent Variables

**Futoshi Futami**[*,1,2,3]**, Masahiro Fujisawa**[*,1,2]**,**
[1] The University of Osaka, [2] RIKEN AIP, [3] The University of Tokyo,
[*] Equal Contribution
futami.futoshi.es@osaka-u.ac.jp, fujisawa@ist.osaka-u.ac.jp

## Abstract

Latent variables (LVs) play a crucial role in encoder–decoder models by enabling effective data compression, prediction, and generation. Although their theoretical properties, such as generalization, have been extensively studied in supervised learning, similar analyses for unsupervised models such as variational autoencoders (VAEs) remain insufficiently explored. In this work, we extend information-theoretic generalization analysis to vector-quantized (VQ) VAEs with discrete latent spaces, introducing a novel data-dependent prior to rigorously analyze the relationship among LVs, generalization, and data generation. We derive a novel generalization error bound of the reconstruction loss of VQ-VAEs, which depends solely on the complexity of LVs and the encoder, independent of the decoder. Additionally, we provide the upper bound of the 2-Wasserstein distance between the distributions of the true data and the generated data, explaining how the regularization of the LVs contributes to the data generation performance.

## 1 Introduction

Encoder–decoder (ED) models have demonstrated remarkable performance [23] in (un)supervised tasks such as classification [2, 4] and data generation [39, 70], compressing input data into latent variables (LVs) via an encoder. The success of ED models hinges on how effectively the encoder can represent essential features of the input in LVs, stimulating analyses of the relationship between LVs and ED model performance, as well as developing algorithms designed to appropriately control LVs.

In supervised learning, the information bottleneck (IB) hypothesis [67, 60] has gained significant attention for proposing that minimizing the mutual information (MI) between input data and LVs enhances generalization by ensuring LVs retain the minimal information necessary for prediction. This hypothesis has motivated numerous learning algorithms for deep neural networks and empirical studies exploring their performance [66, 61, 56, 22, 1, 2]. Moreover, theoretical research about how LVs contribute to generalization has been actively pursued [71, 28, 38, 72] within the IB hypothesis. Recently, Sefidgaran et al. [57] has highlighted the limitations of these analyses, particularly in terms of assumptions and the sample complexity represented by the MI. To address these limitations, they proposed extending the supersample setting of information-theoretic (IT) analysis [63]. Their approach induces a *symmetric, data-dependent prior over LVs* that facilitates rigorous analysis, which successfully characterizes generalization performance using the Kullback–Leibler (KL) divergence between the posterior distribution of the LVs and this prior. These results suggest that, by carefully constructing the data-dependent prior distribution, we can obtain **a decoder-independent bound**, which illustrates clearly how LVs contribute to the generalization for ED models in classification. Their analysis has recently been extended to multi-view learning settings [58, 59].

LVs play a key role in deep generative models for unsupervised learning tasks such as data compression and generation. For example, variational autoencoders (VAEs) [39] are trained by optimizing an

objective function that includes the KL divergence of the posterior from the prior in the LV space as a regularization term. Extended methods such as $\beta$-VAE [34] highlight the importance of appropriately tuning the strength of KL regularization to improve LV representations. Additionally, methods like vector-quantized VAEs (VQ-VAEs) [70], which discretize the latent space, have been developed to address posterior collapse. Numerous empirical studies have also evaluated model performance based on the MI, such as the IB hypothesis and rate-distortion theory [3, 9, 69, 12].

In contrast to supervised learning, theoretical insights into the relationship between the generalization of ED models and LVs in unsupervised learning remain limited. Although Chérief-Abdellatif et al. [13] has employed *probably approximately correct* (PAC) Bayes analysis [47, 6] to investigate the generalization error defined in terms of reconstruction loss, they consider the posterior and prior distributions over the *encoder and decoder parameters*. Similarly, Epstein & Meir [19] focused on the complexity of encoder and decoder parameters to analyze the generalization capability. Therefore, these studies lack the analysis of the relationship between LVs and generalization capability. Mbacke et al. [46] attempted to address this problem by deriving PAC-Bayes bounds based on the KL divergence within prior and posterior distributions over LVs; however, their analysis relies on the impractical assumption that decoders are not trained, leaving significant challenges in achieving a practical understanding of the role of LVs in generalization performance.

To address these challenges, we provide the first rigorous theoretical analysis of the relationship among LVs, generalization, and data generation in ED models, with a focus on VQ-VAEs [70]. Motivated by Sefidgaran et al. [57], we construct a data-dependent prior over LVs using the supersample setting from IT analysis [63, 30, 32]. This approach yields a generalization error bound for the reconstruction loss, characterized by the KL divergence between the prior and the posterior over LVs (Theorem 2). Similar to Sefidgaran et al. [57], our bound remains independent of decoder complexity even when the encoder and decoder are trained jointly, underscoring the critical role of designing the encoder network for the generalization.

However, we observe that the bound based on the supersample setting does not necessarily converge to 0 asymptotically with respect to the number of samples. To address this issue, we extend the supersample framework by introducing a novel data-dependent prior, called the *permutation symmetric prior distribution*, which explicitly accounts for the inherent symmetries specific to unsupervised learning tasks (Theorem 3). This formulation enables us to derive a generalization error bound that asymptotically converges to 0 as the number of samples increases and is independent of the decoder.

Finally, we investigate the data generation capability of VQ-VAEs by deriving the upper bound on the 2-Wasserstein distance between the true data and the generated data distributions (Theorem 5). Our analyses reveal that the generalization and data-generating capabilities of VQ-VAEs depend solely on the parameters of the encoder and LVs, *remaining entirely independent of the decoder*.

## 2 Background

In this section, we introduce the VQ-VAE and define the reconstruction-based generalization error, which forms the basis of our analysis (Sections 2.1 and 2.2). We then present the IT analysis using *supersamples* (Section 2.3), highlighting its limitations in unsupervised settings (Section 2.4).

**Notations:** We use uppercase letters for random variables and lowercase letters for their realizations. The distribution of $X$ is denoted by $p(X)$, and the conditional distribution of $Y$ given $X$ by $p(Y|X)$. Expectations are written as $\mathbb{E}_{p(X)}$ or $\mathbb{E}_X$. The MI and conditional MI (CMI) are denoted by $I(X;Y)$ and $I(X;Y|Z)$, respectively. The KL divergence from $p(X)$ to $p(Y)$ is written as $\mathrm{KL}(p(X)\|p(Y))$. For $a \in \mathbb{N}$, we define $[a] := \{1, \ldots, a\}$.

### 2.1 VQ-VAE and its stochastic extensions

Let $\mathcal{X} \subset \mathbb{R}^d$ denote the data space, and assume an unknown data-generating distribution $\mathcal{D}$. The latent space is represented as $\mathcal{Z} \subset \mathbb{R}^{d_z}$, where both $\mathcal{X}$ and $\mathcal{Z}$ are equipped with the Euclidean metric $\|\cdot\|$. The discrete latent space comprises $K$ distinct points, collectively referred to as the *codebook*, denoted by $\mathbf{e} = \{e_j\}_{j=1}^K \in \mathcal{Z}^K$, which are learned from the training data.

The VQ-VAE model consists of the encoder network $f_\phi \colon \mathcal{X} \to \mathcal{Z}$ and the decoder network $g_\theta \colon \mathcal{Z} \to \mathcal{X}$ responsible for (i) data compression and (ii) reconstruction, where $\phi \in \Phi \subset \mathbb{R}^{d_\phi}$ and $\theta \in \Theta \subset \mathbb{R}^{d_\theta}$

denote the parameters of the encoder and decoder, respectively. In the compression phase, a data point $x$ is mapped to $f_\phi(x)$, and the discrete representation $e_j$ is selected from the codebook $\mathbf{e}$. Then, the posterior distribution of the discrete representation indexed by $j$ is denoted as $q(J = j|\mathbf{e}, \phi, x)$ for all $j = 1, \ldots, K$. In the original VQ-VAE [70], the following deterministic posterior is used:

$$q(J = j|\mathbf{e}, \phi, x) = \begin{cases} 1 & \text{for } j = \operatorname{argmin}_{k \in [K]} \|f_\phi(x) - e_k\|, \\ 0 & \text{otherwise,} \end{cases} \tag{1}$$

where the distance between the encoder output and the codebook entries determines the posterior. Recent extensions of VQ-VAE [82, 62, 55, 64] introduce a stochastic posterior defined by

$$q(J = j|\mathbf{e}, \phi, x) \propto \exp\left(-\beta \|f_\phi(x) - e_j\|^2\right), \tag{2}$$

where a softmax is applied over codebook indices, and the temperature parameter $\beta \in \mathbb{R}^+$ controls the level of stochasticity. The data is then reconstructed by passing the selected latent representation $e_{J=j}$ through the decoder, resulting in $g_\theta(e_{J=j})$. The fidelity of the reconstruction to the original input is measured by the *reconstruction loss*, defined as $l(x, g_\theta(e_{J=j}))$, where $l : \mathcal{X} \times \mathcal{X} \to \mathbb{R}^+$.

## 2.2 Generalization error based on reconstruction loss

Hereafter, let the set of parameters be denoted as $W := \{\mathbf{e}, \phi, \theta\} \in \mathcal{W} (:= \mathcal{Z}^K \times \Phi \times \Theta)$. Given the training dataset $S = (S_1, \ldots, S_n) \in \mathcal{X}^n$ consisting of independently and identically distributed (i.i.d.) data points sampled from the data distribution $\mathcal{D}$, these parameters are learned jointly using a randomized algorithm $\mathcal{A} : \mathcal{X}^n \to \mathcal{W}$ that minimizes the reconstruction loss between a data point $x$ and a reconstructed data $g_\theta(e_j)$, i.e., $l(x, g_\theta(e_j))$. Consequently, the learned parameters $\mathbf{e}, \phi, \theta$ follow the conditional distribution $q(\mathbf{e}, \phi, \theta|S)$. For simplicity, we define the expected reconstruction loss for an input $x$ and $w$ as $l_0 : \mathcal{W} \times \mathcal{X} \to \mathbb{R}$, where $l_0(w, x) := \mathbb{E}_{q(J|\mathbf{e}, \phi, x)}[l(x, g_\theta(e_J))]$. In this study, we consider the squared distance as $l$. Accordingly, our objective is to minimize $l_0(w, x) := \mathbb{E}_{q(J|\mathbf{e}, \phi, x)}[\|x - g_\theta(e_J)\|^2]$ over the training dataset $x \in S$. We introduce the following assumption about the data space imposed on our analysis.

**Assumption 1.** *There exists a positive constant $\Delta$ such that $\sup_{x, x' \in \mathcal{X}} \|x - x'\| < \Delta^{1/2}$.*

This assumption ensures that the reconstruction loss $l(x, g_\theta(e_j))$ is bounded by $\Delta$ for all $x$, $e_j$, and $\theta$.

Our goal is to theoretically characterize the relationship between generalization performance and LVs in VQ-VAEs. To this end, we analyze the following generalization error:

$$\text{gen}(n, \mathcal{D}) := \left| \mathbb{E}_{S,X} \mathbb{E}_{q(W|S)} l_0(W, X) - \frac{1}{n} \sum_{m=1}^n l_0(W, S_m) \right|, \tag{3}$$

where the first term denotes the expected test reconstruction loss, and the second term is the empirical training loss. Following the success of Sefidgaran et al. [57], we also consider analyzing Eq. (3) under the IT analysis framework with the *supersample* (or ghost sample) setting [63, 30, 32].

## 2.3 Supersample settings for IT analysis

Now, we introduce the supersample setting for IT analysis. We begin by defining a supersample $\tilde{X} \in \mathcal{X}^{n \times 2}$ as an $n \times 2$ matrix containing $2n$ data points drawn i.i.d. from $\mathcal{D}$. Each row $m \in [n]$ of this matrix, denoted $\tilde{X}_m$, represents a pair of data points: $(\tilde{X}_{m,0}, \tilde{X}_{m,1})$. We then generate a random binary vector $U = (U_1, \ldots, U_n) \sim \text{Uniform}(\{0, 1\}^n)$, which is independent of $\tilde{X}$. This index vector $U$ determines the training and test sets by selecting exactly one sample from each row. The training dataset is formed as $\tilde{X}_U := (\tilde{X}_{m,U_m})_{m=1}^n$, and the test dataset is composed of the remaining sample from each pair, $\tilde{X}_{\bar{U}} := (\tilde{X}_{m,\bar{U}_m})_{m=1}^n$, where $\bar{U}_m = 1 - U_m$. After training a model $W = \mathcal{A}(\tilde{X}_U)$, we define the loss matrix $l_0(W, \tilde{X})$ by evaluating the loss $l_0(W, \cdot)$ on all $2n$ data points in the original supersample matrix $\tilde{X}$. This results in an $n \times 2$ matrix of loss values. This distinction between the $n$-point training set and the $2n$-point loss evaluation matrix is a key concept for the subsequent analysis. The IT analysis of Eq. (3) under the supersample setting gives the following result.

**Theorem 1** (Hellström & Durisi [32])**.** *Under Assumption 1 and the supersample setting, we have*

$$\text{gen}(n, \mathcal{D}) \le \Delta\sqrt{2I(l_0(W, \tilde{X}); U|\tilde{X})/n}. \tag{4}$$

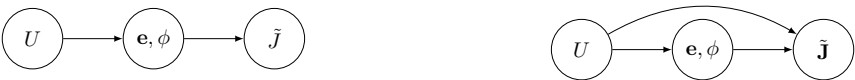

Figure 1: Graphical models illustrating different dependency structures for LVs. The left panel shows the structure considered in the standard supersample setting (Theorem 1). The right panel depicts our proposed structure tailored for unsupervised learning. See Appendix B.3 for further details.

The complete proof is provided in Appendix C. We refer to this bound as the **basic IT-bound**, as it arises from the direct application of existing IT analysis [32] developed for supervised learning. Unfortunately, we find that the basic IT-bound is insufficient to fully understand the role of LVs in the generalization performance of VQ-VAE. The next section elaborates on this limitation.

## 2.4 Limitation of the direct application of IT analysis

The limitation of the basic IT-bound is that it does not offer a clear interpretation of how the LVs contribute to the generalization performance independently of other random variables. Specifically, let $\widetilde{J}$ denote the random variable that follows the distribution $q(\widetilde{J}|\mathbf{e}, \phi, \tilde{X})$, which is defined by applying $q(J|\mathbf{e}, \phi, \cdot)$ elementwise to $\tilde{X}$. With this definition, we can upper bound Eq. (4) as

$$I(l_0(W, \tilde{X}); U|\tilde{X}) \leq I(\theta; U|\tilde{X}) + I(\widetilde{J}; U|\tilde{X}, \theta). \tag{5}$$

See Appendix C.2 for the proof. This result implies that the generalization of VQ-VAE can be bounded by the CMI related to the decoder parameter $\theta$ and the selected index $\widetilde{J}$. Note that selecting $J$ corresponds to selecting an LV $e_J$ from the codebook. Therefore, the second term above illustrates how LVs contribute to generalization. However, since conditioning on $\theta$ is taken, it does not allow the independent analysis of $e_J$ and $\theta$. This dependence hinders a precise theoretical analysis of how LVs affect generalization performance.

We can better understand this difficulty by considering how IT-based generalization analysis is typically formulated: it is framed as the problem of inferring which samples were used for training, given a random supersample index, $U$, that determines the shuffling of the dataset. The randomness introduced by this shuffling is governed by the design of the prior, which plays a central role in applying the Donsker–Varadhan inequality to derive an upper bound on the generalization error. In the basic IT-bound (Theorem 1), shuffling via $U$ leads to randomly altering the training dataset, producing a bound that jointly depends on both model parameters and LVs, thereby entangling $\theta$ and $J$. This illustrates that a straightforward extension of standard IT analysis is insufficient to isolate the contribution of LVs to generalization, motivating the development of a new analytical framework.

## 3 Proposed IT analysis under supersamples and its limitations

In this section, we first present the results of our generalization analysis for VQ-VAE (Section 3.1). We then offer a detailed interpretation of the resulting generalization error bound and discuss its limitations (Section 3.2). All corresponding proofs are provided in Appendix D.

## 3.1 Our supersample setting and result

As discussed in Section 2.4, the naive application of the existing supersample setting in IT analysis is insufficient to capture the role of LVs. To address this limitation, we introduce posterior and prior distributions over $J$ that explicitly encode the dependence between the supersample index $U$ and the LVs, on the basis of the approach of Sefidgaran et al. [57].

To this end, we define the following posterior distributions based on both $\tilde{X}_U$ and $\tilde{X}_{\bar{U}}$: $q(\mathbf{J}|\mathbf{e}, \phi, \tilde{X}_U) := \prod_{m=1}^{n} q(J_m|\mathbf{e}, \phi, \tilde{X}_{m,U_m})$ and $q(\bar{\mathbf{J}}|\mathbf{e}, \phi, \tilde{X}_{\bar{U}}) := \prod_{m=1}^{n} q(\bar{J}_m|\mathbf{e}, \phi, \tilde{X}_{m,\bar{U}_m})$. For notational simplicity, we write $\mathbf{Q}_{\mathbf{J},U} := q(\mathbf{J}|\mathbf{e}, \phi, \tilde{X}_U)$. We then define the following joint distribution to capture the dependence of the LVs on both $\tilde{X}_U$ and $\tilde{X}_{\bar{U}}$: $\mathbf{Q}_{\widetilde{\mathbf{J}},U} := q(\bar{\mathbf{J}}|\mathbf{e}, \phi, \tilde{X}_{\bar{U}}) \cdot q(\mathbf{J}|\mathbf{e}, \phi, \tilde{X}_U)$.

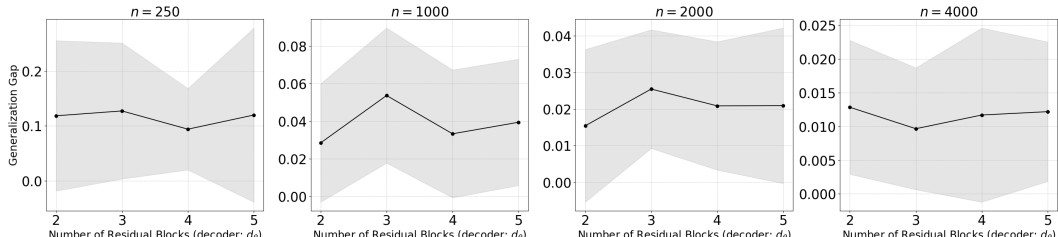

Figure 2: The behavior of the generalization gap on the MNIST dataset when increasing the number of residual blocks to enlarge the decoder dimension $d_\theta$ ($K = 128$, $d_z = 64$). See Appendix G for detailed experimental settings.

We consider two types of prior distribution to facilitate the analysis of VQ-VAEs: a *data-independent prior* $\mathbf{P}$ and a *data-dependent prior* $\mathbf{Q}_{\widetilde{\mathbf{J}}}$ defined as

$$\mathbf{P} := \prod_{m=1}^{n} \pi(J_m | \mathbf{e}, \phi), \quad \text{and} \quad \mathbf{Q}_{\widetilde{\mathbf{J}}} := \mathbb{E}_U \mathbf{Q}_{\widetilde{\mathbf{J}}, U} = \mathbb{E}_U q(\bar{\mathbf{J}} | \mathbf{e}, \phi, \tilde{X}_{\bar{U}}) q(\mathbf{J} | \mathbf{e}, \phi, \tilde{X}_U), \qquad (6)$$

where $\pi(J_m | \mathbf{e}, \phi)$ denotes an *arbitrary* distribution over LVs that is independent of both $\tilde{X}$ and the supersample index $U$. For the data-dependent prior, we adopt the supersample setting specifically tailored to the LVs. The basis for introducing both types of prior is discussed following the main theorem. Figure 1 illustrates the distinction in LV dependencies between the conventional supersample setting (as used in Theorem 1) and our approach. The central idea is to apply supersample-based shuffling to the LVs directly. Under these settings, the following is our main result.

**Theorem 2.** *Under Assumption 1 and the supersample setting, we have*

$$\text{gen}(n, \mathcal{D}) \leq 2\Delta \sqrt{\frac{\mathbb{E}_{\tilde{X}, U} \mathbb{E}_{q(\mathbf{e}, \phi | \tilde{X}_U)}(\text{KL}(\mathbf{Q}_{\mathbf{J}, U} \| \mathbf{P}) + \text{KL}(\mathbf{Q}_{\widetilde{\mathbf{J}}, U} \| \mathbf{Q}_{\widetilde{\mathbf{J}}}))}{n}} + \frac{\Delta}{\sqrt{n}}. \qquad (7)$$

The upper bound comprises two distinct complexity terms. The first, $\text{KL}(\mathbf{Q}_{\mathbf{J}, U} \| \mathbf{P})$, captures the *complexity of the LVs*. The second, $\text{KL}(\mathbf{Q}_{\widetilde{\mathbf{J}}, U} \| \mathbf{Q}_{\widetilde{\mathbf{J}}})$, reflects the complexity of the LVs and the *degree of overfitting when learning parameters* $\mathbf{e}$ *and* $\phi$, as we will further discuss in Section 3.2.

Consistent with the findings of Sefidgaran et al. [57], our bound is *independent of the decoder* $g_\theta$. This indicates that increasing the complexity of $g_\theta$ has a limited effect on the generalization performance. Our empirical results support this implication. Figure 2 shows that adding a single ResBlock—introducing approximately $74,000$ additional parameters—has a negligible effect on the generalization gap. Furthermore, Table 3 in Appendix G shows the corresponding training losses. For larger sample sizes ($n \geq 1000$), the training loss tends to decrease as the decoder becomes more complex, confirming its enhanced ability to fit the training data. Critically, despite this improved expressiveness, the generalization gap remains largely unaffected. This strongly suggests that the key to improving generalization lies not in the decoder's capacity, but in the complexity of the encoder and the LVs. Further experiments across various datasets and decoder architectures in Appendix G reinforce this observation.

We emphasize that our results *do not imply that the decoder is unimportant*. Although our generalization bound is independent of decoder complexity, a sufficiently expressive decoder is still required to fit the training data. Otherwise, the test loss may remain high since *Test Loss $\leq$ Training Loss + Generalization Gap*. Our analysis specifically focuses on bounding the generalization gap, under the implicit assumption that the decoder can adequately fit the training data. In practice, this suggests that improving generalization in VQ-VAEs hinges more on careful encoder design, since overly complex encoders can increase the KL divergence of the LVs. We discuss this point further in Section 6.

**Why two types of prior are required:** Our proof reveals that isolating the LVs from the decoder parameter and obtaining a decoder-independent generalization bound requires the prior to satisfy two essential conditions: (A) **it allows random shuffling without changing the LV distribution**, and (B)

**it supports a swap between training and test samples to assess overfitting**. From this perspective, the shuffling induced by $U$ in the basic IT-bound (Theorem 1) satisfies condition (B) but violates condition (A), as it changes the distribution of LVs. To address this issue, the proof of Theorem 2 decomposes the generalization gap into two components: the term associated with condition (A), which is controlled using a data-independent prior $\mathbf{P}$, and the term associated with condition (B), which is controlled using a data-dependent prior $\mathbf{Q}_{\tilde{\mathbf{J}}}$. By combining both priors, we can derive the final upper bound in Eq. (7). For a detailed explanation, see Appendices B.3 and D.1.

**Remark 1.** *When $K = 1$, VQ-VAEs map all input data to the same LV, effectively estimating the low-dimensional mean of the data distribution. In this case, the generalization error should not depend on the decoder. It is straightforward to show—without using our IT-based analysis—that* $\mathrm{gen}(n, \mathcal{D}) = O(1/\sqrt{n})$. *Notably, our bound in Eq. (7) correctly reflects this behavior, as the square root term vanishes when $K = 1$ (see Appendix C.3 for details).*

## 3.2 Further analyses of our bound and limitations on convergence

In this section, we further analyze the properties of the two KL divergence terms in Theorem 2 and discuss their asymptotic behavior as the sample size $n$ increases.

**Regarding** $\mathrm{KL}(\mathbf{Q}_{\tilde{\mathbf{J}},U}\|\mathbf{Q}_{\tilde{\mathbf{J}}})$**:** We can derive the following upper bound:

$$\mathbb{E}_{\tilde{X},U}\mathbb{E}_{q(\mathbf{e},\phi|\tilde{X}_U)}\mathrm{KL}(\mathbf{Q}_{\tilde{\mathbf{J}},U}\|\mathbf{Q}_{\tilde{\mathbf{J}}}) \leq I(\mathbf{e},\phi;U|\tilde{X}) + I(\tilde{\mathbf{J}};U|\mathbf{e},\phi,\tilde{X}). \tag{8}$$

Since $\tilde{X}_U = S$, the data processing inequality implies that $I(\mathbf{e},\phi;U|\tilde{X}) \leq I(\mathbf{e},\phi;S)$. This quantity captures *how much information about the training data is retained in the encoder*, thereby reflecting the degree of overfitting of the encoder parameters. The term $I(\tilde{\mathbf{J}};U|\mathbf{e},\phi,\tilde{X})$ can be viewed as a regularization term for the LVs, analogous to the IB hypothesis; see Appendix D.6 for further details.

Next, we investigate whether each term in Eq. (8) exhibits asymptotic convergence as the sample size $n$ increases, which is a key requirement for a valid generalization error bound. We begin by analyzing the asymptotic behavior of $I(\tilde{\mathbf{J}};U|\mathbf{e},\phi,\tilde{X})$.

**Lemma 1.** *Let the posterior distribution over $J$ be deterministic as defined in Eq. (1), and we denote the composition of this mapping with the encoder $f_\phi$ by $f'_{\mathbf{e},\phi} : \mathcal{X} \to [K]$. If the function class to which $f'_{\mathbf{e},\phi}$ belongs has a finite Natarajan dimension, then $I(\tilde{\mathbf{J}};U|\mathbf{e},\phi,\tilde{X})/n = \mathcal{O}(\log n/n)$.*

This result implies that if the encoder is appropriately regularized, the quantity $I(\tilde{\mathbf{J}};U|\mathbf{e},\phi,\tilde{X})/n$ converges asymptotically to zero. We also empirically evaluated this term in practical settings (see Appendix G) and observed that it indeed decreases as the sample size $n$ increases.

Next, the CMI term $I(\mathbf{e},\phi;U|\tilde{X})$ has been extensively analyzed under the standard supersample setting of IT analysis [63]. Prior works have established its asymptotic convergence through various approaches, including algorithmic stability [63], analyses of specific optimization methods such as stochastic gradient descent (SGD) [78] and stochastic gradient Langevin dynamics (SGLD) [20], and complexity-based arguments using covering numbers [83], all showing that $I(\mathbf{e},\phi;U|\tilde{X})/n \to 0$ as $n \to \infty$. In conclusion, the term $\mathbb{E}_{\tilde{X},U}\mathbb{E}_{q(\mathbf{e},\phi|\tilde{X}_U)}\mathrm{KL}(\mathbf{Q}_{\tilde{\mathbf{J}},U}\|\mathbf{Q}_{\tilde{\mathbf{J}}}))/n$ can be shown to converge asymptotically under certain algorithmic conditions. For a detailed discussion, see Appendix D.8.

**Regarding** $\mathrm{KL}(\mathbf{Q}_{\mathbf{J},U}\|\mathbf{P})$**:** This term can be rewritten as $\mathrm{KL}(\mathbf{Q}_{\mathbf{J},U}\|\mathbf{P})/n = \frac{1}{n}\sum_{m=1}^{n}\mathrm{KL}(q(J_m|\mathbf{e},\phi,S_m)\|\pi(J_m|\mathbf{e},\phi))$, where the training data is selected via $U$, i.e., $\tilde{X}_U = S = (S_1,\ldots,S_n)$. This quantity corresponds to the *empirical KL divergence*, which also appears in the analysis of Mbacke et al. [46], and reflects the complexity of the LVs. Such a term is commonly used as the regularization term appearing in many VAE training procedures [39, 33, 64].

A key factor in minimizing $\mathrm{KL}(\mathbf{Q}_{\mathbf{J},U}\|\mathbf{P})$ is the choice of the prior $\mathbf{P}$. In VQ-VAEs, a uniform distribution is typically adopted [64]; however, is this choice optimal for minimizing the KL divergence? The following lemma addresses this question.

**Lemma 2.** *Assume that for any fixed training dataset $S = (s_1,\ldots,s_n)$ and any permutation $\tau$, the posterior satisfies permutation invariance, i.e., $q(\mathbf{e},\phi,\theta|S) = q(\mathbf{e},\phi,\theta|S^\tau)$, where $S^\tau = (s_{\tau_1},\ldots,s_{\tau_n})$. Then, the optimal prior that minimizes $\mathbb{E}_S\mathbb{E}_{q(\mathbf{e},\phi|S)}\mathrm{KL}(\mathbf{Q}_{\mathbf{J},U}\|\mathbf{P})$ is*

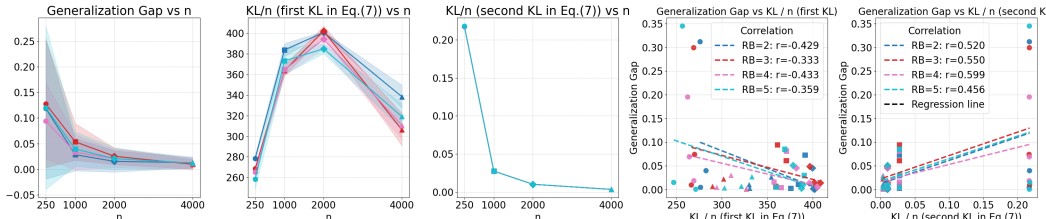

Figure 3: The behavior of the generalization gap and the two KL terms from Eq. (7) on the MNIST dataset ($K = 128$, $d_z = 64$). The three leftmost panels show the asymptotic behavior of the generalization gap, the first KL term, and the second KL term as a function of sample size $n$. The two rightmost panels show scatter plots correlating the generalization gap with the first KL term (fourth panel) and the second KL term (fifth panel). In these plots, the color indicates the number of decoder Residual Blocks (RB=2, 3, 4, or 5) and the marker shape indicates the sample size $n$. (Circle for $n = 250$, Square for $n = 1000$, Diamond for $n = 2000$, and Triangle for $n = 4000$).

given by $\mathbf{P}^* = \prod_{m=1}^n \mathbb{E}_{q(S_m|\mathbf{e},\phi)} q(J_m|\mathbf{e},\phi,S_m)$. *Moreover, under this prior, we obtain* $\mathbb{E}_S \mathbb{E}_{q(\mathbf{e},\phi|S)} \mathrm{KL}(\mathbf{Q}_{\mathbf{J},U}\|\mathbf{P}^*) = \sum_m I(J_m; S_m|\mathbf{e},\phi)$.

This connection provides insight into the choice of prior distributions in practical implementations—for instance, encouraging the use of mixture priors similar to the VampPrior [68] (see Appendix D.2 for further discussion). We also note that the assumption in Lemma 2, namely permutation invariance of the posterior, is standard in the analysis of randomized algorithms [42], and is satisfied by commonly used training methods such as SGD and SGLD [81].

Next, we present the asymptotic behavior of the empirical KL divergence term as follows:

**Lemma 3.** *Suppose the assumptions in Lemma 1 hold. Then, even under the optimal prior $\mathbf{P}^*$ given in Lemma 2, we have $\mathbb{E}_S \mathbb{E}_{q(\mathbf{e},\phi|S)} \mathrm{KL}(\mathbf{Q}_{\mathbf{J},U}\|\mathbf{P}^*)/n = \mathcal{O}(1)$.*

This result indicates that asymptotic convergence cannot be achieved, even when using the optimal prior $\mathbf{P}^*$, which minimizes $\mathrm{KL}(\mathbf{Q}_{\mathbf{J},U}\|\mathbf{P})$, to regularize the complexity of the encoder. Our empirical results provide validation for this theoretical finding. As shown in Figure 3 (left and middle panels), the first KL term, $\mathrm{KL}(\mathbf{Q}_{\mathbf{J},U}\|\mathbf{P})/n$, does not decrease as the sample size $n$ increases, confirming the behavior predicted by Lemma 3 (see Appendix G.4 for additional experimental results). Furthermore, the right two panels of Figure 3 illustrate the relationship between these terms. The second KL term exhibits a consistent positive correlation ($r \approx 0.46\text{-}0.60$) with the generalization gap across all tested decoder complexities. This suggests that the second KL term is the component that effectively captures generalization behavior. Conversely, the non-converging first KL term, $\mathrm{KL}(\mathbf{Q}_{\mathbf{J},U}\|\mathbf{P})/n$, shows a negative correlation, indicating it does not track generalization performance. This experiment empirically justifies our motivation to introduce the new permutation symmetric setting in Section 4 to eliminate this non-converging and poorly correlated term.

In the supervised learning context, it has similarly been observed that empirical KL terms analogous to $\mathrm{KL}(\mathbf{Q}_{\mathbf{J},U}\|\mathbf{P})/n$ do not necessarily converge, even for models that generalize well [21, 57]. Our findings are consistent with these results.

**Remark 2.** *Even when the posterior of $J$ is defined by Eq. (2), a comparable upper bound on the KL regularization term can still be derived by analyzing the encoder's complexity via metric entropy. For further details, see Section 4.2 and Appendix E.4.*

## 4 Proposed IT analysis under the new permutation symmetric setting

The observations presented in the previous section motivate the derivation of a generalization error bound that avoids explicit dependence on $\mathrm{KL}(\mathbf{Q}_{\mathbf{J},U}\|\mathbf{P})$. We conjecture that the appearance of this term in Theorem 2 arises from a fundamental limitation of the supersample setting, which necessitates the use of a data-independent prior $\mathbf{P}$ (as defined in Eq. (6)) to satisfy the necessary conditions (A) and (B) described in Section 3.1. To overcome this limitation, in this section, we introduce an extension of the supersample framework—namely, a novel *permutation symmetric setting*. This new setting enables the construction of a data-dependent prior that satisfies both conditions simultaneously,

thereby yielding a generalization error bound that achieves asymptotic convergence. All the proofs of this section are provided in Appendix E.

## 4.1 Permutation symmetric setting

To simultaneously satisfy the two conditions in Section 3.1, we propose randomly shuffling all $2n$ data points in $\tilde{X}$ using a uniform distribution and taking their expectation as the data-dependent prior distribution. By definition, this distribution is permutation-invariant, thereby satisfying conditions (A) and (B), allowing us to obtain the improved bound.

Formally, let us denote a random permutation of $[2n]$ as $\mathbf{T} = \{T_1, \ldots, T_{2n}\}$, where each permutation appears with uniform probability, $P(\mathbf{T}) = 1/(2n)!$. Given a supersample $\tilde{X} = (\tilde{X}_1, \ldots, \tilde{X}_{2n}) \in \mathcal{X}^{2n}$, a set of $2n$ RVs drawn i.i.d. from $\mathcal{D}$, we reorder the samples using $\mathbf{T}$ expressed as $\tilde{X}_{\mathbf{T}} = (\tilde{X}_{T_1}, \ldots, \tilde{X}_{T_{2n}})$. The first $n$ samples $(\tilde{X}_{T_1}, \ldots, \tilde{X}_{T_n})$ are used for the test dataset and the remaining $n$ samples $(\tilde{X}_{T_{n+1}}, \ldots, \tilde{X}_{T_{2n}})$ are used for the training dataset. We further express $\mathbf{T} = \{\mathbf{T}_0, \mathbf{T}_1\}$, and $\tilde{X}_{\mathbf{T}_0} = (\tilde{X}_{T_1}, \ldots, \tilde{X}_{T_n})$ and $\tilde{X}_{\mathbf{T}_1} = (\tilde{X}_{T_{n+1}}, \ldots, \tilde{X}_{T_{2n}})$ represent the test and training datasets, respectively.

Given $\tilde{X}$ and $\mathbf{T}$, we define the posterior distributions over the LVs of the test and training data, respectively, as $q(\bar{\mathbf{J}}|\mathbf{e}, \phi, \tilde{X}_{\mathbf{T}_0}) := \prod_{m=1}^{n} q(\bar{J}_m|\mathbf{e}, \phi, \tilde{X}_{T_m})$, $q(\mathbf{J}|\mathbf{e}, \phi, \tilde{X}_{\mathbf{T}_1}) := \prod_{m=1}^{n} q(J_m|\mathbf{e}, \phi, \tilde{X}_{T_{n+m}})$. We then define the joint posterior distribution as $\mathbf{Q}_{\tilde{\mathbf{J}}, \mathbf{T}} := q(\bar{\mathbf{J}}|\mathbf{e}, \phi, \tilde{X}_{\mathbf{T}_0}) q(\mathbf{J}|\mathbf{e}, \phi, \tilde{X}_{\mathbf{T}_1})$.

Finally, we define our new data-dependent prior as

$$\mathbf{Q}_{\tilde{\mathbf{J}}} := \mathbb{E}_{\mathbf{T}} \mathbf{Q}_{\tilde{\mathbf{J}}, \mathbf{T}} = \mathbb{E}_{\mathbf{T}} q(\bar{\mathbf{J}}|\mathbf{e}, \phi, \tilde{X}_{\mathbf{T}_0}) q(\mathbf{J}|\mathbf{e}, \phi, \tilde{X}_{\mathbf{T}_1}). \tag{9}$$

We refer to these settings as **the permutation symmetric (supersample) setting**. The following is our main result.

**Theorem 3.** *Under Assumptions 1 and the permutation symmetric setting, we have*

$$\mathrm{gen}(n, \mathcal{D}) \leq 3\Delta \sqrt{\frac{\mathbb{E}_{\tilde{X}, \mathbf{T}} \mathbb{E}_{q(\mathbf{e}, \phi|\tilde{X}_{\mathbf{T}_1})} \mathrm{KL}(\mathbf{Q}_{\tilde{\mathbf{J}}, \mathbf{T}} \| \mathbf{Q}_{\tilde{\mathbf{J}}})}{n}} + \frac{\Delta}{\sqrt{n}}.$$

**Remark 3.** *Unlike the existing supersample setting, where $\{U_m\}$s are independent, the elements of $\mathbf{T}$ are dependent, which makes the analysis more complicated.*

**Explanation of Theorem 3:** Similar to Theorem 2, this bound is *independent of the decoder $g_\theta$*. The key difference is that the empirical KL term, $\mathrm{KL}(\mathbf{Q}_{\mathbf{J}, U} \| \mathbf{P})$, is eliminated owing to our new data-dependent prior distribution $\mathbf{Q}_{\tilde{\mathbf{J}}}$. The proposed permutation satisfies both conditions (A) and (B) in Section 3.1, eliminating the need for a data-independent prior $\mathbf{P}$.

Next, we analyze the KL term in the bound. Similar to Eq. (8), we have

$$\mathbb{E}_{\tilde{X}, \mathbf{T}} \mathbb{E}_{q(\mathbf{e}, \phi|\tilde{X}_{\mathbf{T}_1})} \mathrm{KL}(\mathbf{Q}_{\tilde{\mathbf{J}}, \mathbf{T}} \| \mathbf{Q}_{\tilde{\mathbf{J}}}) \leq I(\mathbf{e}, \phi; \mathbf{T}|\tilde{X}) + I(\tilde{\mathbf{J}}; \mathbf{T}|\mathbf{e}, \phi, \tilde{X}).$$

Since $\tilde{X}_{\mathbf{T}_1}$ corresponds to the training dataset $S$, $I(\mathbf{e}, \phi; \mathbf{T}|\tilde{X}) \leq I(\mathbf{e}, \phi; S)$ holds. Then, we can show that our generalization bound becomes

$$\mathrm{gen}(n, \mathcal{D}) \leq 3\Delta \sqrt{\frac{I(\mathbf{e}, \phi; S) + I(\tilde{\mathbf{J}}; \mathbf{T}|\mathbf{e}, \phi, \tilde{X})}{n}} + \frac{\Delta}{\sqrt{n}}. \tag{10}$$

Our bound consists of the complexity of LV ($I(\tilde{\mathbf{J}}; U|\mathbf{e}, \phi, \tilde{X})$) and the overfitting caused by learning the encoder parameters ($I(\mathbf{e}, \phi; S)$) similar to Theorem 2. This implies the two key factors identified in Theorem 2 of Kawaguchi et al. [38]: how much information the LV retains from the input data and how much information from the training dataset is used to train the encoder.

As discussed in Section 3.2, when using a sufficiently regularized deterministic encoder, $f'_{\mathbf{e}, \phi} : \mathcal{X} \to [K]$, the CMI term satisfies $I(\tilde{\mathbf{J}}; U|\mathbf{e}, \phi, \tilde{X})/n = \mathcal{O}(\log n/n)$; see Appendix D.7 for details. The parameter overfitting term can be controlled by specifying the training algorithm, as discussed in

Section 3.2. Under these conditions, the generalization bound decreases as $n \to \infty$, meaning that Theorem 3 successfully characterizes generalization.

**Comparison with Theorem 2:** Although Theorem 3 shares a similar structure with Theorem 2, it introduces a refined shuffling strategy with $\mathbf{T}$, which resolves the issues of the supersample settings as discussed in Section 3.2. This shuffling is based on the fact that the marginal distribution of the dataset, which is invariant under permutation, can be expressed by the LV model. This new symmetry allows defining a data-dependent prior that satisfies necessary conditions while preserving decoder independence. On the other hand, the shuffling in Theorem 2 is based on the supersample setting and suitable for supervised learning, where overfitting is measured by swapping test and training data points. Practically, however, Theorem 2 relies on an $n$-dimensional variable, $U$ (with independent components), which facilitates CMI estimation and algorithm design. In contrast, Theorem 3 uses a $2n$-dimensional variable, $\mathbf{T}$ (with dependent components), which is theoretically more preferable but more difficult to estimate the CMI.

## 4.2 Generalization bound based on metric entropy

When using a softmax distribution in Eq. (2) for $J$, we show that the generalization bound is governed by the *metric entropy* under the permutation symmetric setting. Consequently, it does not require specifying a learning algorithm, which is required to discuss the convergence of Theorem 3 and provides a *uniform convergence bound* that depends solely on the function class of the encoder.

Let $\mathcal{F}$ be the encoder function class equipped with the metric $\|\cdot\|_\infty$. Given $x^n := (x_1, \ldots, x_n) \in \mathcal{X}^n$, define the pseudo-metric $d_n$ on $\mathcal{F}$ as $d_n(f, g) := \max_{i \in [n]} \|f(x_i) - g(x_i)\|_\infty$ for $f, g \in \mathcal{F}$. The $\delta$-covering number of $\mathcal{F}$ with respect to $d_n$ is denoted as $\mathcal{N}(\delta, \mathcal{F}, x^n)$, and we define $\mathcal{N}(\delta, \mathcal{F}, n) := \sup_{x^n \in \mathcal{X}^n} \mathcal{N}(\delta, \mathcal{F}, x^n)$.

**Theorem 4.** *Assume that there exists a positive constant $\Delta_z$ such that $\sup_{z,z' \in \mathcal{Z}} \|z - z'\| < \Delta_z$. Then, when using Eq. (2) and under the same setting as Theorem 3, for any $\delta \in (0, 1]$, we have*

$$\mathrm{gen}(n, \mathcal{D}) \leq 4\Delta \sqrt{2\beta n \delta \Delta_z} + 3\Delta \sqrt{\frac{2 \log \mathcal{N}(\delta, \mathcal{F}, 2n)}{n}} + \frac{\Delta}{\sqrt{n}}.$$

We note that the parameter overfitting term does not appear in the bound. Since the encoder is parameterized by $\phi \in \mathbb{R}^{d_\phi}$, the metric entropy is $\mathcal{O}(d_\phi d_z \log(1/\delta))$ [74]. Setting $\delta = \mathcal{O}(1/n)$ gives $\mathrm{gen}(n, \mathcal{D}) = \mathcal{O}\left(\sqrt{d_\phi d_z \log n / n}\right)$. This result suggests that regularizing the complexity of the encoder improves generalization, whereas the complexity of the decoder has limited influence on the generalization. See Appendix E.3 for the proof and further discussion.

## 5 IT analysis for data generation performance

Mbacke et al. [46] provided statistical guarantees for the generalization error and *data generation performance* of VAEs, albeit under the strong assumption of an *untrained* decoder. Building on their approach, we provide a theoretical guarantee for the data generation performance of VQ-VAEs from an IT analysis perspective when both the encoder and decoder are trained jointly.

We first briefly summarize the data generation process in VQ-VAEs. After training, new data is generated by sampling an index $J$ from a prior distribution, $\pi(J|\mathbf{e}, \phi)$, often chosen as a uniform distribution [64], and using the decoder network $g_\theta$ to reconstruct the corresponding latent representation $e_J$ from the learned codebook $\mathbf{e}$. Thus, the prior imposed on the latent representation is defined as $\pi(e = e_j|\mathbf{e}, \phi)$ for all $j = 1, \ldots, K$, and the data distribution generated through this procedure can be expressed as $\hat{\mu} := g_\theta \# \pi(e|\mathbf{e}, \phi)$, where $g_\theta \# \pi$ denotes the pushforward of the distribution $\pi$ by the decoder network. See Appendix F for the formal definition.

The following is the result of our analysis on the data generation performance of VQ-VAEs.

**Theorem 5.** *Suppose that $g_\theta$ is measurable for any $\theta$, and Assumption 1 holds. Then, for any data-independent prior $\pi(J|\mathbf{e}, \phi)$ as defined in Eq. (6), we have $\mathbb{E}_S \mathbb{E}_{q(\mathbf{e},\phi,\theta|S)} W_2^2(\mathcal{D}, \hat{\mu}) \leq \frac{2\Delta}{\sqrt{n}} +$*

$$\mathbb{E}_{S} \mathbb{E}_{q(\mathbf{e},\phi,\theta|S)} \left[ \frac{2}{n} \sum_{m=1}^{n} \mathbb{E}_{q(J_m|\mathbf{e},\phi,S_m)} l(S_m, g_\theta(e_{J_m})) + 4\Delta \sqrt{\frac{2}{n} \sum_{m=1}^{n} \mathrm{KL}(q(J_m|\mathbf{e}, \phi, S_m) \| \pi(J|\mathbf{e}, \phi))} \right],$$

*where $W_2(\mathcal{D}, \hat{\mu})$ is the 2-Wasserstein distance between the data distribution $\mathcal{D}$ and the generated-data distribution $\hat{\mu}$.*

The complete proof can be seen in Appendix F. The results indicate that the quality of approximating $\mathcal{D}$ by $\hat{\mu}$ can be enhanced by minimizing the reconstruction loss and the KL regularization term on LVs, which aligns with common training strategies for VQ-VAEs. Furthermore, this bound holds for *any* prior that satisfies the conditions outlined in Theorem 2. Thus, designing a prior that reduces this bound could lead to improved data generation accuracy. One potential approach is to use the data-dependent prior defined in Eq. (9). Although this prior was originally designed to yield a tighter generalization error upper bound, our experiments reveal that it also provides practical benefits for the data generation task, consistently improving test performance over the baseline (see Table 4 in Appendix G). However, we do not claim this specific prior is optimal for minimizing the data generation bound. We expect that our findings will stimulate further discussions on prior designs that effectively improve the generalization performance and data generation capabilities of VQ-VAEs.

## 6 Conclusion and limitations

This work establishes decoder-independent generalization guarantees for VQ-VAEs. Across Theorems 2 to 4, we show that the generalization gap is governed by the encoder parameters $(\mathbf{e}, \phi)$ and the induced LVs, while the decoder complexity $(\theta)$ plays a limited role. This central finding is empirically supported by our extensive experiments (Appendix G.4).

Our theoretical analysis provides several actionable insights. The primary takeaway is that efforts to improve generalization should prioritize the design and regularization of the encoder architecture and LV complexity, rather than investing in an overly complex decoder. Furthermore, our work provides the first formal justification for the widely used practice of KL-based regularization on LVs (Theorems 2 and 5), confirming it functions as a valid regularizer for both generalization and data generation. Finally, our framework highlights the importance of prior design. Our analysis, in line with Sefidgaran et al. [57], shows how a data-dependent prior can improve performance. Our experiments (Table 4) validate this, demonstrating that a learned prior, which approximates a data-dependent prior, consistently outperforms the standard uniform prior in practice.

**Limitation:** Our findings have two main limitations, which point to important avenues for future research. The first limitation is that the upper bound presented in Theorem 3 is challenging to compute numerically, making it impractical as an evaluation metric at present (see Sections 3.2 and 4). This difficulty stems from the CMI term in Eq. (10), where $\mathbf{T}$ is a $2n$-dimensional dependent random variable. Consequently, standard numerical evaluation methods for CMI cannot be directly applied. Developing an alternative, computable bound is an essential next step. The second limitation is that our analysis is currently justified only for VQ-VAEs, which are based on discrete LVs. Our proofs rely on properties of discrete random variables (the codebook index $\tilde{J}$) and apply concentration inequalities for each assignment. These proof techniques cannot be immediately applied to models with continuous latent spaces, where such assignments are not available. While one may consider forcibly discretizing a continuous latent space, the resulting discretization error is non-negligible and would substantially affect the analysis. Extending our information-theoretic framework to continuous settings is therefore a crucial and non-trivial step for future work.

## Acknowledgments and Disclosure of Funding

We sincerely appreciate the anonymous reviewers for their insightful feedback. FF was supported by JSPS KAKENHI Grant Number JP23K16948. FF was supported by JST, PRESTO Grant Number JPMJPR22C8, Japan. MF was supported by KAKENHI Grant Number 25K21286, Japan.

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

# A    Notation used in the main paper

We summarize the notation we used in the main part of our paper.

| Category | Symbol | Meaning |
|---|---|---|
| Data and model | $n \in \mathbb{N}$ | The sample size |
| | $\mathcal{X}, \mathcal{Z}$ | A data and latent space |
| | $\mathcal{D}$ | An unknown data generating distribution |
| | $\Delta \in \mathbb{R}^+$ | A radius of a data space |
| | $X \in \mathcal{X} \subset \mathbb{R}^d$ | A data |
| | $S = \{S_i\}_{i=1}^n \in \mathcal{X}^n$ | A training dataset |
| | $\mathbf{e} = \{e_j\}_{j=1}^K \in \mathcal{Z}^K$ | A codebook, where $K$ is the size of a codebook |
| | $\phi \in \Phi \subset \mathbb{R}^{d_\phi}$ | An encoder parameter |
| | $\theta \in \Theta \subset \mathbb{R}^{d_\theta}$ | A decoder parameter |
| | $W = \{\mathbf{e}, \phi, \theta\}$ | A set of model parameters |
| | $f_\phi : \mathcal{X} \to \mathcal{Z}$ | An encoder network |
| | $g_\theta : \mathcal{Z} \to \mathcal{X}$ | A decoder network |
| | $q(J|\mathbf{e}, \phi, X)$ | A posterior distribution over $J$ given $\mathbf{e}, \phi, X$ |
| | $\beta \in \mathbb{R}^+$ | A temperature parameter used in a softmax |
| | $\mathcal{N}(\delta, \mathcal{F}, n)$ | A $\delta$-covering number with $n$ input for the encoder function class $\mathcal{F}$ |
| Algorithm and loss functions | $\mathcal{A} : \mathcal{X}^n \to \mathcal{W}$ | A randomized algorithm |
| | $q(\mathbf{e}, \phi, \theta|S)$ | A randomized algorithm given $S$ |
| | $l : \mathcal{X} \times \mathcal{X} \to \mathbb{R}$ | a reconstruction loss function |
| | $l_0 : \mathcal{W} \times \mathcal{X} \to \mathbb{R}$ | An expected loss function over $J$ |
| | $\mathrm{gen}(\mu, \mathcal{D})$ | The expected generalization error based on a reconstruction loss |
| | $W_2(\mathcal{D}, \hat{\mu})$ | The 2-Wasserstein distance between $\mathcal{D}$ and $\hat{\mu}$ |
| Supersample setting | $\tilde{X} \in \mathcal{X}^{2n}$ | A supersample used in the IT analysis |
| | $\tilde{X}_m$ | The $m$-th row of $\tilde{X}$ |
| | $U = (U_1, \ldots, U_n) \sim \mathrm{Uniform}(\{0,1\}^n)$ | Random index used in the IT analysis |
| | $\tilde{X}_U := (\tilde{X}_{m,U_m})_{m=1}^n$ | A training dataset in the supersample setting |
| | $\tilde{X}_{\bar{U}} := (\tilde{X}_{m,\bar{U}_m})_{m=1}^n$ | A test dataset in the supersample setting, where $\bar{U}_m = 1 - U_m$ |
| | $q(\mathbf{J}|\mathbf{e}, \phi, \tilde{X}_U) := \prod_{m=1}^n q(J_m|\mathbf{e}, \phi, \tilde{X}_{m,U_m})$ | A joint distribution over index on the training dataset |
| | $q(\bar{\mathbf{J}}|\mathbf{e}, \phi, \tilde{X}_{\bar{U}}) := \prod_{m=1}^n q(\bar{J}_m|\mathbf{e}, \phi, \tilde{X}_{m,\bar{U}_m})$ | A joint distribution over index on the test dataset |
| | $\mathbf{Q}_{\bar{\mathbf{J}},U} := q(\bar{\mathbf{J}}|\mathbf{e}, \phi, \tilde{X}_{\bar{U}})q(\mathbf{J}|\mathbf{e}, \phi, \tilde{X}_U)$ | A joint posterior distribution over $J$ |
| | $\mathbf{Q}_{\bar{\mathbf{J}}} := \mathbb{E}_U q(\bar{\mathbf{J}}|\mathbf{e}, \phi, \tilde{X}_{\bar{U}})q(\mathbf{J}|\mathbf{e}, \phi, \tilde{X}_U)$ | A data-dependent prior distribution over $J$ |
| | $\pi(J|\mathbf{e}, \phi)$ | A data-independent prior distribution over $J$ |
| Permutation symmetric setting | $\mathbf{T} = \{T_1, \ldots, T_{2n}\} \sim P(\mathbf{T}) = 1/(2n)!$ | A random permutation following a uniform distirubiton |
| | $\tilde{X}_{\mathbf{T}} = (\tilde{X}_{T_1}, \ldots, \tilde{X}_{T_{2n}})$ | Randomly permuted supersamples |
| | $\tilde{X}_{\mathbf{T}_0} = (\tilde{X}_{T_1}, \ldots, \tilde{X}_{T_n})$ | The test dataset |
| | $\tilde{X}_{\mathbf{T}_1} = (\tilde{X}_{T_{n+1}}, \ldots, \tilde{X}_{T_{2n}})$ | The training dataset |
| | $q(\bar{\mathbf{J}}|\mathbf{e}, \phi, \tilde{X}_{\mathbf{T}_0}) = \prod_{m=1}^n q(\bar{J}_m|\mathbf{e}, \phi, \tilde{X}_{T_m})$ | A joint distribution over index on the test dataset |
| | $q(\mathbf{J}|\mathbf{e}, \phi, \tilde{X}_{\mathbf{T}_1}) = \prod_{m=1}^n q(J_m|\mathbf{e}, \phi, \tilde{X}_{T_{n+m}})$ | A joint distribution over index on the training dataset |
| | $\mathbf{Q}_{\bar{\mathbf{J}},\mathbf{T}} = q(\bar{\mathbf{J}}|\mathbf{e}, \phi, \tilde{X}_{\mathbf{T}_0})q(\mathbf{J}|\mathbf{e}, \phi, \tilde{X}_{\mathbf{T}_1})$ | A joint posterior distribution over $J$ |
| | $\mathbf{Q}_{\bar{\mathbf{J}}} = \mathbb{E}_{\mathbf{T}} q(\bar{\mathbf{J}}|\mathbf{e}, \phi, \tilde{X}_{\mathbf{T}_0})q(\mathbf{J}|\mathbf{e}, \phi, \tilde{X}_{\mathbf{T}_1})$ | A data-dependent prior distribution over $J$ |

# B    Additional discussion and related work

Here, we provide additional discussion and a comparison between our study and existing work.

## B.1    Related work

Here we briefly introduce additional related existing work, especially about the IT analysis. In IT analysis [83], the generalization error is evaluated on the basis of the MI between learned parameters and training data. This approach is closely related to the PAC-Bayes theory and has been extended through supersample settings [63] to exploit the symmetry between test and training data. This setting has been applied to the study of generalization based on outputs of functions [30], losses [32, 79], and hypothesis entropy [17]. The relationship between IT analysis and the IB hypothesis has been discussed from numerical and algorithmic perspectives [80, 44]. More recently, Sefidgaran et al. [57] theoretically studied latent variable models using IT analysis, demonstrating that generalization can be characterized by the complexity of the encoder and latent variables without relying on decoder information. They also developed a theoretical link among IT analysis, the IB hypothesis, and MDL by using compression bounds [10].

There exist several analyses focusing on VQ-VAE and related architectures. Vuong [73] investigated the supervised setting of vector-quantized models, whereas our analysis is purely unsupervised. Beyond the supervised–unsupervised distinction, our work differs from theirs in several fundamental

ways. First, they study a continuous and differentiable relaxation of the discrete latent-variable model, making it more amenable to optimization analysis. Second, this relaxation allows their framework to examine not only the generalization gap but also how the magnitude of the training loss itself is influenced by the discrete representation. Finally, their generalization guarantees rely on uniform convergence bounds, effectively reducing the problem to $K$-class clustering. In contrast, our analysis is based on algorithm-dependent, information-theoretic bounds, following the line of work by Sefidgaran et al. [57].

Classical quantization theory also provides useful insights. Pollard [52] and Telgarsky & Dasgupta [65] established asymptotic consistency results for $k$-means clustering. Although their setting—clustering without deep models—differs significantly from ours, the quantization procedure they analyze is conceptually related to the discrete latent representations used in VQ-VAE. Importantly, Vuong [73] build their analysis upon these classical results. The main distinction is that the latter works provide asymptotic guarantees specific to clustering, whereas our analysis provides finite-sample, non-asymptotic guarantees for deep generative models. Nonetheless, the connection highlights how classical quantization theory can inform the study of modern deep architectures, and it suggests that such tools may prove valuable when extending our framework to analyze loss minimization in addition to generalization.

## B.2    Comparison with existing bounds

Here, we compare our bounds with those in existing work. Theorem 2 resembles the results of Mbacke et al. [46] since both bounds include the empirical KL term in the upper bounds, and the posterior distribution corresponds to the variational posterior distribution. The key difference is that Mbacke et al. [46] assumed a fixed decoder, whereas our analysis incorporates the learning process under the assumption of a discrete latent space and a squared reconstruction loss. Another distinction is that their generalization bound does not become 0 as $n \to \infty$ due to two reasons. One is the presence of the empirical KL term, which we address in Theorem 3 using permutation symmetry. Our technique can be regarded as developing the appropriate prior distribution in PAC-Bayes bound. The second reason is the presence of the average distance $\frac{1}{n} \sum_{m=1}^{n} \mathbb{E}_X \|X - S_m\|$ in the existing bound, which is inherent to the data distribution and may not vanish as $n \to \infty$. Our use of the squared loss in the analysis mitigates this problematic term, as detailed in Appendix D.1.

Our proof techniques are based on Sefidgaran et al. [57]. However, we could not directly apply their methods, as the reconstruction loss reuses input data, unlike in classification settings. We resolve this by combining the data regeneration technique used in the proof of Mbacke et al. [46]. Additionally, we introduced a new permutation symmetric setting, leading to a bound that controls mutual information in Theorem 3. Our setting is closely related to the type-2 symmetry proposed in Sefidgaran et al. [57], which involves random permutations selecting $n$ indices from $2n$ with a uniform distribution $1/\binom{2n}{n}$, whereas our setting requires the consideration of the order of the permutation index to evaluate the exponential moment (see Appendix E.1). Finally, we theoretically studied the behavior of the CMI (Theorem 4) focusing on the complexity of the encoder, whereas Sefidgaran et al. [57] provided the bounds based on the CMI without such discussion.

The existing analyses based on the IB hypothesis [71, 28, 38, 72] assumed that both the latent variables and data are discrete, and their obtained bounds explicitly depend on the latent space size or show exponential dependence on the MI. In contrast, we assume that only latent variables are discrete and the resulting bound does not explicitly depend on the number of discrete states nor exhibit exponential dependence on MI. Furthermore, our bound shows the dependency on $d_z$ not $K$, which is the significant difference compared with existing bounds.

## B.3    Discussion and comparison of our prior and posterior and existing work

Here, we explain how the prior distribution is used in our proof and why two prior distributions are introduced in our bound. First, the IT analysis with supersample reformulates generalization analysis as the problem of estimating which samples were used for training when data is randomly shuffled based on $U$. If this estimation is difficult, our model generalizes well. In the basic IT analysis (Theorem 1), such difficulty is measured by the CMI between $U$ and the loss function $l_0$.

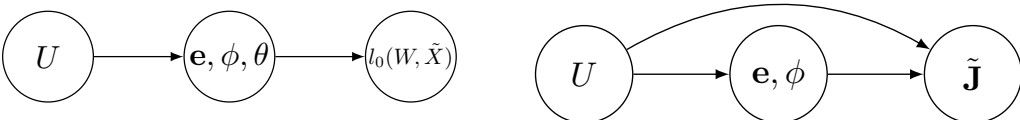

Figure 4: Graphical models illustrating the different dependency structures of the random variables considered in the basic IT analysis and in this study. The left figure represents the dependency structure in the basic IT analysis, which simply evaluates the loss function in supervised learning settings, whereas the right figure corresponds to our analysis in the unsupervised learning setting.

Of course, such shuffling is not performed in actual algorithms; it is introduced only for theoretical analysis using the Donsker-Varadhan inequality [25], where such shuffling is defined by the prior and posterior distributions dependent on $U$.

In basic IT analyses (Theorem 1) for supervised learning, by shuffling with $U$, we observe how the loss $l_0(W, \tilde{X})$ changes. Here, the goal is to estimate $U$ from the observed losses. As depicted in Figure 4, $U$ and $l_0(W, \tilde{X})$ depend on all parameters, including the decoder, resulting in a bound that depends on all parameters.

Our goal is to eliminate the dependency between the decoder and latent variables (LVs). To achieve this, we introduce a prior and posterior that establish the dependency as depicted in Figure 4. The key idea is that by introducing a new dependency between $U$ and LVs, we can directly shuffle $U$, leading to a bound that isolates the role of LVs without involving the decoder. For additional discussions on the necessary conditions for the prior, see Appendix D.2.

Finally, we show the additional explanation of Figure 1. The figure illustrates the difference between the existing fCMI and our new CMI. The left figure illustrates the setting of existing fCMI where $\widetilde{J}$ follows the distribution in the setting of Eq. (5), see Appnexdix C.2 for the detail. Thus, in the existing fCMI, $\widetilde{J}$ and $U$ are conditionally independent given $\mathbf{e}$ and $\phi$ and $\tilde{X}$. On the other hand, the right figure is our setting and there is an edge between $U$ and $\widetilde{J}$ directly, and thus $\widetilde{J}$ and $U$ are conditionally independent given $\mathbf{e}$ and $\phi$ and $\tilde{X}$, which results in the difference of existing fCMI and our CMI. See Appendix D.5 and Appendix D.3 for the additional discussion about the fCMI.

## C   Proofs for Section 2 and additional discussion

### C.1   Proof of Theorem 1

This is just the consequence of the existing eCMI bound [32]. We can confirm this as follows;

Note that the generalization error can be expressed as the supersample

$$
\begin{aligned}
&\text{gen}(n, \mathcal{D}) \\
&= \left| \mathbb{E}_{S,X} \mathbb{E}_{q(\mathbf{e},\phi,\theta|S)} \Big( \mathbb{E}_{q(J|\mathbf{e},\phi,X)} l(X, g_\theta(e_J)) - \frac{1}{n} \sum_{m=1}^{n} \mathbb{E}_{q(J_m|\mathbf{e},\phi,S_m)} l(S_m, g_\theta(e_{J_m})) \Big) \right| \\
&= \Bigg| \mathop{\mathbb{E}}_{\tilde{X},U} \mathop{\mathbb{E}}_{q(\mathbf{e},\phi,\theta|X_U)} \Big( \frac{1}{n} \sum_{m=1}^{n} \mathbb{E}_{q(\bar{J}_m|\mathbf{e},\phi,X_{m,\bar{U}_m})} l(X_{m,\bar{U}_m}, g_\theta(e_{J_m})) \\
&\qquad\qquad - \frac{1}{n} \sum_{m=1}^{n} \mathbb{E}_{q(J_m|\mathbf{e},\phi,X_{m,U_m})} l((X_{m,U_m}, g_\theta(e_{J_m}))) \Big) \Bigg|.
\end{aligned}
$$

Given that the loss is bounded by $[0, \Delta]$, the integrated is a $\Delta$-sub-Gaussian random variable. Thus, from Hellström & Durisi [32], the generalization error bound that satisfies the $\sigma^2$ sub Gaussianity is bounded as $\sqrt{\frac{2\sigma^2}{n} I(l(\mathcal{A}(\tilde{X}_U), \tilde{X}); U|\tilde{X})}$, we obtain the result. Finally $I(l_0(W, \tilde{X}); U|\tilde{X}) \leq I(W; U|\tilde{X})$ holds by the data processing inequality.

## C.2 Proof for Eq. (5) and additional discussion

Here we prove Eq. (5). It is important to note that this upper bound is characterized by the CMI $I(l_0(W, \tilde{X}); U|\tilde{X})$. This CMI depends on the decoder and encoder information, distinguishing it from the results presented in our main Theorems 2 and 3, which do not require the decoder's information.

To clarify this distinction, let us introduce the necessary notation. Following the notation in Section 3.1, we define $\tilde{Y} = g_\theta(e_{\tilde{j}})$, where $g_\theta(e_{\tilde{j}})$ implies applying $g_\theta(\cdot)$ elementwise to $e_{\tilde{j}}$. Under these notations, we have the following relations:

$$I(l_0(W, \tilde{X}); U|\tilde{X}) \leq I(\tilde{Y}; U|\tilde{X}) \leq I(\theta; U|\tilde{X}) + I(e_{\tilde{\mathbf{j}}}; U|\tilde{X}, \theta),$$

where the first inequality is obtained by the data processing inequality (DPI) and the second inequality is obtained by the chain rule of CMI and the DPI. This result demonstrates that the decoder information cannot be eliminated from the basic IT bound, which clarifies the fundamental difference compared to our result (Theorems 2 and 3). Moreover, since the decoder and encoder are learned simultaneously using the same training data, they are not independent. This makes it unclear how the latent variables and the encoder's capacity affect generalization, as it is difficult to eliminate the decoder's dependency on them.

## C.3 Additional discussion when $K = 1$

Another limitation of the basic IT-bound arises when considering $K = 1$ as a limiting setting. From the definition of the squared loss, the generalization error is given by:

$$\text{gen}(n, \mathcal{D}) \leq \sqrt{\text{Var}[X] \frac{\mathbb{E}\|g_\theta(e)\|^2}{n}} \leq \frac{\Delta}{\sqrt{n}}. \tag{11}$$

The proof of this is described below. This upper bound is intuitive: for $K = 1$, the model effectively ignores the input data and embeds all samples into the same latent variable, which can be interpreted as a form of strong regularization. Consequently, the impact of overfitting due to training the decoder network is relatively limited, and the generalization error can be seen, in a sense, as being comparable to the inherent variability of the data itself.

The above observations motivate us to develop a more sophisticated generalization bound that explicitly captures the role of representation.

*Proof of Eq. (11).* Since $K = 1$, we express $\mathbf{e} = \{e\}$. By using the definition of the squared loss, we have

$$\text{gen}(n, \mathcal{D}) = \left| \mathbb{E}_S \mathbb{E}_{q(e, \phi, \theta|S)} \left( \mathbb{E}[X] - \frac{1}{n} \sum_{m=1}^n S_m \right) \cdot g_\theta(e)) \right|,$$

where we used the fact that the generated data always use $e$ as a latent variable since $\mathbf{e} = \{e\}$ when $K = 1$. Then by using the Cauchy-Schwartz inequality, we have

$$\text{gen}(n, \mathcal{D}) \leq \sqrt{\text{Var}[X] \frac{\mathbb{E}\|g_\theta(e)\|^2}{n}} \leq \frac{\Delta}{\sqrt{n}},$$

where we used the fact that the diameter of the instance space is bounded by $\Delta$. $\qquad\square$

## D Proofs for Section 3

In the proofs, we repeatedly use the following type of exponential moment inequality, which is often used in the proof of McDiarmid's inequality. A function $f : \mathcal{X}^n \to \mathbb{R}$ has the bounded differences property if for some nonnegative constants $c_1, \ldots, c_n$, the following holds for all $i$:

$$\sup_{x_1, \ldots, x_n, x_i' \in \mathcal{X}} |f(x_1, \ldots, x_n) - f(x_1, \ldots, x_{i-1}, x_i', x_{i+1}, \ldots, x_n)| \leq c_i, \quad 1 \leq i \leq n.$$

Assuming $X_1, \ldots, X_n$ are independent random variables taking values in $\mathcal{X}$, we have the following lemma:

**Lemma 4** (Used in the proof of McDiarmid's inequality). *Given a function $f$ with the bounded differences property, for any $t \in \mathbb{R}$, we have:*

$$\mathbb{E}\left[e^{t(f(X_1,\ldots,X_n)-\mathbb{E}[f(X_1,\ldots,X_n)])}\right] \le e^{\frac{t^2}{8}\sum_{i=1}^n c_i^2}.$$

## D.1 Proof of Theorem 2

We express $q(\tilde{\mathbf{J}}|\mathbf{e}, \phi, \tilde{X}) = q(\bar{\mathbf{J}}, \mathbf{J}|\mathbf{e}, \phi, \tilde{X}_{\bar{U}}, \tilde{X}_U) = q(\bar{\mathbf{J}}|\mathbf{e}, \phi, \tilde{X}_{\bar{U}})q(\mathbf{J}|\mathbf{e}, \phi, \tilde{X}_U)$. Hereinafter, we simplify the notation by expressing $\tilde{X}$ as $X$. For simplification in the proof, we omit the absolute operation for the generalization gap. The reverse bound can be proven in a similar manner. We first express the generalization error of the reconstruction loss using the supersample as follows

$$\sum_{k=1}^K \frac{1}{n} \sum_{m=1}^n \mathbb{E}_{q(\bar{J}_m|\mathbf{e},\phi,X_{m,\bar{U}_m})q(\mathbf{e},\phi,\theta|X_U)} l(X_{m,\bar{U}_m}, g_\theta(e_k)) \mathbb{1}_{k=\bar{J}_m}$$

$$-\sum_{k=1}^K \frac{1}{n} \sum_{m=1}^n \mathbb{E}_{q(J_m|\mathbf{e},\phi,X_{m,U_m})q(\mathbf{e},\phi,\theta|X_U)} l((X_{m,U_m}, g_\theta(e_k)) \mathbb{1}_{k=J_m}$$

$$=\sum_{k=1}^K \frac{1}{n} \sum_{m=1}^n \mathbb{E}_{q(\bar{J}_m|\mathbf{e},\phi,X_{m,\bar{U}_m})q(\mathbf{e},\phi,\theta|X_U)} \|X_{m,\bar{U}_m} - g_\theta(e_k)\|^2 \mathbb{1}_{k=\bar{J}_m}$$

$$-\sum_{k=1}^K \frac{1}{n} \sum_{m=1}^n \mathbb{E}_{q(J_m|\mathbf{e},\phi,X_{m,U_m})q(\mathbf{e},\phi,\theta|X_U)} \|X_{m,U_m} - g_\theta(e_k)\|^2 \mathbb{1}_{k=J_m}, \quad (12)$$

where the first term corresponds to the test loss and the second term corresponds to the training loss.

Recall the learning algorithm and posterior distribution:

$$\mathbf{e}, \phi, \theta \sim q(\mathbf{e}, \phi, \theta|X_U),$$
$$J_m \sim q(\mathbf{J}|\mathbf{e}, \phi, S_m).$$

Here $\mathbf{e} = \{e_1, \ldots, e_K\}$ is the codebook, and $J$ and $\mathbf{J} = \{J_1, \ldots, j_n\}$ represents the index chosen from the codebook.

Conditioned on $X$ and $U$, we then decompose Eq. (12) as follows

$$\sum_{k=1}^K \frac{1}{n} \sum_{m=1}^n \mathbb{E}_{q(\mathbf{e},\phi,\theta|X_U)} l(X_{m,\bar{U}_m}, g_\theta(e_k)) \mathbb{E}_{q(\bar{J}_m|\mathbf{e},\phi,X_{m,\bar{U}_m})} \mathbb{1}_{k=\bar{J}_m}$$

$$-\sum_{k=1}^K \frac{1}{n} \sum_{m=1}^n \mathbb{E}_{q(\mathbf{e},\phi,\theta|X_U)} l(X_{m,\bar{U}_m}, g_\theta(e_k)) \mathbb{E}_{q(J_m|\mathbf{e},\phi,X_{m,U_m})} \mathbb{1}_{k=J_m}$$

$$+\sum_{k=1}^K \frac{1}{n} \sum_{m=1}^n \mathbb{E}_{q(\mathbf{e},\phi,\theta|X_U)} l(X_{m,\bar{U}_m}, g_\theta(e_k)) \mathbb{E}_{q(J_m|\mathbf{e},\phi,X_{m,U_m})} \mathbb{1}_{k=J_m}$$

$$-\sum_{k=1}^K \frac{1}{n} \sum_{m=1}^n \mathbb{E}_{q(\mathbf{e},\phi,\theta|X_U)} l(X_{m,U_m}, g_\theta(e_k)) \mathbb{E}_{q(J_m|\mathbf{e},\phi,X_{m,U_m})} \mathbb{1}_{k=J_m}. \quad (13)$$

We will separately upper bound these terms.

### D.1.1 Bounding first and second terms

The decomposition of the generalization error, as shown in Eq. (13), allows us to bound the first and second terms as follows.

We apply Donsker-Varadhan's inequality between the following two distributions:

$$\mathbf{Q} := P(U)q(\mathbf{e}, \phi, \theta|X_U)q(\bar{\mathbf{J}}, \mathbf{J}|\mathbf{e}, \phi, X_{\bar{U}}, X_U)$$
$$\mathbf{P}_S := P(U)q(\mathbf{e}, \phi, \theta|X_U) \mathbb{E}_{P(U')} q(\bar{\mathbf{J}}, \mathbf{J}|\mathbf{e}, \phi, X_{\bar{U}'}, X_{U'}). \quad (14)$$

These correspond to the posterior and data-dependent prior distributions defined in Section 3.1.

Then, for any $\lambda \in \mathbb{R}^+$, we have

$$\sum_{k=1}^{K} \frac{1}{n} \sum_{m=1}^{n} \mathbb{E}_{q(\mathbf{e},\phi,\theta|X_U)} l(X_{m,\bar{U}_m}, g_\theta(e_k)) \left( \mathbb{E}_{q(\bar{J}_m|\mathbf{e},\phi,X_{m,\bar{U}_m})} \mathbb{1}_{k=\bar{J}_m} - \mathbb{E}_{q(J_m|\mathbf{e},\phi,X_{m,U_m})} \mathbb{1}_{k=J_m} \right)$$

$$\leq \frac{1}{\lambda} \mathrm{KL}(\mathbf{Q}|\mathbf{P}_S) + \frac{1}{\lambda} \log \mathbb{E}_{\mathbf{P}_S} \exp \left( \frac{\lambda}{n} \sum_{k=1}^{K} \sum_{m=1}^{n} l(X_{m,\bar{U}_m}, g_\theta(e_k)) \left( \mathbb{1}_{k=\bar{J}_m} - \mathbb{1}_{k=J_m} \right) \right).$$

To simplify the notation, we express $\bar{\mathbf{J}} = \mathbf{J}_0$, $\bar{J}_m = J_{m,0}$, $\mathbf{J} = \mathbf{J}_1$, and $J_m = J_{m,1}$. Let $U''$ be a random variable taking $0, 1$ with a uniform distribution. Since $\mathbf{P}_S$ is symmetric with respect to the permutation of $\mathbf{J}_0$ and $\mathbf{J}_1$, we can bound the exponential moment as:

$$\log \mathbb{E}_{P(U)q(\mathbf{e},\phi,\theta|X_U)} \underset{P(U')}{\mathbb{E}} q(\mathbf{J}_0,\mathbf{J}_1|\mathbf{e},\phi,X_{\bar{U}'},X_{U'}) \exp \left( \frac{\lambda}{n} \sum_{k=1}^{K} \sum_{m=1}^{n} l(X_{m,\bar{U}_m}, g_\theta(e_k)) \left( \mathbb{1}_{k=J_{m,0}} - \mathbb{1}_{k=J_{m,1}} \right) \right)$$

$$= \log \mathbb{E}_{P(U)q(\mathbf{e},\phi,\theta|X_U)P(U'')^n} \underset{P(U')}{\mathbb{E}} q(\mathbf{J}_0,\mathbf{J}_1|\mathbf{e},\phi,X_{\bar{U}'},X_{U'})P(U'')^N$$

$$\exp \left( \frac{\lambda}{n} \sum_{k=1}^{K} \sum_{m=1}^{n} l(X_{m,\bar{U}_m}, g_\theta(e_k)) \left( \mathbb{1}_{k=J_{m,\bar{U}''}} - \mathbb{1}_{k=J_{m,U''}} \right) \right)$$

$$= \log \mathbb{E}_{P(U)q(\mathbf{e},\phi,\theta|X_U)} \underset{P(U')}{\mathbb{E}} q(\mathbf{J}_0,\mathbf{J}_1|\mathbf{e},\phi,X_{\bar{U}'},X_{U'}) \mathbb{E}_{P(U'')^n}$$

$$\exp \left( \frac{\lambda}{n} \sum_{k=1}^{K} \sum_{m=1}^{n} l(X_{m,\bar{U}_m}, g_\theta(e_k)) \left( \mathbb{1}_{k=J_{m,\bar{U}''}} - \mathbb{1}_{k=J_{m,U''}} \right) \right).$$

In the final line, we apply McDiarmid's inequality since $U''^n$ are $n$ i.i.d. random variables. To use McDiarmid's inequality in Lemma 4, we use the stability caused by replacing one of the elements of $n$ i.i.d. random variables. To estimate the coefficients of stability in Lemma 4, let $U''^n = (U_1'', \ldots, U_N'')$, then

$$\sup_{\{U_m''\}_{m=1}^n, U_{m'}'''} \left| \frac{\lambda}{n} \sum_{k=1}^{K} \sum_{m=1}^{n} l(X_{m,\bar{U}_m}, g_\theta(e_k)) \left( \mathbb{1}_{k=J_{m,\bar{U}_m''}} - \mathbb{1}_{k=J_{m,U_m''}} \right) \right. \tag{15}$$

$$- \frac{\lambda}{n} \sum_{k=1}^{K} \sum_{m \neq m'}^{n} l(X_{m,\bar{U}_m}, g_\theta(e_k)) \left( \mathbb{1}_{k=J_{m,\bar{U}_m''}} - \mathbb{1}_{k=J_{m,U_m''}} \right)$$

$$- \frac{\lambda}{n} \sum_{k=1}^{K} l(X_{m',\bar{U}_m'}, g_\theta(e_k)) \left( \mathbb{1}_{k=J_{m',\bar{U}_{m'}'''}} - \mathbb{1}_{k=J_{m',U_{m'}'''}} \right) \Bigg|$$

$$= \sup_{\{U_m''\}_{m=1}^n, U_{m'}'''} \left| \frac{\lambda}{n} \sum_{k=1}^{K} l(X_{m',\bar{U}_m'}, g_\theta(e_k)) \left( \mathbb{1}_{k=J_{m',\bar{U}_{m'}''}} - \mathbb{1}_{k=J_{m',U_{m'}''}} \right) \right.$$

$$- \frac{\lambda}{n} \sum_{k=1}^{K} l(X_{m',\bar{U}_m'}, g_\theta(e_k)) \left( \mathbb{1}_{k=J_{m',\bar{U}_{m'}'''}} - \mathbb{1}_{k=J_{m',U_{m'}'''}} \right) \Bigg| \leq \frac{2\lambda\Delta}{n}.$$

Here, the maximum change caused by replacing one element of $U''$ is $2\lambda\Delta/n$, thus, its log of the exponential moment is bounded by $(2\lambda\Delta/n)^2/8 \times n = \lambda^2\Delta^2/2n$. Thus from Lemma 4, we have

$$\log \mathbb{E}_{P(U)q(\mathbf{e},\phi,\theta|X_U)} \underset{P(U')}{\mathbb{E}} q(\mathbf{J}_0,\mathbf{J}_1|\mathbf{e},\phi,X_{\bar{U}'},X_{U'}) \exp \left( \frac{\lambda}{n} \sum_{k=1}^{K} \sum_{m=1}^{n} l(X_{m,\bar{U}_m}, g_\theta(e_k)) \left( \mathbb{1}_{k=J_{m,0}} - \mathbb{1}_{k=J_{m,1}} \right) \right)$$

$$\leq \frac{\lambda^2\Delta^2}{2n}.$$

The first and second terms in Eq. (13) are upper bounded by

$$\frac{1}{\lambda} \mathbb{E}_X \mathrm{KL}(\mathbf{Q}|\mathbf{P}_S) + \frac{\lambda\Delta^2}{2n}. \tag{16}$$

### D.1.2 Bounding third and fourth terms

Next, we upper bound the third and fourth terms in Eq. (13);

$$\sum_{k=1}^{K}\frac{1}{n}\sum_{m=1}^{n}\mathbb{E}_{q(\mathbf{e},\phi,\theta|X_U)}l(X_{m,\bar{U}_m},g_\theta(e_k))\mathbb{E}_{q(J_m|\mathbf{e},\phi,X_{m,U_m})}\mathbb{1}_{k=J_m}$$

$$-\sum_{k=1}^{K}\frac{1}{n}\sum_{m=1}^{n}\mathbb{E}_{q(\mathbf{e},\phi,\theta|X_U)}l(X_{m,U_m},g_\theta(e_k))\mathbb{E}_{q(J_m|\mathbf{e},\phi,X_{m,U_m})}\mathbb{1}_{k=J_m}. \tag{17}$$

We simplify the notation by expressing $\mathbb{E}_{q(J_m|\mathbf{e},\phi,X_{m,U_m})}\mathbb{1}_{k=J_m}$ as $P_{k,m}$ and use the square loss:

$$\mathbb{E}_{X,U}\sum_{k=1}^{K}\frac{1}{n}\sum_{m=1}^{n}\mathbb{E}_{q(\mathbf{e},\phi,\theta|X_U)}l(X_{m,\bar{U}_m},g_\theta(e_k))P_{k,m}-\sum_{k=1}^{K}\frac{1}{n}\sum_{m=1}^{n}\mathbb{E}_{q(\mathbf{e},\phi,\theta|X_U)}l(X_{m,U_m},g_\theta(e_k))P_{k,m}$$

$$=\mathbb{E}_{X,U}\sum_{k=1}^{K}\frac{1}{n}\sum_{m=1}^{n}\mathbb{E}_{q(\mathbf{e},\phi,\theta|X_U)}\left(\|X_{m,\bar{U}_m}\|^2-\|X_{m,U_m}\|^2\right)P_{k,m}$$

$$+\mathbb{E}_{X,U}\sum_{k=1}^{K}\frac{2}{n}\sum_{m=1}^{n}\mathbb{E}_{q(\mathbf{e},\phi,\theta|X_U)}\left(X_{m,\bar{U}_m}-X_{m,U_m}\right)\cdot g_\theta(e_k)P_{k,m}$$

$$=\mathbb{E}_{X,U}\frac{1}{n}\sum_{m=1}^{n}\left(\|X_{m,\bar{U}_m}\|^2-\|X_{m,U_m}\|^2\right)\mathbb{E}_{q(\mathbf{e},\phi,\theta|X_U)}\sum_{k=1}^{K}P_{k,m}$$

$$+\mathbb{E}_S\frac{2}{n}\sum_{m=1}^{n}\left(\mathbb{E}_XX-S_m\right)\cdot\mathbb{E}_{q(\mathbf{e},\phi,\theta|S)}\sum_{k=1}^{K}g_\theta(e_k)P_{k,m}$$

$$=\mathbb{E}_S\frac{2}{n}\sum_{m=1}^{n}\left(\mathbb{E}_XX-S_m\right)\cdot\mathbb{E}_{q(\mathbf{e},\phi,\theta|S)}\sum_{k=1}^{K}g_\theta(e_k)P_{k,m}, \tag{18}$$

where we express $S=(X_{1,U_1},\ldots,X_{n,U_n})=(S_1,\ldots,S_n)$ as the training samples. In the last inequality, we used $\sum_{k=1}^{K}P_{k,m}=1$ and $\mathbb{E}_{X,U}\frac{1}{n}\sum_{m=1}^{n}\left(\|X_{m,\bar{U}_m}\|^2-\|X_{m,U_m}\|^2\right)=0$ since $X$ and $U$ are i.i.d.

To evaluate the final line, we use the Donsker-Valadhan inequality between

$$\mathbf{Q}:=q(\mathbf{e},\phi,\theta|S)\prod_{m=1}^{n}q(J_m|\mathbf{e},\phi,S_m),$$

$$\mathbf{P}_S:=q(\mathbf{e},\phi,\theta|S)\prod_{m=1}^{n}\pi(J_m|\mathbf{e},\phi),$$

where $\pi(J_m|\mathbf{e},\phi)$ is the prior distribution, which never depends on the training data.

Then we have

$$\mathbb{E}_S\frac{2}{n}\sum_{m=1}^{n}\left(\mathbb{E}_XX-S_m\right)\cdot\mathbb{E}_{q(\mathbf{e},\phi,\theta|S)}\sum_{k=1}^{K}g_\theta(e_k)P_{k,m}$$

$$\leq\mathbb{E}_S\frac{1}{\lambda}\mathrm{KL}(\mathbf{Q}|\mathbf{P}_S)+\mathbb{E}_S\frac{1}{\lambda}\log\mathbb{E}_{\mathbf{P}_S}\exp\left(\frac{2\lambda}{n}\sum_{m=1}^{n}\left(\mathbb{E}_XS-X_m\right)\cdot\mathbb{E}_{q(\mathbf{e},\phi,\theta|S)}\sum_{k=1}^{K}g_\theta(e_k)\mathbb{1}_{k=J_m}\right)$$

$$\leq\mathbb{E}_S\frac{1}{\lambda}\mathrm{KL}(\mathbf{Q}|\mathbf{P}_S)$$

$$+\mathbb{E}_S\frac{1}{\lambda}\log\mathbb{E}_{\mathbf{P}_S}\exp\left(\frac{2\lambda}{n}\sum_{m=1}^{n}\left(\mathbb{E}_XX-S_m\right)\cdot\sum_{k=1}^{K}g_\theta(e_k)(\mathbb{1}_{k=J_m}-P''_{k,m})\right)$$

$$+\mathbb{E}_S\mathbb{E}_{\mathbf{P}_S}\frac{2}{n}\sum_{m=1}^{n}\left(\mathbb{E}_XX-S_m\right)\cdot\sum_{k=1}^{K}g_\theta(e_k)P''_{k,m}, \tag{19}$$

where $P''_{k,m} = \mathbb{E}_{q(J_m|\phi,\mathbf{e})}\mathbb{1}_{k=J_m}$. Clearly, this does not depend on the index $m$, so we express $P''_{k,m} = P''_k$. Then the last term becomes

$$\mathbb{E}_S\mathbb{E}_{\mathbf{P}_S}\frac{1}{n}\sum_{m=1}^{n}(\mathbb{E}_X X - S_m)\cdot\sum_{k=1}^{K}g_\theta(e_k)P''_k \leq \mathbb{E}_S\mathbb{E}_{\mathbf{P}_S}\left\|\mathbb{E}_X X - \frac{1}{n}\sum_{m=1}^{n}S_m\right\|\,\|\sum_{k=1}^{K}g_\theta(e_k)P''_k\|$$

$$\leq \mathbb{E}_S\left\|\mathbb{E}_X X - \frac{1}{n}\sum_{m=1}^{n}S_m\right\|\sqrt{\Delta}$$

$$\leq \sqrt{\Delta\,\mathrm{Var}\left(\frac{1}{n}\sum_{m=1}^{n}S_m\right)}$$

$$\leq \sqrt{\Delta\frac{\mathrm{Var}(X)}{n}}$$

$$\leq \sqrt{\frac{\Delta}{4n}}\sqrt{\Delta} = \frac{\Delta}{2\sqrt{n}}, \tag{20}$$

where we used the fact that the variance of random variables with bounded in $(a,b]$ is upper bounded by $(b-a)^2/4n$ (the extension to the $d$-dimensional random variable is straightforward) and thus, $\mathrm{Var}(X) \leq \Delta/4$. Then the exponential moment term becomes

$$\mathbb{E}_S\frac{1}{\lambda}\log\mathbb{E}_{\mathbf{P}_S}\exp\left(\frac{2\lambda}{n}\sum_{m=1}^{n}(\mathbb{E}_X X - S_m)\cdot\sum_{k=1}^{K}g_\theta(e_k)(\mathbb{1}_{k=J_m} - P''_{k,m})\right)$$

$$= \mathbb{E}_S\frac{1}{\lambda}\log\mathbb{E}_{\mathbf{P}_S}\exp\left(\frac{2\lambda}{n}\sum_{m=1}^{n}(\mathbb{E}_X X - S_m)\cdot\sum_{k=1}^{K}g_\theta(e_k)(\mathbb{1}_{k=J} - P''_k)\right).$$

Here we use the McDiarmid's inequality for $n$ random variables $\mathbf{J}$. Then we estimate the stability coefficient similarly to Eq. (15), which is upper bounded by $\lambda\Delta/n$. Then from Lemma 4, the exponential moment is bounded by $(2\lambda\Delta/n)^2/8 \times n = \lambda\Delta^2/2n$ Thus, the second term is upper bounded by

$$\frac{1}{\lambda}\mathrm{KL}(\mathbf{Q}|\mathbf{P}_S) + \frac{\lambda\Delta^2}{2n} + \frac{\Delta}{\sqrt{n}}. \tag{21}$$

By optimizing the first and second terms of Eqs. (16) and (21), we have

$$2\Delta\sqrt{\frac{(\mathbb{E}_{\tilde{X},U}\mathbb{E}_{q(\mathbf{e},\phi,\theta|X_U)}\mathrm{KL}(\mathbf{Q}_1\|\mathbf{Q}_2) + \mathbb{E}_S\mathbb{E}_{q(\mathbf{e},\phi,\theta|S)}\mathrm{KL}(\mathbf{Q}|\mathbf{P}_S))}{n}} + \frac{\Delta}{\sqrt{n}},$$

where

$$\mathbf{Q}_1 := q(\bar{\mathbf{J}},\mathbf{J}|\mathbf{e},\phi,X_{\bar{U}},X_U)$$

$$\mathbf{Q}_2 := \mathbb{E}_{P(U')}q(\bar{\mathbf{J}},\mathbf{J}|\mathbf{e},\phi,X_{\bar{U'}},X_{U'}),$$

$$\mathbf{Q} := \prod_{m=1}^{n}q(J_m|\mathbf{e},\phi,S_m),$$

$$\mathbf{P}_S := \prod_{m=1}^{n}\pi(J_m|\mathbf{e},\phi).$$

## D.2 Necessarily conditions for the prior and the limitation of the existing supersample setting

Here, we further discuss the necessary conditions for the prior distribution to derive a meaningful generalization bound. The proof strategy in Appendix D.1 clarifies this point: in the proof, we decompose the generalization bound in Eq. (13) and separately upper bound the first two terms and the latter two terms.

For the first and second terms, the analysis follows standard generalization error techniques. When using a prior and posterior distribution characterized by the shuffling of the supersample $\tilde{X}$, such as the index variable $U$, the shuffling must swap test and training data to enable generalization evaluation. By ensuring this swap, we can properly assess overfitting.

For the third and fourth terms, after applying the Donsker-Valadhan lemma, it is crucial to ensure that the probability $P''_{k,m}$ does not depend on the sample index $m$ to control the exponential moment in Eq. (19). This requires satisfying $P''_{k,m} = P''_k$, meaning that the probability of assigning the $m$-th data point to the $k$-th codebook must be independent of $m$. By definition, this condition holds when the distribution of the latent variables remains invariant after shuffling.

From these observations, we conclude that the prior used for shuffling must: (A) **Preserve the distribution of the LVs to eliminate interdependencies between LVs and the decoder**, and (B) **Swap test and training data points to evaluate overfitting**, as discussed in Section 3.1.

Using the supersample ensures condition **(B)**. For condition **(A)**, we employ the prior distribution $\pi(J_m|\mathbf{e}, \phi)$, which removes sample index dependency and guarantees $P''_{k,m} = P''_k$. Consequently, the empirical KL divergence in Theorem 2 arises from the third and fourth terms in Eq. (13), as detailed in AppendixD.1.2.

Based on these findings, we propose the following type of prior distribution:

$$\mathbf{P}_S := q(\mathbf{e}, \phi, \theta|S) \prod_{m=1}^{n} \sum_{m'=1}^{n} \frac{1}{N} q(J_m|\mathbf{e}, \phi, S_{m'}),$$

which provides an empirical approximation of the marginal distribution using available samples. Since this distribution does not explicitly depend on the sample index, we can bound the exponential moment similarly to the approach in Appendix D.1.2.

However, using the prior distribution in Eq. (14) to bound the third and fourth terms of Eq. (13) is not feasible. The issue is that applying the Donsker-Valadhan lemma with Eq. (14) to these terms does not yield a bound of order $\mathcal{O}(1/\sqrt{n})$, as achieved in Eq. (20). This limitation arises because the dependency on the sample index in Eq. (14) prevents us from leveraging the symmetry between the test and training datasets via the supersample index $U$. As a result, the prior distribution's symmetry cannot be exploited to simplify the bounds for these terms.

### D.3   Comparison with the fCMI

Here, we analyze the relationship between our CMI and existing forms of fCMI in more detail. As highlighted in the main paper, a key distinction is that our CMI is conditioned on all model parameters, whereas existing fCMI methods marginalize over these parameters.

To further explore this difference, we consider marginalizing over the encoder parameter, $\phi$. In the proof of Theorem 2, we perform this marginalization over $\phi$ in Eq. (12) and obtain

$$\sum_{k=1}^{K} \frac{1}{n} \sum_{m=1}^{n} \mathbb{E}_{q(\bar{J}_m|\mathbf{e},\phi,X_{m,\bar{U}_m})q(\mathbf{e},\phi,\theta|X_U)} l(X_{m,\bar{U}_m}, g_\theta(e_k)) \mathbb{1}_{k=\bar{J}_m}$$

$$- \sum_{k=1}^{K} \frac{1}{n} \sum_{m=1}^{n} \mathbb{E}_{q(J_m|\mathbf{e},\phi,X_{m,U_m})q(\mathbf{e},\phi,\theta|X_U)} l((X_{m,U_m}, g_\theta(e_k)) \mathbb{1}_{k=J_m}$$

$$= \sum_{k=1}^{K} \frac{1}{n} \sum_{m=1}^{n} \mathbb{E}_{q(\bar{J}_m|\theta,\mathbf{e},X_{m,\bar{U}_m})q(\mathbf{e},\theta|X_U)} \|X_{m,\bar{U}_m} - g_\theta(e_k)\|^2 \mathbb{1}_{k=\bar{J}_m}$$

$$- \sum_{k=1}^{K} \frac{1}{n} \sum_{m=1}^{n} \mathbb{E}_{q(J_m|\theta,\mathbf{e},X_{m,U_m})q(\mathbf{e},\theta|X_U)} \|X_{m,U_m} - g_\theta(e_k)\|^2 \mathbb{1}_{k=J_m},$$

and proceed with the proof in the same way. We apply the Donsker-Varadhan inequality between the following distributions, instead of Eq. (14):

$$\mathbf{Q} := P(U)P(U')q(\mathbf{e}, \theta|X_U)q(\bar{\mathbf{J}}, \mathbf{J}|, \mathbf{e}, \theta, X_{\bar{U}}, X_U)$$

$$\mathbf{P} := P(U)q(\mathbf{e}, \theta|X_U)\mathbb{E}_{P(U')}q(\bar{\mathbf{J}}, \mathbf{J}|\mathbf{e}, \theta, X_{\bar{U}'}, X_{U'}).$$

This incorporates marginalization over $\phi$ in Eq. (14), resulting in the following KL divergence in the upper bound:

$$\mathbb{E}_X \text{KL}(\mathbf{Q}|\mathbf{P}) = \mathbb{E}_X \mathbb{E}_{P(U)q(\mathbf{e},\phi|X_U)} \text{KL}(q(\bar{\mathbf{J}}, \mathbf{J}|\mathbf{e}, \theta, X_{\bar{U}}, X_U)|\mathbb{E}_{P(U')}q(\bar{\mathbf{J}}, \mathbf{J}|\mathbf{e}, \theta, X_{\bar{U}'}, X_{U'}))$$
$$= I(\bar{\mathbf{J}}, \mathbf{J}; U|\mathbf{e}, \theta, X).$$

Unlike Theorem 2, this CMI explicitly involves the decoder parameter $\theta$. By marginalizing over $\phi$, decoder information is integrated into the upper bound, making Theorem 2 distinct from existing fCMI bounds. In Appendix D.5, further discussion from the viewpoint of the difference of the graphical model between our CMI and existing fCMI is given.

### D.4 Proof of Lemma 2

We remark that the following relationship holds for $m = 1 \ldots, n$ by definition;

$$I(J_m; S_m|\mathbf{e}, \phi) = \mathbb{E}_{q(\mathbf{e},\phi)}\mathbb{E}_{q(S_m|\mathbf{e},\phi)}\mathbb{E}_{q(J_m|\mathbf{e},\phi,S_m)} \log \frac{q(J_m|\mathbf{e}, \phi, S_m)}{\mathbb{E}_{q(S_m|\mathbf{e},\phi)}q(J_m|\mathbf{e}, \phi, S_m)} \quad (22)$$
$$= \mathbb{E}_S\mathbb{E}_{q(\mathbf{e},\phi|S)}\mathbb{E}_{q(J_m|\mathbf{e},\phi,S_m)} \log \frac{q(J_m|\mathbf{e}, \phi, S_m)}{\mathbb{E}_{q(S_m|\mathbf{e},\phi)}q(J_m|\mathbf{e}, \phi, S_m)}.$$

Next, we show $\mathbb{E}_{q(S_1|\mathbf{e},\phi)}q(J_1|\mathbf{e}, \phi, S_1) = \cdots = \mathbb{E}_{q(S_n|\mathbf{e},\phi)}q(J_n|\mathbf{e}, \phi, S_n)$ holds under the given assumption. To prove this, it is suffice to show that $q(S_1|\mathbf{e}, \phi) = \cdots = q(S_n|\mathbf{e}, \phi)$ holds. Under the given assumption

$$q(\mathbf{e}, \phi|S_1) = \cdots = q(\mathbf{e}, \phi|S_n)$$

holds, see Li et al. [42] for the proof. Then for $i \in [n]$, we have

$$q(\mathbf{e}, \phi|S_i)p(S_i) = q(S_i|\mathbf{e}, \phi)p(\mathbf{e}, \phi)$$

and since all training data points are drawn i.i.d form $\mathcal{D}$, we have

$$q(\mathbf{e}, \phi|S_i)\mathcal{D} = q(S_i|\mathbf{e}, \phi)p(\mathbf{e}, \phi).$$

Then, for any $j \neq i \in [n]$, we also have

$$q(\mathbf{e}, \phi|S_j)\mathcal{D} = q(S_j|\mathbf{e}, \phi)p(\mathbf{e}, \phi)$$

since $q(\mathbf{e}, \phi|S_j) = q(\mathbf{e}, \phi|S_i)$, we conclude that $q(S_i|\mathbf{e}, \phi) = q(S_j|\mathbf{e}, \phi)$. This implies $\mathbb{E}_{q(S_1|\mathbf{e},\phi)}q(J_1|\mathbf{e}, \phi, S_1) = \cdots = \mathbb{E}_{q(S_n|\mathbf{e},\phi)}q(J_n|\mathbf{e}, \phi, S_n)$ holds under the given assumption. So we use the joint distribution these as $\mathbf{P} = \prod_{m=1}^{n} \mathbb{E}_{q(S_m|\mathbf{e},\phi)}q(J_m|\mathbf{e}, \phi, S_m)$. From Eq. (22), we have

$$\mathbb{E}_S\mathbb{E}_{q(\mathbf{e},\phi|S)}\text{KL}(\mathbf{Q}_{\mathbf{J},U}\|\mathbf{P}) = I(J_m; S_m|\mathbf{e}, \phi).$$

Finally, we show that above $\mathbf{P}$ minimizes the $\mathbb{E}_S\mathbb{E}_{q(\mathbf{e},\phi|S)}\text{KL}(\mathbf{Q}_{\mathbf{J},U}\|\mathbf{P})$. We consider the prior $\mathbf{P}'$ that satisfies the assumption of the Theorem 7, that is, prepare some distributions that satisfies $q(J_1|\mathbf{e}, \phi) = \cdots = q(J_n|\mathbf{e}, \phi)$ and define $\mathbf{P}' := \prod_{m=1}^{n} \pi(J_m|\mathbf{e}, \phi)$

By the definition, we have that

$$\mathbb{E}_S\mathbb{E}_{q(\mathbf{e},\phi|S)}\text{KL}(\mathbf{Q}_{\mathbf{J},U}\|\mathbf{P}') = I(J_m; S_m|\mathbf{e}, \phi) + \mathbb{E}_S\mathbb{E}_{q(\mathbf{e},\phi|S)}\text{KL}(\mathbf{P}\|\mathbf{P}').$$

Thus, when using $\mathbf{P}' = \mathbf{P}$ minimizes the empirical KL divergence.

## D.5 Proof of Eq. (8)

Here we discuss how we can upper bound of the complexity term of the obtained bound. From the definition, we have the following relation;

$$\mathbb{E}_{X,U}\mathbb{E}_{q(\mathbf{e},\phi,\theta|X_U)}\mathrm{KL}(\mathbf{Q}_{\bar{\mathbf{J}},U}\|\mathbf{Q}_{\bar{\mathbf{J}}})$$

$$= \mathbb{E}_{P(X)P(U)q(\mathbf{e},\phi,\theta|X_U)q(\bar{\mathbf{J}},\mathbf{J}|\mathbf{e},\phi,X_{\bar{U}},X_U)} \log \frac{q(\bar{\mathbf{J}},\mathbf{J}|\mathbf{e},\phi,X_{\bar{U}},X_U)}{\displaystyle\mathop{\mathbb{E}}_{P(U')} q(\bar{\mathbf{J}},\mathbf{J}|\mathbf{e},\phi,X_{\bar{U'}},X_{U'})}$$

$$= \mathbb{E}_{P(X)P(U)q(\mathbf{e},\phi|X_U)q(\bar{\mathbf{J}},\mathbf{J}|\mathbf{e},\phi,X_{\bar{U}},X_U)} \log \frac{q(\bar{\mathbf{J}},\mathbf{J}|\mathbf{e},\phi,X_{\bar{U}},X_U)}{\displaystyle\mathop{\mathbb{E}}_{P(U')} q(\bar{\mathbf{J}},\mathbf{J}|\mathbf{e},\phi,X_{\bar{U'}},X_{U'})}$$

$$= \mathbb{E}_{P(X)P(\mathbf{e},\phi|X)}\mathbb{E}_{P(U|\mathbf{e},\phi,X)q(\bar{\mathbf{J}},\mathbf{J}|\mathbf{e},\phi,X,U)} \log \frac{q(\bar{\mathbf{J}},\mathbf{J}|\mathbf{e},\phi,X_{\bar{U}},X_U)}{\displaystyle\mathop{\mathbb{E}}_{P(U')} q(\bar{\mathbf{J}},\mathbf{J}|\mathbf{e},\phi,X_{\bar{U'}},X_{U'})}$$

$$= \mathbb{E}_{P(X)P(\mathbf{e},\phi|X)}\mathbb{E}_{P(U|\mathbf{e},\phi,X)q(\bar{\mathbf{J}},\mathbf{J}|\mathbf{e},\phi,X,U)} \log \frac{q(\bar{\mathbf{J}},\mathbf{J}|\mathbf{e},\phi,X_{\bar{U}},X_U)}{\displaystyle\mathop{\mathbb{E}}_{P(U'|\mathbf{e},\phi,X)} q(\bar{\mathbf{J}},\mathbf{J}|\mathbf{e},\phi,X_{\bar{U'}},X_{U'})}$$

$$+ \mathbb{E}_{P(X)P(\mathbf{e},\phi,|X)}\mathbb{E}_{P(U|\mathbf{e},\phi,X)q(\bar{\mathbf{J}},\mathbf{J}|\mathbf{e},\phi,X,U)} \log \frac{\displaystyle\mathop{\mathbb{E}}_{P(U'|\mathbf{e},\phi,,X)} q(\bar{\mathbf{J}},\mathbf{J}|\mathbf{e},\phi,X_{\bar{U'}},X_{U'})}{\displaystyle\mathop{\mathbb{E}}_{P(U')} q(\bar{\mathbf{J}},\mathbf{J}|\mathbf{e},\phi,X_{\bar{U'}},X_{U'})}$$

$$= I(\bar{\mathbf{J}},\mathbf{J};U|\mathbf{e},\phi,X) + \mathbb{E}_{P(X)P(\mathbf{e},\phi|X)}\mathbb{E}_{P(U|\mathbf{e},\phi,X)q(\bar{\mathbf{J}},\mathbf{J}|\mathbf{e},\phi,X,U)} \log \frac{\displaystyle\mathop{\mathbb{E}}_{P(U'|\mathbf{e},\phi,X)} q(\bar{\mathbf{J}},\mathbf{J}|\mathbf{e},\phi,X_{\bar{U'}},X_{U'})}{\displaystyle\mathop{\mathbb{E}}_{P(U')} q(\bar{\mathbf{J}},\mathbf{J}|\mathbf{e},\phi,X_{\bar{U'}},X_{U'})}$$

$$\leq I(\bar{\mathbf{J}},\mathbf{J};U|\mathbf{e},\phi,X) + \mathbb{E}_{P(X)P(\mathbf{e},\phi|X)}\mathbb{E}_{P(U|\mathbf{e},\phi,X)} \log \frac{P(U'|\mathbf{e},\phi,X)}{P(U')}$$

$$= I(\bar{\mathbf{J}},\mathbf{J};U|\mathbf{e},\phi,X) + \mathbb{E}_{P(X)P(\mathbf{e},\phi|X)}\mathbb{E}_{P(U|\mathbf{e},\phi,X)} \log \frac{P(\mathbf{e},\phi|X,U')P(U'|X)}{\mathbb{E}_{P(U''|X)}P(\mathbf{e},\phi|X,U'')P(U')}$$

$$= I(\bar{\mathbf{J}},\mathbf{J};U|\mathbf{e},\phi,X) + \mathbb{E}_{P(X)P(\mathbf{e},\phi|X)}\mathbb{E}_{P(U|\mathbf{e},\phi,X)} \log \frac{P(\mathbf{e},\phi|X,U')P(U')}{\mathbb{E}_{P(U'')}P(\mathbf{e},\phi|X,U'')P(U')}$$

$$= I(\bar{\mathbf{J}},\mathbf{J};U|\mathbf{e},\phi,X) + I(\mathbf{e},\phi;U|X),$$

where we used the data processing inequality of the KL divergence.

## D.6 The role of $I(\tilde{\mathbf{J}};U|\mathbf{e},\phi,\tilde{X})$

The role of $I(\tilde{\mathbf{J}};U|\mathbf{e},\phi,\tilde{X})$ is clarified through the following upper bound:

$$I(\tilde{\mathbf{J}};U|\mathbf{e},\phi,\tilde{X}) \leq \sum_{m=1}^{n} I(e_J;\tilde{X}_{m,\bar{U}_m}|\mathbf{e},\phi)$$
$$+ \mathbb{E}_S\mathbb{E}_{q(\mathbf{e},\phi|S)}\mathrm{KL}(\mathbf{Q}_{\mathbf{J},U}\|\mathbf{P}). \tag{23}$$

The first term represents the information retained by the LVs from the training data in the IB hypothesis, while the second term corresponds to the regularization based on the empirical KL divergence discussed earlier.

Here we prove Eq. (23). We define $\pi(\bar{\mathbf{J}}|\mathbf{e},\phi) = \prod_{m=1}^{n} \pi(\bar{J}_m|\mathbf{e},\phi)$, $\pi(\mathbf{J}|\mathbf{e},\phi) = \prod_{m=1}^{n} \pi(J_m|\mathbf{e},\phi)$, and $\pi(\tilde{\mathbf{J}}|\mathbf{e},\phi) = \pi(\bar{\mathbf{J}},\mathbf{J}|\mathbf{e},\phi) = \pi(\bar{\mathbf{J}}|\mathbf{e},\phi)\pi(\mathbf{J}|\mathbf{e},\phi)$ where each $\pi(\bar{J}_m|\mathbf{e},\phi)$ is the marginal distribution of $\pi(J_m|\mathbf{e},\phi,X_m)$.

Then by the definition of the CMI, we have

$I(\tilde{\mathbf{J}}; U | \mathbf{e}, \phi, \tilde{X})$

$= \mathbb{E}_{\tilde{X}, U} \mathbb{E}_{q(\mathbf{e}, \phi | \tilde{X}_U)} \mathrm{KL}(q(\tilde{\mathbf{J}} | \mathbf{e}, \phi, \tilde{X}) \| \mathbb{E}_{U'} q(\bar{\mathbf{J}}, \mathbf{J} | \mathbf{e}, \phi, \tilde{X}_{\bar{U}'}, \tilde{X}_{U'}))$

$\leq \mathbb{E}_{\tilde{X}, U} \mathbb{E}_{q(\mathbf{e}, \phi | \tilde{X}_U)} \mathrm{KL}(q(\tilde{\mathbf{J}} | \mathbf{e}, \phi, \tilde{X}) \| \pi(\bar{\mathbf{J}}, \mathbf{J} | \mathbf{e}, \phi))$

$= \mathbb{E}_{\tilde{X}, U} \mathbb{E}_{q(\mathbf{e}, \phi | \tilde{X}_U)} \mathrm{KL}(q(\bar{\mathbf{J}} | \mathbf{e}, \phi, \tilde{X}_{\bar{U}}) \| \pi(\bar{\mathbf{J}} | \mathbf{e}, \phi)) + \mathbb{E}_{\tilde{X}, U} \mathbb{E}_{q(\mathbf{e}, \phi | \tilde{X}_U)} \mathrm{KL}(q(\mathbf{J} | \mathbf{e}, \phi, \tilde{X}_U) \| \pi(\mathbf{J} | \mathbf{e}, \phi))$

$= \mathbb{E}_{\tilde{X}, U} \mathbb{E}_{q(\mathbf{e}, \phi | \tilde{X}_U)} \sum_{m=1}^{n} \mathrm{KL}(q(\bar{J}_m | \mathbf{e}, \phi, \tilde{X}_{m, \bar{U}_m}) \| \pi(\bar{J}_m | \mathbf{e}, \phi))$

$\qquad + \mathbb{E}_{\tilde{X}, U} \mathbb{E}_{q(\mathbf{e}, \phi | \tilde{X}_U)} \sum_{m=1}^{n} \mathrm{KL}(q(J_m | \mathbf{e}, \phi, \tilde{X}_{m, U_m}) \| \pi(J_m | \mathbf{e}, \phi))$

$= n I(J; X | \mathbf{e}, \phi) + \mathbb{E}_S \mathbb{E}_{q(\mathbf{e}, \phi | S)} \frac{1}{n} \sum_{m=1}^{n} \mathrm{KL}(q(J_m | \mathbf{e}, \phi, S_m) \| \pi(J_m | \mathbf{e}, \phi))$

$\leq n I(e_J; X | \mathbf{e}, \phi) + \mathbb{E}_S \mathbb{E}_{q(\mathbf{e}, \phi | S)} \frac{1}{n} \sum_{m=1}^{n} \mathrm{KL}(q(J_m | \mathbf{e}, \phi, S_m) \| \pi(J_m | \mathbf{e}, \phi)).$

### D.7 Proof of Lemma 1 and 3 and additional discussion

*Proof of Lemma 1.* From the definition of the CMI, we have

$$I(\bar{\mathbf{J}}, \mathbf{J}; U | \mathbf{e}, \phi, X) = H[\tilde{\mathbf{J}} | \mathbf{e}, \phi, X] - H[\tilde{\mathbf{J}} | U, \mathbf{e}, \phi, X] \leq H[\tilde{\mathbf{J}} | \mathbf{e}, \phi, X] \leq H[\tilde{\mathbf{J}} | X].$$

Here, we consider the case where $f_\phi : \mathcal{X} \to [K]$ represents a deterministic encoder that maps input data to one of the $K$ indices. This scenario can be viewed as a $K$-class classification problem, allowing us to directly apply the results from Harutyunyan et al. [30]. They demonstrated that the CMI for multi-class classification problems can be upper-bounded using the Natarajan dimension, a combinatorial measure that generalizes the VC dimension to the multiclass setting.

Using this concept, we obtain the following characterization:

When employing a deterministic encoder network $f'_\phi : \mathcal{X} \to [K]$ that belongs to a class with finite Natarajan dimension $d_K$ and assuming $2n > d_K + 1$, we derive the following bound:

$$I(\tilde{\mathbf{J}}; U | \mathbf{e}, \phi, \tilde{X}) \leq d_K \log \left( \binom{K}{2} \frac{2en}{d_K} \right). \tag{24}$$

The proof follows exactly as in Theorem 8 of Harutyunyan et al. [30]. $\qquad \square$

Thus, by regularizing the capacity of the encoder model (via the Natarajan dimension), the CMI term scales as $\mathcal{O}(\log n)$, ensuring controlled generalization behavior. Examples of models that satisfy the finite Natarajan dimension are shown in Jin [36] and Daniely et al. [16]. Also, see Bendavid et al. [8], which shows that the VC dimension of the multiclass loss function characterizes the graph dimension, and the graph dimension upper bounds the Natarajan dimension.

*Proof of Lemma 3.* Since we consider the setting of Lemma 2, we consider the case of $\mathrm{KL}(\mathbf{Q}_{\mathbf{J}, U} \| \mathbf{P}) = \sum_m I(J_m; S_m | \mathbf{e}, \phi)$. Following the above setting of $I(\tilde{\mathbf{J}}; U | \mathbf{e}, \phi, \tilde{X})$, that is, $f'_{\mathbf{e}, \phi} : \mathcal{X} \to [K]$ satisfies the Natarajan dimension $d_K > 1$. Then for each $m$, we have

$$I(J_m; S_m | \mathbf{e}, \phi) = H[J_m | \mathbf{e}, \phi] - H[J_m | S_m \mathbf{e}, \phi] = H[J_m | \mathbf{e}, \phi] \leq \log K \leq (d_K + 1) \log K.$$

Thus $\mathrm{KL}(\mathbf{Q}_{\mathbf{J}, U} \| \mathbf{P}) / n = \sum_m I(J_m; S_m | \mathbf{e}, \phi) / n \leq \log K = \mathcal{O}(1)$. $\qquad \square$

The difference between $I(\tilde{\mathbf{J}}; U | \mathbf{e}, \phi, \tilde{X})$ and $I(J_m; S_m | \mathbf{e}, \phi)$ lies in their conditioning. Since $I(\tilde{\mathbf{J}}; U | \mathbf{e}, \phi, \tilde{X})$ is conditioned on all $2n$ data points, it only depends on the combinatorial number of distinct index values. In contrast, $I(J_m; S_m | \mathbf{e}, \phi)$ does not condition on the input data, making regularization based solely on the Natarajan dimension insufficient to control complexity.

For the discussion of the stochastic encoder, see Appendix E.4, where we consider the metric entropy of $f_\phi(\cdot)$, which leads to a similar discussion.

### D.7.1 Additional discussion for the Natarajan dimension

Here, we briefly discuss the Natarajan dimension. First, it can be both upper and lower bounded by the graph dimension, another common combinatorial measure for multi-class classification problems (see Lemma 4 and Proposition 1 in Guermeur [27]).

The Natarajan dimension can also be upper bounded by the $\gamma$ fat-shattering dimension of each class. Specifically, given $f'\mathbf{e}, \phi : \mathcal{X} \to [K]$, let the $k$-th element of its output be denoted as $f\mathbf{e}, \phi^{'(k)}$ for $k = 1, \ldots, K$. If each $f_{\mathbf{e},\phi}^{'(k)}$ has a finite $\gamma$-shattering dimension, then the Natarajan dimension of $f'_{\mathbf{e},\phi}$ can be bounded by the sum of the $\gamma$-shattering dimensions of its components, multiplied by a constant coefficient (see Lemma 10 in Guermeur [27]).

Examples of fat-shattering dimension evaluations can be found in Bartlett & Maass [7], which analyzes neural network models, and Gottlieb et al. [24], which examines the fat-shattering dimension of Lipschitz function classes. If our encoder network satisfies these properties, its covering number can be appropriately bounded.

### D.8 Discussion about the overfitting term

Here, we discuss how the overfitting terms relate to different algorithms. First, from the data processing inequality [15], we obtain

$$I(\mathbf{e}, \phi; U|\tilde{X}) \leq I(\mathbf{e}, \phi; S),$$

where we express $\tilde{X}_U$ as the training dataset $S$. Since this expression does not include conditioning, we refer to it as the parameter MI. Several existing studies have analyzed parameter MI under commonly used algorithms.

Pensia et al. [51] first established the relationship between noisy iterative algorithms and parameter MI. Subsequently, Wang et al. [76] and Wang et al. [77] investigated the parameter MI of the SGLD algorithm from the perspective of noisy iterative algorithms, while Futami & Fujisawa [20] analyzed it in the continuous-time limit. Neu et al. [50] was the first to examine parameter MI in SGD, with Wang & Mao [78] later improving its dependency on the step size. Furthermore, Haghifam et al. [29] provided formal limitations in the context of stochastic convex optimization.

In addition to these, in the Bayesian setting, where we assume that the training dataset is conditionally i.i.d (see Clarke & Barron [14] for the formal settings), Clarke & Barron [14] (see also Rissanen [53], Haussler & Opper [31]) clarified that the mutual information between learned parameter and training dataset is described as follows: if $w$ takes a value in a $d$-dimensional compact subset of $\mathbb{R}^d$ and $p(y|x; w)$ is smooth in $w$, then as $n \to \infty$, we have

$$I(W; S) = \frac{d}{2} \log \frac{n}{2\pi e} + h(W) + \mathbb{E} \log \det J + o(1),$$

where $h(W)$ is the differential entropy of $W$, and $J$ is the Fisher information matrix of $p(Y|X; W)$.

Steinke & Zakynthinou [63] clarified that the CMI is upper bounded by the the stability. For example, if the training algorithm satisfies $\sqrt{2\epsilon}$-differentially private (DP) algorithm, then CMI is upper-bounded by $\epsilon n$. So this $\epsilon$ is controlled by the DP algorithm. The Gibbs algorithm equipped with $[0, 1]$ bounded loss function, satisfies $\mathcal{O}(1/n)$-DP, thus its CMI is controlled adequately. Steinke & Zakynthinou [63] also clarified that if the algorithm is $\delta$ stable in total variation distance, then CMI is upper bounded by $\delta n$. Li et al. [42] studied the total variation stability for the SGD, and Mou et al. [48] studied such stability of the SGLD algorithm and its relation to the PAC-Bayesian bound. [49] investigated the CMI of SGLD as the noisy iterative algorithm.

## E Proofs for Section 4

### E.1 Proof of Theorem 3

We define $\mathbf{T} = \{\mathbf{T}_0, \mathbf{T}_1\}$, where $\tilde{X}_{\mathbf{T}_0} = (\tilde{X}_{T_1}, \ldots, \tilde{X}_{T_n})$ serves as the test dataset and $\tilde{X}_{\mathbf{T}_1} = (\tilde{X}_{T_{n+1}}, \ldots, \tilde{X}_{T_{2n}})$ serves as the training dataset. We further express $\tilde{X}_{\mathbf{T}_0} = (\tilde{X}_{T_1}, \ldots, \tilde{X}_{T_n}) = (\tilde{X}_{\mathbf{T}_{0,1}}, \ldots, \tilde{X}_{\mathbf{T}_{0,n}})$ and $\tilde{X}_{\mathbf{T}_1} = (\tilde{X}_{\mathbf{T}_{1,1}}, \ldots, \tilde{X}_{\mathbf{T}_{1,n}})$. To emphasize the dependence of the

dataset on $\mathbf{T}$, we write the posterior distribution as $q(\tilde{\mathbf{J}}|\mathbf{e}, \phi, \tilde{X}_{\mathbf{T}}) = q(\bar{\mathbf{J}}, \mathbf{J}|\mathbf{e}, \phi, \tilde{X}_{\mathbf{T}}) = q(\bar{\mathbf{J}}, \mathbf{J}|\mathbf{e}, \phi, \tilde{X}_{\mathbf{T}_0}, \tilde{X}_{\mathbf{T}_1}) = q(\bar{\mathbf{J}}|\mathbf{e}, \phi, \tilde{X}_{\mathbf{T}_0}) q(\mathbf{J}|\mathbf{e}, \phi, \tilde{X}_{\mathbf{T}_1})$.

Hereinafter, we express $\tilde{X}$ as $X$ to simplify the notation. Under the permutation symmetric settings, the generalization error can be expressed as

$$
\mathbb{E}_{S,X} \mathbb{E}_{q(\mathbf{e},\phi,\theta|S)} \left( \mathbb{E}_{q(J|\mathbf{e},\phi,X)} l(X, g_\theta(e_J)) - \frac{1}{n} \sum_{m=1}^{n} \mathbb{E}_{q(J_m|\mathbf{e},\phi,S_m)} l(S_m, g_\theta(e_{J_m})) \right)
$$

$$
= \mathbb{E}_{X,\mathbf{T}} \sum_{k=1}^{K} \frac{1}{n} \sum_{m=1}^{n} \mathbb{E}_{q(\bar{J}_m|\mathbf{e},\phi,X_{\mathbf{T}_{0,m}}) q(\mathbf{e},\phi,\theta|X_{\mathbf{T}_1})} l((X_{\mathbf{T}_{0,m}}, g_\theta(e_k)) \mathbb{1}_{k=\bar{J}_m}
$$

$$
- \mathbb{E}_{X,\mathbf{T}} \sum_{k=1}^{K} \frac{1}{n} \sum_{m=1}^{n} \mathbb{E}_{q(J_m|\mathbf{e},\phi,X_{\mathbf{T}_{1,m}}) q(\mathbf{e},\phi,\theta|X_{\mathbf{T}_1})} l(X_{\mathbf{T}_{1,m}}, g_\theta(e_k)) \mathbb{1}_{k=J_m}
$$

$$
= \mathbb{E}_{X,\mathbf{T}} \sum_{k=1}^{K} \frac{1}{n} \sum_{m=1}^{n} \mathbb{E}_{q(\bar{J}_m|\mathbf{e},\phi,X_{\mathbf{T}_{0,m}}) q(\mathbf{e},\phi,\theta|X_{\mathbf{T}_1})} \|X_{\mathbf{T}_{0,m}} - g_\theta(e_k)\|^2 \mathbb{1}_{k=\bar{J}_m}
$$

$$
- \mathbb{E}_{X,\mathbf{T}} \sum_{k=1}^{K} \frac{1}{n} \sum_{m=1}^{n} \mathbb{E}_{q(J_m|\mathbf{e},\phi,X_{\mathbf{T}_{1,m}}) q(\mathbf{e},\phi,\theta|X_{\mathbf{T}_1})} \|X_{\mathbf{T}_{1,m}} - g_\theta(e_k)\|^2 \mathbb{1}_{k=J_m}.
$$

We then decompose the loss as follows

$$
\mathrm{gen}(n, \mathcal{D}) \tag{25}
$$

$$
= \mathbb{E}_{X,\mathbf{T}} \sum_{k=1}^{K} \frac{1}{n} \sum_{m=1}^{n} \mathbb{E}_{q(\bar{J}_m|\mathbf{e},\phi,X_{\mathbf{T}_{0,m}}) q(\mathbf{e},\phi,\theta|X_{\mathbf{T}_1})} \|X_{\mathbf{T}_{0,m}} - g_\theta(e_k)\|^2 \mathbb{1}_{k=\bar{J}_m}
$$

$$
- \mathbb{E}_{X,\mathbf{T}} \sum_{k=1}^{K} \frac{1}{n} \sum_{m=1}^{n} \mathbb{E}_{q(J_m|\mathbf{e},\phi,X_{\mathbf{T}_{1,m}}) q(\mathbf{e},\phi,\theta|X_{\mathbf{T}_1})} \|X_{\mathbf{T}_{0,m}} - g_\theta(e_k)\|^2 \mathbb{1}_{k=J_m}
$$

$$
+ \mathbb{E}_{X,\mathbf{T}} \sum_{k=1}^{K} \frac{1}{n} \sum_{m=1}^{n} \mathbb{E}_{q(J_m|\mathbf{e},\phi,X_{\mathbf{T}_{1,m}}) q(\mathbf{e},\phi,\theta|X_{\mathbf{T}_1})} \|X_{\mathbf{T}_{0,m}} - g_\theta(e_k)\|^2 \mathbb{1}_{k=J_m}
$$

$$
- \mathbb{E}_{X,\mathbf{T}} \sum_{k=1}^{K} \frac{1}{n} \sum_{m=1}^{n} \mathbb{E}_{q(J_m|\mathbf{e},\phi,X_{\mathbf{T}_{1,m}}) q(\mathbf{e},\phi,\theta|X_{\mathbf{T}_1})} \|X_{\mathbf{T}_{1,m}} - g_\theta(e_k)\|^2 \mathbb{1}_{k=J_m}.
$$

First, we upper bound the first two terms by applying the Donsker-Varadhan inequality. Consider the joint distribution and the prior distribution, defined as follows:

$$
\mathbf{Q} := P(\mathbf{T}) q(\mathbf{e}, \theta, \phi|X_{\mathbf{T}_1}) q(\bar{\mathbf{J}}, \mathbf{J}|\mathbf{e}, \phi, X_{\mathbf{T}}), \tag{26}
$$

$$
\mathbf{P} := P(\mathbf{T}) q(\mathbf{e}, \theta, \phi|X_{\mathbf{T}_1}) \mathbb{E}_{P(\mathbf{T}')} q(\bar{\mathbf{J}}, \mathbf{J}|\mathbf{e}, \phi, X_{\mathbf{T}'}).
$$

This corresponds to the posterior and data-dependent prior distributions defined in Section 4.1.

Then we then obtain

$$
\mathbb{E}_{X,\mathbf{T}} \sum_{k=1}^{K} \frac{1}{n} \sum_{m=1}^{n} \mathbb{E}_{q(\mathbf{e},\phi,\theta|X_{\mathbf{T}_1})} \|X_{\mathbf{T}_{0,m}} - g_\theta(e_k)\|^2 \left( \mathbb{E}_{q(\bar{J}_m|\mathbf{e},\phi,X_{\mathbf{T}_{1,m}})} \mathbb{1}_{k=\bar{J}_m} - \mathbb{E}_{q(J_m|\mathbf{e},\phi,X_{\mathbf{T}_{0,m}})} \mathbb{1}_{k=J_m} \right)
$$

$$
\leq \mathbb{E}_X \frac{1}{\lambda} \mathrm{KL}(\mathbf{Q}|\mathbf{P}) + \mathbb{E}_X \frac{1}{\lambda} \log \mathbb{E}_{\mathbf{P}} \exp \left( \frac{\lambda}{n} \sum_{k=1}^{K} \sum_{m=1}^{n} \|X_{\mathbf{T}_{0,m}} - g_\theta(e_k)\|^2 \left( \mathbb{1}_{k=\bar{J}_m} - \mathbb{1}_{k=J_m} \right) \right). \tag{27}
$$

Note that $\underset{P(\mathbf{T}')}{\mathbb{E}}\, q(\bar{\mathbf{J}}, \mathbf{J}|\mathbf{e}, \phi, X_{\mathbf{T}'})$ is symmetric with respect to the permutation of $\mathbf{T}$. Thus, we have

$$\log \mathbb{E}_{P(\mathbf{T})q(\mathbf{e},\theta,\phi|X_{\mathbf{T}_1})\underset{P(\mathbf{T}')}{\mathbb{E}}\, q(\bar{\mathbf{J}},\mathbf{J}|\mathbf{e},\phi,X_{\mathbf{T}'})} \exp\left(\frac{\lambda}{n}\sum_{k=1}^{K}\sum_{m=1}^{n} l(X_{\mathbf{T}_{0,m}}, g_\theta(e_k))\left(\mathbb{1}_{k=\bar{J}_m} - \mathbb{1}_{k=J_m}\right)\right)$$

$$= \log \mathbb{E}_{P(\mathbf{T})q(\mathbf{e},\theta,\phi|X_{\mathbf{T}_1})\underset{P(\mathbf{T}')}{\mathbb{E}}\, q(\bar{\mathbf{J}},\mathbf{J}|\mathbf{e},\phi,X_{\mathbf{T}'})P(\mathbf{T}'')}$$

$$\exp\left(\frac{\lambda}{n}\sum_{k=1}^{K}\sum_{m=1}^{n} l(X_{\mathbf{T}_{0,m}}, g_\theta(e_k))\left(\mathbb{1}_{k=J_{\mathbf{T}''_{0,m}}} - \mathbb{1}_{k=J_{\mathbf{T}''_{1,m}}}\right)\right)$$

$$= \log \mathbb{E}_{P(\mathbf{T})q(\mathbf{e},\theta,\phi|X_{\mathbf{T}_1})\underset{P(\mathbf{T}')}{\mathbb{E}}\, q(\bar{\mathbf{J}},\mathbf{J}|\mathbf{e},\phi,X_{\mathbf{T}'})}$$

$$\mathbb{E}_{P(\mathbf{T}'')} \exp\left(\frac{\lambda}{n}\sum_{k=1}^{K}\sum_{m=1}^{n} l(X_{\mathbf{T}_{0,m}}, g_\theta(e_k))\left(\mathbb{1}_{k=J_{\mathbf{T}''_{0,m}}} - \mathbb{1}_{k=J_{\mathbf{T}''_{1,m}}}\right)\right).$$

To simplify the notation, we define $\mathbf{T}'' = \{\mathbf{T}''_0, \mathbf{T}''_1\} = \{\mathbf{T}''_{0,1}, \ldots, \mathbf{T}''_{0,n}, \mathbf{T}''_{1,1}, \ldots, \mathbf{T}''_{1,n}\}$. Note that $\mathbf{T}''_{j,m}$ for $m = 1, \ldots, n$ and $j = 0, 1$ are not independent of each other due to the permutation that generates them. Therefore, we cannot directly apply standard concentration inequalities, as is possible in the existing supersample setting.

To address this, we use the results from Joag-Dev & Proschan [37], which concern the negative association of permutation variables. From Theorem 2.11 in Joag-Dev & Proschan [37], the distribution $P(\mathbf{T})$ satisfies negative association. Additionally, as discussed in Section 3.3 of Joag-Dev & Proschan [37] and further in Proposition 4 and 5 of Dubhashi & Ranjan [18], we have that

$$\log \mathbb{E}_{P(\mathbf{T})q(\mathbf{e},\theta,\phi|X_{\mathbf{T}_1})\underset{P(\mathbf{T}')}{\mathbb{E}}\, q(\bar{\mathbf{J}},\mathbf{J}|\mathbf{e},\phi,X_{\mathbf{T}'})}$$

$$\mathbb{E}_{P(\mathbf{T}'')} \exp\left(\frac{\lambda}{n}\sum_{k=1}^{K}\sum_{m=1}^{n} l(X_{\mathbf{T}_{0,m}}, g_\theta(e_k))\left(\mathbb{1}_{k=J_{\mathbf{T}''_{0,m}}} - \mathbb{1}_{k=J_{\mathbf{T}''_{1,m}}}\right)\right)$$

$$\leq \log \mathbb{E}_{P(\mathbf{T})q(\mathbf{e},\theta,\phi|X_{\mathbf{T}_1})\underset{P(\mathbf{T}')}{\mathbb{E}}\, q(\bar{\mathbf{J}},\mathbf{J}|\mathbf{e},\phi,X_{\mathbf{T}'})}$$

$$\mathbb{E}_{\prod_{m=1}^{n}\prod_{j=0,1}P(\mathbf{T}''_{j,m})} \exp\left(\frac{\lambda}{n}\sum_{k=1}^{K}\sum_{m=1}^{n} l(X_{\mathbf{T}_{0,m}}, g_\theta(e_k))\left(\mathbb{1}_{k=J_{\mathbf{T}''_{0,m}}} - \mathbb{1}_{k=J_{\mathbf{T}''_{1,m}}}\right)\right),$$

where $P(\mathbf{T}''_{j,m})$ is the marginal distribution, implying that $\mathbf{T}''_{j,m}$ are now $2n$ independent random variables. Intuitively, the results in Joag-Dev & Proschan [37] indicate that the elements of the permutation index, which follow the permutation distribution, are negatively correlated. As a result, the expectation of the marginal distribution is larger than that of the joint distribution.

Since $\{\mathbf{T}''_{j,m}\}$ are independent, we can apply McDiarmid's inequality, which leads to the results in

$$\log \mathbb{E}_{P(\mathbf{T})q(\mathbf{e},\theta,\phi|X_{\mathbf{T}_1})\underset{P(\mathbf{T}')}{\mathbb{E}}\, q(\bar{\mathbf{J}},\mathbf{J}|\mathbf{e},\phi,X_{\mathbf{T}'})}$$

$$\exp\left(\frac{\lambda}{n}\sum_{k=1}^{K}\sum_{m=1}^{n} l(X_{\mathbf{T}_{0,m}}, g_\theta(e_k))\left(\mathbb{1}_{k=\bar{J}_m} - \mathbb{1}_{k=J_m}\right)\right)$$

$$\leq \log \mathbb{E}_{P(\mathbf{T})q(\mathbf{e},\theta,\phi|X_{\mathbf{T}_1})\underset{P(\mathbf{T}')}{\mathbb{E}}\, q(\bar{\mathbf{J}},\mathbf{J}|\mathbf{e},\phi,X_{\mathbf{T}'})}$$

$$\mathbb{E}_{\prod_{m=1}^{n}\prod_{j=0,1}P(\mathbf{T}''_{j,m})} \exp\left(\frac{\lambda}{n}\sum_{k=1}^{K}\sum_{m=1}^{n} l(X_{\mathbf{T}_{0,m}}, g_\theta(e_k))\left(\mathbb{1}_{k=J_{\mathbf{T}''_{0,m}}} - \mathbb{1}_{k=J_{\mathbf{T}''_{1,m}}}\right)\right)$$

$$\leq \frac{\lambda^2 \Delta^2}{n}. \tag{28}$$

This is derived similarly to Eq. (15). Note that there are $2n$ variables so the calculation of the upper bound is $(\Delta\lambda/n)^2/8 \times 2n = \lambda^2\Delta^2/4n$.

Next, we focus on the third and fourth terms in Eq. (25). Similarly to Eq. (18), we have

$$\mathbb{E}_{X,\mathbf{T}} \sum_{k=1}^{K} \frac{1}{n} \sum_{m=1}^{n} \mathbb{E}_{q(J_m|\mathbf{e},\phi,X_{\mathbf{T}_{1,m}})q(\mathbf{e},\phi,\theta|X_{\mathbf{T}_1})} \|X_{\mathbf{T}_{0,m}} - g_\theta(e_k)\|^2 \mathbb{1}_{k=J_m}$$

$$- \mathbb{E}_{X,\mathbf{T}} \sum_{k=1}^{K} \frac{1}{n} \sum_{m=1}^{n} \mathbb{E}_{q(J_m|\mathbf{e},\phi,X_{\mathbf{T}_{1,m}})q(\mathbf{e},\phi,\theta|X_{\mathbf{T}_1})} \|X_{\mathbf{T}_{1,m}} - g_\theta(e_k)\|^2 \mathbb{1}_{k=J_m}$$

$$= \mathbb{E}_{X,\mathbf{T}} \frac{2}{n} \sum_{m=1}^{n} \left(X_{\mathbf{T}_{1,m}} - X_{\mathbf{T}_{0,m}}\right) \cdot \mathbb{E}_{q(J_m|\mathbf{e},\phi,X_{\mathbf{T}_{1,m}})q(\mathbf{e},\phi,\theta|X_{\mathbf{T}_1})} \sum_{k=1}^{K} g_\theta(e_k) \mathbb{1}_{k=J_m}$$

$$\leq \mathbb{E}_X \frac{1}{\lambda} \mathrm{KL}(\mathbf{Q}|\mathbf{P}) + \mathbb{E}_X \frac{1}{\lambda} \log \mathbb{E}_\mathbf{P} \exp\left(\frac{2\lambda}{n} \sum_{m=1}^{n} \left(X_{\mathbf{T}_{1,m}} - X_{\mathbf{T}_{0,m}}\right) \cdot \sum_{k=1}^{K} g_\theta(e_k) \mathbb{1}_{k=J_m}\right)$$

$$\leq \mathbb{E}_X \frac{1}{\lambda} \mathrm{KL}(\mathbf{Q}|\mathbf{P})$$

$$+ \mathbb{E}_X \frac{1}{\lambda} \log \mathbb{E}_{P(\mathbf{T})q(\mathbf{e},\theta,\phi|X_{\mathbf{T}_1})} \mathbb{E}_{\substack{\mathbb{E} \\ P(\mathbf{T}')}} q(\bar{\mathbf{J}},\mathbf{J}|\mathbf{e},\phi,X_{\mathbf{T}'}) \mathbb{E}_{\prod_{m=1}^{n} \prod_{j=0,1} P(\mathbf{T}''_{j,m})}$$

$$\exp\left(\frac{2\lambda}{n} \sum_{m=1}^{n} \left(X_{\mathbf{T}_{1,m}} - X_{\mathbf{T}_{0,m}}\right) \cdot \sum_{k=1}^{K} g_\theta(e_k) \mathbb{1}_{k=J_m}\right). \tag{29}$$

We first evaluate the expectation of the exponential moment;

$$\Omega := \mathbb{E}_{P(\mathbf{T})q(\mathbf{e},\theta,\phi|X_{\mathbf{T}_1})} \frac{2}{n} \sum_{m=1}^{n} \left(X_{\mathbf{T}_{1,m}} - X_{\mathbf{T}_{0,m}}\right) \cdot \mathbb{E}_{\substack{\mathbb{E} \\ P(\mathbf{T}')}} q(\bar{\mathbf{J}},\mathbf{J}|\mathbf{e},\phi,X_{\mathbf{T}'}) \sum_{k=1}^{K} g_\theta(e_k) \mathbb{1}_{k=J_m}. \tag{30}$$

Let us now focus on the expectation $\mathbb{E}_{\substack{\mathbb{E} \\ P(\mathbf{T}')}} q(\bar{\mathbf{J}},\mathbf{J}|\mathbf{e},\phi,X_{\mathbf{T}'})$. Due to the permutation symmetry, $\mathbb{E}_{\substack{\mathbb{E} \\ P(\mathbf{T}')}} q(\bar{\mathbf{J}},\mathbf{J}|\mathbf{e},\phi,X_{\mathbf{T}'}) \sum_{k=1}^{K} \mathbb{1}_{k=J_m}$ is the same for all $m$.

For instance, when $n = 2$, the possible permutations of $\mathbf{T}$ are $\mathbf{T} = (1,2,3,4), (1,2,4,3), (1,3,2,4), \ldots$, resulting in 24 distinct patterns and thus

$$P_{k,1} = \mathbb{E}_{\substack{\mathbb{E} \\ P(\mathbf{T}')}} q(\bar{\mathbf{J}},\mathbf{J}|\mathbf{e},\phi,X_{\mathbf{T}'}) \mathbb{1}_{k=\bar{J}_1} = \mathbb{E}_{\frac{1}{4}q(J_1|\mathbf{e},\phi,X_1)+\frac{1}{4}q(J_1|\mathbf{e},\phi,X_2)+\frac{1}{4}q(J_1|\mathbf{e},\phi,X_3)+\frac{1}{4}q(J_1|\mathbf{e},\phi,X_4)} \mathbb{1}_{k=J_1}$$

$$P_{k,2} = \mathbb{E}_{\substack{\mathbb{E} \\ P(\mathbf{T}')}} q(\bar{\mathbf{J}},\mathbf{J}|\mathbf{e},\phi,X_{\mathbf{T}'}) \mathbb{1}_{k=\bar{J}_2} = \mathbb{E}_{\frac{1}{4}q(J_2|\mathbf{e},\phi,X_1)+\frac{1}{4}q(J_2|\mathbf{e},\phi,X_2)+\frac{1}{4}q(J_2|\mathbf{e},\phi,X_3)+\frac{1}{4}q(J_2|\mathbf{e},\phi,X_4)} \mathbb{1}_{k=J_2}$$

$\vdots$

Thus, all $P_{k,m}$ does not depend on the index $m$. So we express $\mathbb{E}_{\underset{P(\mathbf{T}')}{\mathbb{E}}\, q(\bar{\mathbf{J}},\mathbf{J}|\mathbf{e},\phi,X_{\mathbf{T}'})} \sum_{k=1}^{K} \mathbb{1}_{k=J_m}$ as $P_k$. Then Eq. (30) can be written as

$$\mathbb{E}_X \mathbb{E}_{P(\mathbf{T})q(\mathbf{e},\theta,\phi|X_{\mathbf{T}_1})} \left( \frac{1}{n} \sum_{m=1}^{n} X_{\mathbf{T}_{1,m}} - \frac{1}{n} \sum_{m=1}^{n} X_{\mathbf{T}_{0,m}} \right) \cdot \sum_{k=1}^{K} g_\theta(e_k) P_k$$

$$= \mathbb{E}_{P(\mathbf{T})} \mathbb{E}_X \left( \frac{1}{n} \sum_{m=1}^{n} X_{\mathbf{T}_{1,m}} - \frac{1}{n} \sum_{m=1}^{n} X_{\mathbf{T}_{0,m}} \right) \cdot q(\mathbf{e},\theta,\phi|X_{\mathbf{T}_1}) \sum_{k=1}^{K} g_\theta(e_k) P_k$$

$$= \mathbb{E}_{P(\mathbf{T})} \mathbb{E}_{X_{\mathbf{T}_1}} \mathbb{E}_{X_{\mathbf{T}_0}} \left( \frac{1}{n} \sum_{m=1}^{n} X_{\mathbf{T}_{1,m}} - \frac{1}{n} \sum_{m=1}^{n} X_{\mathbf{T}_{0,m}} \right) \cdot q(\mathbf{e},\theta,\phi|X_{\mathbf{T}_1}) \sum_{k=1}^{K} g_\theta(e_k) P_k$$

$$= \mathbb{E}_{P(\mathbf{T})} \mathbb{E}_{X_{\mathbf{T}_1}} \left( \frac{1}{n} \sum_{m=1}^{n} X_{\mathbf{T}_{1,m}} - \mathbb{E}_{X_{\mathbf{T}_0}} \frac{1}{n} \sum_{m=1}^{n} X_{\mathbf{T}_{0,m}} \right) \cdot q(\mathbf{e},\theta,\phi|X_{\mathbf{T}_1}) \sum_{k=1}^{K} g_\theta(e_k) P_k$$

$$= \mathbb{E}_{P(\mathbf{T})} \mathbb{E}_{X_{\mathbf{T}_1}} \left( \frac{1}{n} \sum_{m=1}^{n} X_{\mathbf{T}_{1,m}} - \mathbb{E}_X X \right) \cdot q(\mathbf{e},\theta,\phi|X_{\mathbf{T}_1}) \sum_{k=1}^{K} g_\theta(e_k) P_k$$

$$\leq \mathbb{E}_{P(\mathbf{T})} \mathbb{E}_{X_{\mathbf{T}_1}} q(\mathbf{e},\theta,\phi|X_{\mathbf{T}_1}) \left\| \frac{1}{n} \sum_{m=1}^{n} X_{\mathbf{T}_{1,m}} - \mathbb{E}_X X \right\| \mathbb{E}_{P(\mathbf{T})} \mathbb{E}_{X_{\mathbf{T}_1}} q(\mathbf{e},\theta,\phi|X_{\mathbf{T}_1}) \left\| \sum_{k=1}^{K} g_\theta(e_k) P_k \right\|_\infty$$

$$\leq \mathbb{E}_{P(\mathbf{T})} \mathbb{E}_{X_{\mathbf{T}_1}} q(\mathbf{e},\theta,\phi|X_{\mathbf{T}_1}) \left\| \frac{1}{n} \sum_{m=1}^{n} X_{\mathbf{T}_{1,m}} - \mathbb{E}_X X \right\| \mathbb{E}_{P(\mathbf{T})} \mathbb{E}_{X_{\mathbf{T}_1}} q(\mathbf{e},\theta,\phi|X_{\mathbf{T}_1}) \left\| \sum_{k=1}^{K} g_\theta(e_k) P_k \right\|_\infty$$

$$\leq \mathbb{E}_{P(\mathbf{T})} \mathbb{E}_{X_{\mathbf{T}_1}} \left\| \frac{1}{n} \sum_{m=1}^{n} X_{\mathbf{T}_{1,m}} - \mathbb{E}_X X \right\| \sqrt{\Delta}.$$

We bound the above exactly same ways as Eq. (20), that is, we can upper bound the above by the variance of bounded random variable and thus, we have

$$\mathbb{E}_{P(\mathbf{T})} \mathbb{E}_{X_{\mathbf{T}_1}} \left\| \frac{1}{n} \sum_{m=1}^{n} X_{\mathbf{T}_{1,m}} - \mathbb{E}_X X \right\| \leq \sqrt{\frac{\Delta}{4n}}.$$

Thus, we have

$$\Omega = \mathbb{E}_X \mathbb{E}_{P(\mathbf{T})q(\mathbf{e},\theta,\phi|X_{\mathbf{T}_1})} \left( \frac{2}{n} \sum_{m=1}^{n} X_{\mathbf{T}_{1,m}} - \frac{2}{n} \sum_{m=1}^{n} X_{\mathbf{T}_{0,m}} \right) \cdot \sum_{k=1}^{K} g_\theta(e_k) P_k \leq \frac{\Delta}{\sqrt{n}},$$

Let us back to the evaluation of the exponential moment in Eq. (29), we will evaluate the following

$$\mathbb{E}_X \frac{1}{\lambda} \mathrm{KL}(\mathbf{Q}|\mathbf{P}) + \mathbb{E}_X \frac{1}{\lambda} \log \mathbb{E}_\mathbf{P} \exp \left( \frac{2\lambda}{n} \sum_{m=1}^{n} \left( X_{\mathbf{T}_{1,m}} - X_{\mathbf{T}_{0,m}} \right) \cdot \sum_{k=1}^{K} g_\theta(e_k) \mathbb{1}_{k=J_m} - \lambda\Omega \right) + \Omega. \tag{31}$$

We then evaluate this similarly to Eq. (28), which uses the negative association of the permutation distribution and McDiarmid's inequality. The the exponential moment is upper bounded by $(2\Delta\lambda/n)^2/8 \times 2n = \lambda^2\Delta^2/n$ We then obtain

$$\mathbb{E}_{X,\mathbf{T}} \sum_{k=1}^{K} \frac{1}{n} \sum_{m=1}^{n} \mathbb{E}_{q(J_m|\mathbf{e},\phi,X_{\mathbf{T}_{1,m}})q(\mathbf{e},\phi,\theta|X_{\mathbf{T}_1})} \| X_{\mathbf{T}_{1,m}} - g_\theta(e_k) \|^2 \mathbb{1}_{k=J_m}$$

$$- \mathbb{E}_{X,\mathbf{T}} \sum_{k=1}^{K} \frac{1}{n} \sum_{m=1}^{n} \mathbb{E}_{q(J_m|\mathbf{e},\phi,X_{\mathbf{T}_{0,m}})q(\mathbf{e},\phi,\theta|X_{\mathbf{T}_1})} \| X_{\mathbf{T}_{0,m}} - g_\theta(e_k) \|^2 \mathbb{1}_{k=J_m}$$

$$\leq \mathbb{E}_X \frac{1}{\lambda} \mathrm{KL}(\mathbf{Q}|\mathbf{P}) + \mathbb{E}_X \frac{1}{\lambda} \log \mathbb{E}_\mathbf{P} \exp \left( \frac{2\lambda}{n} \sum_{m=1}^{n} \left( X_{\mathbf{T}_{1,m}} - X_{\mathbf{T}_{0,m}} \right) \cdot \sum_{k=1}^{K} g_\theta(e_k) \mathbb{1}_{k=J_m} - \lambda\Omega \right) + \Omega$$

$$\leq \mathbb{E}_X \frac{1}{\lambda} \mathrm{KL}(\mathbf{Q}|\mathbf{P}) + \frac{\lambda\Delta^2}{n} + \frac{\Delta}{\sqrt{n}}. \tag{32}$$

In conclusion, from Eqs. (28) and (32) we have

$$\text{gen}(n, \mathcal{D}) \leq \mathbb{E}_X \frac{2}{\lambda} \text{KL}(\mathbf{Q}|\mathbf{P}) + \frac{5\lambda\Delta^2}{4n} + \frac{\Delta}{\sqrt{n}},$$

and optimizing the $\lambda$, we have

$$\text{gen}(n, \mathcal{D}) \leq 2\Delta \sqrt{\frac{5\mathbb{E}_X \text{KL}(\mathbf{Q}|\mathbf{P})}{2n}} + \frac{\Delta}{\sqrt{n}}.$$

We can slightly improve the coefficient of the first term in the above bound as follows. The above proof follows the approach in Appendix D.1. We separately apply the Donsker-Valadhan lemma for the first two terms and latter two terms in Eq. (25). However, since the posterior and prior distributions used for the Donsker-Valadhan lemma are the same as shown in Eq. (26), we only need to use the Donsker-Valadhan lemma once. This leads to an improved coefficient.

Specifically, the proof goes as follows; combining Eqs. (27) and (31), we have simultaneously treat all terms in Eq. (25). By Donsker-Valadhan lemma, we have
$$\text{gen}(n, \mathcal{D})$$

$$\leq \mathbb{E}_X \frac{1}{\lambda} \text{KL}(\mathbf{Q}|\mathbf{P}) + \mathbb{E}_X \frac{1}{\lambda} \log \mathbb{E}_{\mathbf{P}}$$

$$\exp\left( \frac{\lambda}{n} \sum_{k=1}^{K} \sum_{m=1}^{n} l(X_{\mathbf{T}_{0,m}}, g_\theta(e_k)) \left( \mathbb{1}_{k=\bar{J}_m} - \mathbb{1}_{k=J_m} \right) + \frac{2\lambda}{n} \sum_{m=1}^{n} \left( X_{\mathbf{T}_{1,m}} - X_{\mathbf{T}_{0,m}} \right) \cdot \sum_{k=1}^{K} g_\theta(e_k) \mathbb{1}_{k=J_m} - \lambda\Omega \right) + \Omega.$$

From the negative association property, the exponential moment term can be upper-bounded as

$$\log \mathbb{E}_{P(\mathbf{T})q(\mathbf{e},\theta,\phi|X_{\mathbf{T}_1})} \underset{P(\mathbf{T}')}{\mathbb{E}} q(\bar{\mathbf{J}},\mathbf{J}|\mathbf{e},\phi,X_{\mathbf{T}'}) \mathbb{E}_{\prod_{m=1}^{n} \prod_{j=0,1} P(\mathbf{T}''_{j,m})}$$

$$\exp\left( \frac{\lambda}{n} \sum_{k=1}^{K} \sum_{m=1}^{n} l(X_{\mathbf{T}_{0,m}}, g_\theta(e_k)) \left( \mathbb{1}_{k=J_{\mathbf{T}''_{0,m}}} - \mathbb{1}_{k=J_{\mathbf{T}''_{1,m}}} \right) + \frac{2\lambda}{n} \sum_{m=1}^{n} \left( X_{\mathbf{T}_{1,m}} - X_{\mathbf{T}_{0,m}} \right) \cdot \sum_{k=1}^{K} g_\theta(e_k) \mathbb{1}_{k=J_{\mathbf{T}''_{1,m}}} - \lambda\Omega \right),$$

Since $\{\mathbf{T}''_{j,m}\}$ are independent, we can apply McDiarmid's inequality. The the exponential moment is upper bounded by $((1+2)\Delta\lambda/n)^2/8 \times 2n = 9\lambda^2\Delta^2/4n$. Thus, we have

$$\text{gen}(n, \mathcal{D}) \leq \mathbb{E}_X \frac{1}{\lambda} \text{KL}(\mathbf{Q}|\mathbf{P}) + 9\lambda^2\Delta^2/4n + \frac{\Delta}{\sqrt{n}}.$$

By optimizing $\lambda$, we have

$$\text{gen}(n, \mathcal{D}) \leq 3\Delta \sqrt{\frac{\mathbb{E}_X \text{KL}(\mathbf{Q}|\mathbf{P})}{n}} + \frac{\Delta}{\sqrt{n}}.$$

### E.2 Proof of Eq. (10) and discussion about the deterministic encoder

First, we can show
$$\mathbb{E}_{\tilde{X},\mathbf{T}} \mathbb{E}_{q(\mathbf{e},\phi|\tilde{X}_{\mathbf{T}_1})} \text{KL}(\mathbf{Q}_{\tilde{\mathbf{J}},\mathbf{T}} \| \mathbf{Q}_{\tilde{\mathbf{J}}}) \leq I(\mathbf{e}, \phi; \mathbf{T}|\tilde{X}) + I(\tilde{\mathbf{J}}; \mathbf{T}|\mathbf{e}, \phi, \tilde{X}).$$
exactly same way as Appendix D.6.

By the definition of the CMI, the CMI is expressed as the difference of entropy and conditional entropy. Since $\tilde{J}$ is discrete, the entropy is always larger than 0. Thus, we have
$$I(\tilde{\mathbf{J}}; \mathbf{T}|\mathbf{e}, \phi, \tilde{X}) \leq H[\tilde{\mathbf{J}}|\mathbf{e}, \phi, \tilde{X}] \leq H[\tilde{\mathbf{J}}|\tilde{X}].$$
where $H$ is the Shannon entropy. Note that the entropy is bounded by the growth function, i.e., the maximum number of different ways in which a dataset of size $2n$ can be classified in $K$. And such quantity is bounded in the proof of Theorem 8 of Harutyunyan et al. [30], thus

$$I(\tilde{\mathbf{J}}; \mathbf{T}|\mathbf{e}, \phi, \tilde{X}) \leq d_K \log\left( \binom{K}{2} \frac{2en}{d_K} \right).$$

holds similarly to Eq. (24).

Thus, by regularizing the capacity of the encoder model (via the Natarajan dimension), the CMI term $I(\tilde{\mathbf{J}}; \mathbf{T}|\mathbf{e}, \phi, \tilde{X})/n$ scales as $\mathcal{O}(\log n)$. See Appendix D.7 for the additional discussion.

### E.3 Proof of Theorem 4

To prove the theorem, we prove a more general result than Theorem 4, and then we apply that result to the specific setting of Theorem 4. Therefore, we first derive such a general result.

#### E.3.1 Discretization in encoder function

Here, we present the results for a general stochastic encoder. For fixed $\phi$ and $\mathbf{e}$, assume that for all $\mathbf{x} \in \tilde{X}$, for any $j \in [K]$, and for a fixed $\delta \in \mathbb{R}^+$, the following holds: $q(J = j|\mathbf{e}, f_\phi(x)) \leq e^{h(\delta)} q(J = j|\mathbf{e}, \hat{f}(x)))$ with $h : \mathbb{R}^+ \to \mathbb{R}^+$.

**Theorem 6.** *Assume that there exists a positive constant $\Delta_z$ such that $\sup_{z,z' \in \mathcal{Z}} \|z - z'\| < \Delta_z$. Then, when using Eq. (2) and under the same setting as Theorem 3, for any $\delta \in (0, 1]$, we have*

$$\text{gen}(n, \mathcal{D}) \leq 2\Delta\sqrt{nh(\delta)} + 3\Delta\sqrt{\frac{2\log\mathcal{N}(\delta, \mathcal{F}, 2n)}{n}} + \frac{\Delta}{\sqrt{n}}.$$

We can show that Eq. (2) satisfies $h(\delta) = 8\beta\Delta_z\delta$, see Appendix E.3.3 for this proof. Thus by substituting this into the above Theorem, we obtain Theorem 4.

*Proof.* When analyzing the contribution of the encoder model to generalization, it is often necessary to discretize the function or parameters of the encoder to control the CMI using the metric entropy of the model. To achieve this, we consider a $\delta$-cover $\hat{f}$ of the function . In this derivation, we examine both the supersample and permutation-invariant settings, highlighting that the supersample setting fails to establish a uniform convergence bound.

First, we begin with the supersample setting. Given a supersample $\tilde{X}$, we recall the definition of the indices. In this theorem, we focus on the distribution of the index defined by the codebook $\mathbf{e}$ and $z \in \mathcal{Z}$, where $z$ represents the output of the encoder $f_\phi(\cdot)$. Thus, we express it as $q(J|\mathbf{e}, z)$. Moreover, in this section, we use the notation $q(\mathbf{e}, \phi, \theta|\tilde{X}, U) = q(\mathbf{e}, \phi, \theta|\tilde{X}_U)$. The joint distribution is then given by:

$$\mathbf{Q}' := P(\tilde{X})P(U)q(\mathbf{e}, \phi, \theta|\tilde{X}, U)q(\tilde{\mathbf{J}}|\mathbf{e}, \tilde{\mathbf{f}})p(\tilde{\mathbf{f}}|\mathbf{f}, U)p(\mathbf{f}|\phi, \tilde{X}),$$
$$\mathbf{Q}'_\delta := P(\tilde{X})P(U)q(\mathbf{e}, \phi, \theta|\tilde{X}, U)q(\tilde{\mathbf{J}}|\mathbf{e}, \tilde{\mathbf{f}})p(\tilde{\mathbf{f}}|\hat{\mathbf{f}}, U)p(\hat{\mathbf{f}}|\mathbf{f})p(\mathbf{f}|\phi, \tilde{X}),$$

where $q(\tilde{\mathbf{J}}|\mathbf{e}, \tilde{\mathbf{f}})$ represents the elementwise application of $q(J|\mathbf{e}, \cdot)$ to $\tilde{\mathbf{f}} \in \mathcal{Z}^{2n}$. And $p(\mathbf{f}|\phi, \tilde{X})$ is the elementwise application of $p(\mathbf{f}|\phi, \cdot)$ to $\tilde{X}$, which simply computes the encoder output for each sample in $\tilde{X}$.

Then, in $p(\hat{\mathbf{f}}|\mathbf{f})$, the discretization process is performed using the $\delta$-cover (thus, it is represented by the Dirac mass). We express this as $p(\hat{\mathbf{f}}|\mathbf{f}) = \delta(\hat{\mathbf{f}}, \hat{\mathbf{f}}^\phi)$, where $\hat{\mathbf{f}}^\phi$ is the selected point from the $\delta$-cover. Then, for $p(\tilde{\mathbf{f}}|\hat{\mathbf{f}}, U)$, we randomly shuffle $\hat{\mathbf{f}} \in \mathbb{R}^{2n}$ with $U$, formally defining $\hat{\mathbf{f}}_{\tilde{U}} := (\mathbf{f}_U, \mathbf{f}_{\bar{U}})$. Thus, we write $p(\tilde{\mathbf{f}}|\hat{\mathbf{f}}, U) = \delta(\tilde{\mathbf{f}}, \hat{\mathbf{f}}_{\tilde{U}})$. Similarly, we define $p(\tilde{\mathbf{f}}|\mathbf{f}, U) = \delta(\tilde{\mathbf{f}}, \mathbf{f}_{\tilde{U}})$.

This definition differs slightly from the posterior distribution in Eq. (14), where we first shuffle $\tilde{X}$ with $U$ before passing it through the encoder. This simple modification allows us to derive the bound based on metric entropy. When evaluating the generalization error bound, we are only concerned with $\tilde{J}$. By integrating out $\tilde{\mathbf{f}}$, $\phi$, and $\hat{\mathbf{f}}$, we focus on the following posterior distributions:

$$\mathbf{Q} := P(\tilde{X})P(U)q(\mathbf{e}, \mathbf{f}, \theta|\tilde{X}, U)p(\tilde{\mathbf{J}}|\mathbf{e}, \mathbf{f}_{\tilde{U}}),$$
$$\mathbf{Q}_\delta := P(\tilde{X})P(U)q(\mathbf{e}, \mathbf{f}, \theta|\tilde{X}, U)p(\tilde{\mathbf{J}}|\mathbf{e}, \hat{\mathbf{f}}_{\tilde{U}}^\phi).$$

To prove this lemma, we first replace the output of the encoder with that obtained using the $\delta$-cover of the encoder network. First note that the generalization error can be written as

$$\text{gen}(n,\mathcal{D}) = \mathbb{E}_{p(\tilde{X})P(U)} \sum_{k=1}^{K} \frac{1}{n} \sum_{m=1}^{n} \mathbb{E}_{q(\bar{J}_m|\mathbf{e},\mathbf{f}_\phi(X_{m,\bar{U}_m}))q(\mathbf{e},\phi,\theta|X_U)} l(X_{m,\bar{U}_m}, g_\theta(e_k)) \mathbb{1}_{k=\bar{J}_m}$$

$$- \sum_{k=1}^{K} \frac{1}{n} \sum_{m=1}^{n} \mathbb{E}_{q(J_m|\mathbf{e},\mathbf{f}_\phi(X_{m,U_m})))q(\mathbf{e},\phi,\theta|X_U)} l((X_{m,U_m}, g_\theta(e_k)) \mathbb{1}_{k=J_m}$$

$$= \mathbb{E}_{p(\tilde{X})p(U)q(\mathbf{e},\phi,\theta|X,U)p(\tilde{\mathbf{J}}|\mathbf{e},\mathbf{f}_{\tilde{U}})} \left[ \sum_{k=1}^{K} \frac{1}{n} \sum_{m=1}^{n} l(X_{m,\bar{U}_m}, g_\theta(e_k)) \mathbb{1}_{k=\bar{J}_m} - l((X_{m,U_m}, g_\theta(e_k)) \mathbb{1}_{k=J_m} \right].$$

We also define the generalization under the delta cover of original function, conditioned o

$$\text{gen}(n,\mathcal{D},\delta) := \mathbb{E}_{p(\tilde{X})p(U)q(\mathbf{e},\phi,\theta|X,U)} \left[ \sum_{k=1}^{K} \frac{1}{n} \sum_{m=1}^{n} \mathbb{E}_{q(\bar{J}_m|\mathbf{e},\hat{\mathbf{f}}(X_{m,\bar{U}_m}))} l(X_{m,\bar{U}_m}, g_\theta(e_k)) \mathbb{1}_{k=\bar{J}_m} \right.$$

$$\left. - \sum_{k=1}^{K} \frac{1}{n} \sum_{m=1}^{n} \mathbb{E}_{q(J_m|\mathbf{e},\hat{\mathbf{f}}(X_{m,U_m}))} l((X_{m,U_m}, g_\theta(e_k)) \mathbb{1}_{k=J_m} \right]$$

$$= \mathbb{E}_{p(\tilde{X})p(U)q(\mathbf{e},\phi,\theta|X,U)p(\tilde{\mathbf{J}}|\mathbf{e},\hat{\mathbf{f}}_{\tilde{U}}^\phi)} \left[ \sum_{k=1}^{K} \frac{1}{n} \sum_{m=1}^{n} l(X_{m,\bar{U}_m}, g_\theta(e_k)) \mathbb{1}_{k=\bar{J}_m} - l((X_{m,U_m}, g_\theta(e_k)) \mathbb{1}_{k=J_m} \right].$$

For the latter purpose, we define

$$\Delta_L := \sum_{k=1}^{K} \frac{1}{n} \sum_{m=1}^{n} l(X_{m,\bar{U}_m}, g_\theta(e_k)) \mathbb{1}_{k=\bar{J}_m} - \sum_{k=1}^{K} \frac{1}{n} \sum_{m=1}^{n} l((X_{m,U_m}, g_\theta(e_k)) \mathbb{1}_{k=J_m}.$$

To evaluate these gap, we apply the Donsker-Valadhan lemma between the two distributions $\mathbf{Q}_J$ and $\mathbf{Q}_{\delta,J}$.

$$\text{gen}(n,\mathcal{D}) \tag{33}$$

$$\leq \text{gen}(n,\mathcal{D},\delta) + \mathop{\mathbb{E}}_{U,X} \mathbb{E}_{q(\mathbf{e},\phi,\theta|X_U)} \frac{1}{\lambda} \text{KL}(\mathbf{Q}\|\mathbf{Q}_\delta) + \mathop{\mathbb{E}}_{U,X} \mathbb{E}_{q(\mathbf{e},\phi,\theta|X_U)} \frac{1}{\lambda} \log \mathbb{E}_{p(\tilde{\mathbf{J}}|\mathbf{e},\hat{\mathbf{f}}_{\tilde{U}}^\phi)} \exp\left(\lambda\Delta_L - \mathbb{E}_{p(\tilde{\mathbf{J}}|\mathbf{e},\hat{\mathbf{f}}_{\tilde{U}}^\phi)}\lambda\Delta_L\right)$$

$$\leq \text{gen}(n,\mathcal{D},\delta) + \frac{2nh(\delta)}{\lambda} + \frac{\lambda\Delta^2}{2},$$

where we evaluated the KL divergence as

$$\text{KL}(\mathbf{Q}\|\mathbf{Q}_\delta) = \mathbb{E}_{\mathbf{Q}} \log \frac{\mathbf{Q}}{\mathbf{Q}_\delta} \leq 2nK \log e^{h(\delta)} = 2nh(\delta).$$

The inequality is owing to the proper that for all $\mathbf{x} \in \tilde{X}$, for any $j \in [K]$, and for a fixed $\delta \in \mathbb{R}^+$, $q(J = j|\mathbf{e}, f_\phi(x)) \leq e^{h(\delta)} q(J = j|\mathbf{e}, \hat{f}(x)))$ holds by assumption. We also evaluated the exponential moment term by using the fact that $-\lambda\Delta \leq \lambda l(X, g_\theta(e_J)) - \frac{\lambda}{n} \sum_{m=1}^{n} l(S_m, g_\theta(e_{J_m})) \leq \lambda\Delta$ to upper bound the exponential moment.

This implies that the first term corresponds to the generalization bound when using the $\delta$-cover of the encoder network. We can bound this term similarly to Theorem 2,

$$\text{gen}(n,\mathcal{D},\delta) \leq 2\Delta \sqrt{\frac{\text{KL}(\mathbf{Q}_\delta'\|\mathbf{P}_\delta') + \text{KL}(\mathbf{Q}_\delta'\|\mathbf{P})}{n}} + \frac{\Delta}{\sqrt{n}},$$

where we consider the following posterior and data-dependent, and data-independent prior distributions:

$$\mathbf{Q}_\delta' := P(\tilde{X})P(U)q(\mathbf{e},\phi,\theta|\tilde{X},U)q(\tilde{\mathbf{J}}|\mathbf{e},\tilde{\mathbf{f}})p(\hat{\mathbf{f}}|\mathbf{f},U)p(\hat{\mathbf{f}}|\mathbf{f})p(\mathbf{f}|\phi,\tilde{X}),$$

$$\mathbf{P}_\delta' := P(\tilde{X})P(U)q(\mathbf{e},\phi,\theta|\tilde{X},U)p(\tilde{\mathbf{J}}|\mathbf{e},\tilde{\mathbf{f}})\mathbb{E}_{U'}p(\tilde{\mathbf{f}}|\hat{\mathbf{f}},U')p(\hat{\mathbf{f}}|\mathbf{f})p(\mathbf{f}|\phi,\tilde{X}),$$

$$\mathbf{P} := P(\tilde{X})P(U)q(\mathbf{e},\phi,\theta|\tilde{X},U)q(\tilde{\mathbf{J}}|\mathbf{e},\tilde{\mathbf{f}})p(\tilde{\mathbf{f}}|\hat{\mathbf{f}},U)\pi(\hat{\mathbf{f}})p(\mathbf{f}|\phi,\tilde{X}),$$

where $\pi(\hat{\mathbf{f}})$ is the data independent prior distribution over the $\delta$-covering, such as the uniform distribution.

Combining these, we have

$$\text{gen}(n, \mathcal{D}) \leq 2\Delta\sqrt{nh(\delta)} + 2\Delta\sqrt{\frac{\text{KL}(\mathbf{Q}'_\delta\|\mathbf{P}'_\delta) + \text{KL}(\mathbf{Q}'_\delta\|\mathbf{P})}{n}} + \frac{\Delta}{\sqrt{n}}.$$

As for the CMI term, we have

$$\text{KL}(\mathbf{Q}'_\delta\|\mathbf{P}'_\delta) \leq 2\log\mathcal{N}(\delta, \mathcal{F}, 2n). \tag{34}$$

The proof of Eq. (34) is shown in below and this term can be bounded $\mathcal{O}(\log n)$ under moderate assumptions.

However, the second term $\text{KL}(\mathbf{Q}'_\delta\|\mathbf{P})$, which corresponds to the empirical KL term, cannot be small as discussed in Theorem 2. That is, under the settings of Lemma 2, the empirical KL behaves $\mathcal{O}(1)$, which is undesirable behavior.

So we consider using the permutation symmetric setting. We can proceed the discretization almost the same in the above super sample setting. IUnder this distribution, the generalization gap can again upper bounded similar to Eq. (33). Then from Theorem 3, we have

$$\text{gen}(n, \mathcal{D}) \leq \mathbb{E}_{\tilde{X}, \mathbf{T}} \sum_{k=1}^{K} \frac{1}{n} \sum_{m=1}^{n} \mathbb{E}_{q(\bar{J}_m|\mathbf{e}, \hat{f}(X_{\mathbf{T}_{0,m}}))q(\mathbf{e}, \phi, \theta|X_{\mathbf{T}_1})} \|X_{\mathbf{T}_{0,m}} - g_\theta(e_k)\|^2 \mathbb{1}_{k=\bar{J}_m}$$

$$- \mathbb{E}_{\tilde{X}, \mathbf{T}} \sum_{k=1}^{K} \frac{1}{n} \sum_{m=1}^{n} \mathbb{E}_{q(J_m|\mathbf{e}, \hat{f}(X_{\mathbf{T}_{1,m}}))q(\mathbf{e}, \phi, \theta|X_{\mathbf{T}_1})} \|X_{\mathbf{T}_{1,m}} - g_\theta(e_k)\|^2 \mathbb{1}_{k=J_m} + 2\Delta\sqrt{nh(\delta)}$$

$$\leq 3\Delta\sqrt{\frac{\text{KL}(\mathbf{Q}'_\delta\|\mathbf{P}'_\delta)}{n}} + \frac{\Delta}{\sqrt{n}} + 2\Delta\sqrt{nh(\delta)},$$

where

$$\mathbf{Q}'_\delta := P(\tilde{X})P(\mathbf{T})q(\mathbf{e}, \phi, \theta|\tilde{X}, \mathbf{T})q(\tilde{\mathbf{J}}|\mathbf{e}, \tilde{\mathbf{f}})p(\tilde{\mathbf{f}}|\hat{\mathbf{f}}, \mathbf{T})p(\hat{\mathbf{f}}|\mathbf{f})p(\mathbf{f}|\phi, \tilde{X}),$$

$$\mathbf{P}'_\delta := P(\tilde{X})P(\mathbf{T})q(\mathbf{e}, \phi, \theta|\tilde{X}, \mathbf{T})p(\tilde{\mathbf{J}}|\mathbf{e}, \tilde{\mathbf{f}})\mathbb{E}_{\mathbf{T}'}p(\tilde{\mathbf{f}}|\hat{\mathbf{f}}, \mathbf{T}')p(\hat{\mathbf{f}}|\mathbf{f})p(\mathbf{f}|\phi, \tilde{X}),$$

We can show that

$$\text{KL}(\mathbf{Q}'_\delta\|\mathbf{P}'_\delta) \leq 2\log\mathcal{N}(\delta, \mathcal{F}, 2n). \tag{35}$$

see Appendix E.3.2 for the proof. We can analyze the behavior of the upper bound of Eq. (35) in Appendix E.4.

Thus, we have

$$\text{gen}(n, \mathcal{D}) \leq 3\Delta\sqrt{\frac{2\log\mathcal{N}(\delta, \mathcal{F}, 2n)}{n}} + \frac{\Delta}{\sqrt{n}} + 2\Delta\sqrt{nh(\delta)}.$$

$\square$

### E.3.2 Proof of Eq. (34)

We consider the following posterior and data-dependent prior distributions

$$\mathbf{Q} := P(\tilde{X})P(U)q(\mathbf{e}, \phi, \theta|\tilde{X}, U)p(\tilde{\mathbf{J}}|\mathbf{e}, \tilde{\mathbf{f}})p(\tilde{\mathbf{f}}|\mathbf{f}, U)p(\mathbf{f}|\phi, \tilde{X})$$

$$\mathbf{P}_S := P(\tilde{X})P(U)q(\mathbf{e}, \phi, \theta|\tilde{X}, U)p(\tilde{\mathbf{J}}|\mathbf{e}, \mathbf{f})\mathbb{E}_{p(U')}p(\tilde{\mathbf{f}}|\mathbf{f}, U')p(\mathbf{f}|\phi, \tilde{X})$$

When using the Donsker-Valadhan inequality, all calculation remains the same except for the KL divergence term as described below

$$\mathbb{E}_{\mathbf{Q}}\log\frac{\mathbf{Q}}{\mathbf{P}_S} = \mathbb{E}_{p(\tilde{X})P(U)q(\mathbf{e},\phi,\theta|\tilde{X},U)p(\tilde{\mathbf{J}}|\mathbf{e},\tilde{\mathbf{f}})p(\tilde{\mathbf{f}}|\mathbf{f},U)p(\mathbf{f}|\phi,\tilde{X})}\log\frac{p(\tilde{\mathbf{f}}|\mathbf{f},U)}{\mathbb{E}_{p(U')}p(\tilde{\mathbf{f}}|\mathbf{f},U')}$$

$$= \mathbb{E}_{p(\tilde{X})P(U)q(\mathbf{f}|X,U)p(\tilde{\mathbf{f}}|\mathbf{f},U)}\log\frac{p(\tilde{\mathbf{f}}|\mathbf{f},U)}{\mathbb{E}_{p(U')}p(\tilde{\mathbf{f}}|\mathbf{f},U')}$$

$$= \mathbb{E}_{p(\tilde{X})P(\mathbf{f}|X)}\mathbb{E}_{P(U|\mathbf{f},X)p(\tilde{\mathbf{f}}|\mathbf{f},U)}\log\frac{p(\tilde{\mathbf{f}}|\mathbf{f},U)}{\mathbb{E}_{p(U')}p(\tilde{\mathbf{f}}|\mathbf{f},U')}$$

$$= \mathbb{E}_{p(\tilde{X})P(\mathbf{f}|X)}\mathbb{E}_{P(U|\mathbf{f},X)p(\tilde{\mathbf{f}}|\mathbf{f},U)}\log\frac{p(\tilde{\mathbf{f}}|\mathbf{f},U)}{\mathbb{E}_{p(U'|\mathbf{f},X)}p(\tilde{\mathbf{f}}|\mathbf{f},U')}$$

$$\quad + \mathbb{E}_{p(\tilde{X})P(\mathbf{f}|X)}\mathbb{E}_{P(U|\mathbf{f},X)p(\tilde{\mathbf{f}}|\mathbf{f},U)}\log\frac{\mathbb{E}_{p(U'|\mathbf{f},X)}p(\tilde{\mathbf{f}}|\mathbf{f},U')}{\mathbb{E}_{p(U')}p(\tilde{\mathbf{f}}|\mathbf{f},U')}$$

$$= I(\tilde{\mathbf{f}};U|\mathbf{f},X) + \mathbb{E}_{p(\tilde{X})P(\mathbf{f}|X)}\mathbb{E}_{P(U|\mathbf{f},X)p(\tilde{\mathbf{f}}|\mathbf{f},U)}\log\frac{\mathbb{E}_{p(U'|\mathbf{f},X)}p(\tilde{\mathbf{f}}|\mathbf{f},U')}{\mathbb{E}_{p(U')}p(\tilde{\mathbf{f}}|\mathbf{f},U')}$$

$$\leq I(\tilde{\mathbf{f}};U|\mathbf{f},X) + \mathbb{E}_{p(\tilde{X})P(\mathbf{f}|X)}\mathbb{E}_{P(U|\mathbf{f},X)p(\tilde{\mathbf{f}}|\mathbf{f},U)}\log\frac{p(U'|\mathbf{f},X)}{p(U')}$$

$$= I(\tilde{\mathbf{f}};U|\mathbf{f},X) + \mathbb{E}_{p(\tilde{X})P(\mathbf{f}|X)}\mathbb{E}_{P(U|\mathbf{f},X)p(\tilde{\mathbf{f}}|\mathbf{f},U)}\log\frac{p(U'|X)p(\mathbf{f}|U',X)}{\mathbb{E}_{p(U'|X)}p(\mathbf{f}|U',X)p(U')}$$

$$= I(\tilde{\mathbf{f}};U|\mathbf{f},X) + I(\mathbf{f};U|X)$$

We can derive the similar arguments for $\mathbf{Q}'_\delta$ and $\mathbf{P}'_\delta$, and we have

$$\mathrm{KL}(\mathbf{Q}'_\delta\|\mathbf{P}'_\delta) \leq I(\tilde{\mathbf{f}};U|\hat{\mathbf{f}},X) + I(\hat{\mathbf{f}};U|X)$$

Note that we consider the CMI for the discrete variable, it is upper bounded by the entropy [15], and we have

$$I(\tilde{\mathbf{f}};U|\hat{\mathbf{f}},X) \leq H[\tilde{\mathbf{f}}|\hat{\mathbf{f}},X] - H[\tilde{\mathbf{f}}|U,\mathbf{f},X] \leq H[\tilde{\mathbf{f}}|X] \leq \log\mathcal{N}(\delta,\mathcal{F},2n).$$

and

$$I(\hat{\mathbf{f}};U|X) \leq H[\hat{\mathbf{f}}|X] - H[\hat{\mathbf{f}}|U,X] \leq H[\hat{\mathbf{f}}|X] \leq \log\mathcal{N}(\delta,\mathcal{F},2n).$$

The first inequality follows from the fact that MI is defined as the difference between the entropy and the conditional entropy, and the entropy of discrete variables is always non-negative. The second inequality arises because $\bar{\mathbf{J}}, \mathbf{J}$ are outputs of a function evaluated at $2n$ points. Thus, we considered the covering number at $2n$ points, defined as $\mathcal{N}(\delta,\mathcal{F},n) := \sup_{x^{2n}\in\mathcal{X}^{2n}}\mathcal{N}(\delta,\mathcal{F},x^{2n})$. Since the entropy is bounded above by the logarithm of the maximum cardinality, we obtain the second inequality.

### E.3.3   Behavior of Eq. (2)

Finally, we show that Eq. (2) satisfies $h(\delta) = 8\beta\Delta_z\delta$ because

$$\frac{q(J=j|\mathbf{e},f_\phi(x))}{q(J=j|\mathbf{e},\hat{f}(x))}$$

$$= \frac{e^{-\beta\|f_\phi(x)-e_j\|^2}}{e^{-\beta\|\hat{f}(x)-e_j\|^2}} \times \frac{\sum_{k=1}^{K}e^{-\beta\|\hat{f}(x)-e_k\|^2}}{\sum_{k=1}^{K}e^{-\beta\|f_\phi(x)-e_k\|^2}}$$

$$= e^{-\beta\|f_\phi(x)-e_j\|^2+\beta\|\hat{f}(x)-e_j\|^2} \times \frac{\sum_{k=1}^{K}e^{\beta\|f_\phi(x)-e_k\|^2}}{\sum_{k=1}^{K}e^{\beta\|\hat{f}(x)-e_k\|^2}}$$

$$\leq e^{\beta(\hat{f}(x)-f_\phi(x))\cdot(\hat{f}(x)+f_\phi(x))-2\beta e_j\cdot(\hat{f}(x)-f_\phi(x))} \times \sup_{k\in[K]}e^{-\beta\|\hat{f}(x)-e_k\|^2+\beta\|f_\phi(x)-e_k\|^2}$$

$$\leq e^{4\beta\Delta_z\delta} \times e^{4\beta\Delta_z\delta}.$$

### E.4  Discussion about the metric entropy for regularized model

Here we discuss the upper bound of metric entropy in our setting. Since the latent variable lies in $\mathbb{R}^{d_z}$, the encoder network operates as $f_\phi : \mathbb{R}^d \to \mathbb{R}^{d_z}$, making it a multivariate function.

Let us define a function class $\mathcal{F}_i : \mathcal{X} \to \mathbb{R}$ for $i = 1 \dots, d_z$ and define $\mathcal{F}_0 = \prod_{i=1}^{d_z} \mathcal{F}$. Then by definition, $\mathcal{F} \subset \mathcal{F}_0$ holds. We define the covering number for each $\mathcal{F}_i$; Given $x^n := (x_1, \dots, x_n) \in \mathcal{X}^n$, define the pseudo-metric $d'_n$ on $\mathcal{F}_i$ as $d'_n(f, g) := \max_{i \in [n]} |f(x_i) - g(x_i)|$ for $f, g \in \mathcal{F}_i$. The $\delta$-covering number of $\mathcal{F}_i$ with respect to $d'_n$ is denoted as $\mathcal{N}(\delta, \mathcal{F}_i, x^n)$, and we define $\mathcal{N}(\delta, \mathcal{F}_i, n) := \sup_{x^n \in \mathcal{X}^n} \mathcal{N}(\delta, \mathcal{F}_i, x^n)$. Then by definition, the cardinality of $\mathcal{F}$ is smaller than $\mathcal{F}_0$, so we have

$$\mathcal{N}(\delta, \mathcal{F}, n) \leq \prod_{i=1}^{d_z} \mathcal{N}(\delta, \mathcal{F}_i, n).$$

We can see a similar argument in Lemma 1 in Guermeur [26], which considers more general settings.

For simplicity, we assume that $\mathcal{F}' = \mathcal{F}_1 = \cdots = \mathcal{F}_{d_z}$ holds. Then, we can rewrite Theorem 4 as follows

$$\text{gen}(n, \mathcal{D}) \leq 4\Delta \sqrt{2n\beta\Delta_z \delta} + 3\Delta \sqrt{\frac{2d_z \log \mathcal{N}(\delta, \mathcal{F}', 2n)}{n}} + \frac{\Delta}{\sqrt{n}}.$$

For example, assume that the encoder function, which has $d_\phi$ dimensional parameters, shows $L_0$-Lipschitz continuity ($L_0 > 0$) with respect to parameter, then we can obtain $\log \mathcal{N}(\mathcal{F}, \|\cdot\|_\infty, \delta) \asymp d_\phi \log \frac{L_0}{\delta}$ [75]. Thus, by setting $\delta = \mathcal{O}(1/(n))$, we have that

$$\text{gen}(n, \mathcal{D}) = \mathcal{O}\left( \sqrt{\frac{d_\phi d_z \log(n)}{n}} \right)$$

Instead of using the assumption of parametric function class, the metric entropy can be bounded by the fat-shattering dimension of each function, as discussed in Lemma 3.5 of Alon et al. [5]. Examples of fat-shattering dimension evaluations can be found, for instance, in Bartlett & Maass [7], which discusses neural network models, and Gottlieb et al. [24], which addresses the fat-shattering dimension of Lipschitz function classes. If our encoder network adheres to these properties, we can bound its covering number accordingly.

As discussed in Appendix D.7.1, when we use the deterministic decoder, we can use the Natarajan dimension to quantify the complexity of the LVs and such Natarajan dimension can be bounded by the fat-shattering dimension. Thus, it is essential to bound the fat-shattering dimension in both deterministic and stochastic settings.

## F  Proof of Theorem 5

Before the proof, we define the Wasserstein distance. Given a metric $d(\cdot, \cdot)$ and probability distributions $p$ and $q$ on $\mathcal{X}$, let $\Pi(p, q)$ denote the set of all couplings of $p$ and $q$. The 2-Wasserstein distance is defined as:

$$W_2(p, q) = \sqrt{\inf_{\rho \in \Pi} \int_{\mathcal{X} \times \mathcal{X}} d(x, x')^2 d\rho(x, x')}.$$

In this work, we use the Euclidean metric $|\cdot|$ as $d(\cdot, \cdot)$.

Next, we define the pushforward. Let $\pi$ represent a distribution on $\mathcal{Z}$, and let us assume that for any $\theta \in \Theta$, the decoder $g_\theta(\cdot) : \mathcal{Z} \to \mathcal{X}$ is measurable. The pushforward of the distribution $\pi$ by the decoder, denoted as $g_\theta \# \pi$, defines a distribution on $\mathcal{X}$ as $g_\theta \# \pi(A) = \pi(g_\theta^{-1}(A))$ for any measurable set $A \subseteq \mathcal{X}$.

*Proof.* Conditioned on the encoder parameter, codebook, and input $X$, selecting the index $J$ corresponds to selecting the latent representation $e_J$. Since the posterior over the index is $q(J|\mathbf{e}, \phi, X)$, we express the posterior imposed on the latent representation as $q(e = e_j|\mathbf{e}, \phi, X)$ for all $j = 1, \dots, K$.

Using this notation, we first define the distribution obtained by the training dataset as follows; conditioned on $\mathbf{e}, \phi, S$, we have

$$\hat{\mu}_S = \frac{1}{n} \sum_{m=1}^{n} g_\theta \# q(e|\mathbf{e}, \phi, S_m).$$

From the triangle inequality, we have

$$W_2(\mathcal{D}, \hat{\mu}) \leq W_2(\mathcal{D}, \hat{\mu}_S) + W_2(\hat{\mu}_S, \hat{\mu}). \tag{36}$$

We then have

$$W_2^2(\mathcal{D}, \hat{\mu}) \leq 2W_2^2(\mathcal{D}, \hat{\mu}_S) + 2W_2^2(\hat{\mu}_S, \hat{\mu}).$$

The first term of Eq. (36) is bounded as follows;

$$\mathbb{E}_S \mathbb{E}_{q(\mathbf{e}, \phi, \theta|S)} W_2^2(\mathcal{D}, \hat{\mu}_S) \leq \mathbb{E}_S \mathbb{E}_{q(\mathbf{e}, \phi, \theta|S)} \mathbb{E}_X \frac{1}{n} \sum_{m=1}^{n} \mathbb{E}_{q(e|\mathbf{e}, \phi, S_m)} \|X - g_\theta(e)\|^2$$

$$= \mathbb{E}_S \mathbb{E}_{q(\mathbf{e}, \phi, \theta|S)} \mathbb{E}_X \sum_{k=1}^{K} \|X - g_\theta(e_k)\|^2 \frac{1}{n} \sum_{m=1}^{n} \mathbb{E}_{q(J_m|\mathbf{e}, \phi, S_m)} \mathbb{1}_{k=J_m} \tag{37}$$

The first inequality is obtained by the definition of the Wasserstein distance.

This term corresponds to the first term of Eq. (17), where $X$ corresponds to the test data $X_{m, \bar{U}_m}$. Therefore, Eq. (37) can be upper-bounded by applying Eq. (21), which serves as the upper bound for Eq. (17).

$$\mathbb{E}_S \mathbb{E}_{q(\mathbf{e}, \phi, \theta|S)} W_2^2(\mathcal{D}, \hat{\mu}_S) \tag{38}$$

$$\leq \mathbb{E}_S \mathbb{E}_{q(\mathbf{e}, \phi, \theta|S)} \frac{1}{n} \sum_{m=1}^{n} \mathbb{E}_{q(J_m|\mathbf{e}, \phi, S_m)} \|S_m - g_\theta(e_{J_m})\|^2 + \frac{1}{\lambda} \mathrm{KL}(\mathbf{Q}|\mathbf{P}) + \frac{\lambda \Delta^2}{2n} + \frac{\Delta}{\sqrt{n}},$$

where

$$\mathbf{Q} := q(\mathbf{e}, \phi, \theta|S) \prod_{m=1}^{n} q(J_m|\mathbf{e}, \phi, S_m), \qquad \mathbf{P} := q(\mathbf{e}, \phi, \theta|S) \prod_{m=1}^{n} \pi(J_m|\mathbf{e}, \phi).$$

Next, the second term of Eq. (36) is bounded as follows; we use the weighted CKP inequality [11].From the particular case 2.5. in Bolley & Villani [11], we directly have

$$\mathbb{E}_S \mathbb{E}_{q(\mathbf{e}, \phi, \theta|S)} W_2^2(\hat{\mu}_S, \hat{\mu}) \leq \Delta \sqrt{2\mathrm{KL}(\hat{\mu}_S\|\hat{\mu})} \leq \Delta \sqrt{2\frac{1}{n} \sum_{m=1}^{n} \mathrm{KL}(g_\theta \# q(e|\mathbf{e}, \phi, S_m)\|g_\theta \# \pi(e|\mathbf{e}, \phi))}$$

$$\tag{39}$$

$$\leq \Delta \sqrt{2\frac{1}{n} \sum_{m=1}^{n} \mathrm{KL}(q(J_m|\mathbf{e}, \phi, S_m)\|\pi(J_m|\mathbf{e}, \phi))}$$

Combining Eqs. (38) and (39), we have

$$\mathbb{E}_S \mathbb{E}_{q(\mathbf{e}, \phi, \theta|S)} W_2^2(\mathcal{D}, \hat{\mu}) \leq 2\mathbb{E}_S \mathbb{E}_{q(\mathbf{e}, \phi, \theta|S)} \frac{1}{n} \sum_{m=1}^{n} \mathbb{E}_{q(e_{(m)}|\mathbf{e}, \phi, S_m)} \|S_m - g_\theta(e_{(m)})\|^2$$

$$+ \frac{2}{\lambda} \mathrm{KL}(\mathbf{Q}|\mathbf{P}) + \frac{\lambda \Delta^2}{n} + \frac{2\Delta}{\sqrt{n}} + 2\Delta \sqrt{2\frac{1}{n} \sum_{m=1}^{n} \mathrm{KL}(q(J_m|\mathbf{e}, \phi, S_m)\|\pi(J_m|\mathbf{e}, \phi))}.$$

Then by optimizing $\lambda$, we have

$$\mathbb{E}_S \mathbb{E}_{q(\mathbf{e}, \phi, \theta|S)} W_2^2(\mathcal{D}, \hat{\mu})$$

$$\leq \mathbb{E}_S \mathbb{E}_{q(\mathbf{e}, \phi, \theta|S)} \frac{2}{n} \sum_{m=1}^{n} \mathbb{E}_{q(e_{(m)}|\mathbf{e}, \phi, S_m)} \|S_m - g_\theta(e_{(m)})\|^2 + 4\Delta \sqrt{2\frac{1}{n} \sum_{m=1}^{n} \mathrm{KL}(q(J_m|\mathbf{e}, \phi, S_m)\|\pi(J_m|\mathbf{e}, \phi))} + \frac{2\Delta}{\sqrt{n}}.$$

$$\square$$

# G Experimental settings and additional experimental results

Our experiments were based on the Gaussian stochastically quantized VAE (SQ-VAE) model proposed by Takida et al. [64], and were conducted by adapting the code from their GitHub [1]to suit our experimental configurations. Therefore, we first introduce the basics of (Gaussian) SQ-VAE in Sections G.1 and G.2 and finally explain our experimental settings in Section G.3.

## G.1 Overview of SQ-VAE

The SQ-VAE is a generative model that, similar to VQ-VAE, employs a learnable codebook $\mathbf{e} = \{e_k\}_{k=1}^K \in \mathcal{Z}^K$. The objective of SQ-VAE is to learn the *stochastic decoder* $x \sim p_\theta(x|Z_q)$ using latent variables $Z_q$ to generate samples belonging to the data distribution $p_{\text{data}}(x)$, where $p_\theta(x|Z_q) = \mathcal{N}(g_\theta(Z_q), \sigma^2 \mathbf{I})$, $\mathcal{N}(m, \sigma \mathbf{I})$ is the Gaussian distribution with mean and equal variance parameter $\{m, \sigma^2 \mathbf{I}\}$, $\sigma^2 \in \mathbb{R}_+$, and $\mathbf{I}$ is the identity matrix. Here, $Z_q$ is sampled from a prior distribution $P(Z_q)$ over the discrete latent space $\mathbf{e}^{d_z}$.

**In the main training process** of SQ-VAE, we assume $P(Z_q)$ to be an i.i.d. uniform distribution, identical to VQ-VAE, meaning each codebook element is selected with equal probability ($P(z_{q,i} = b_k) = 1/K$ for $k \in [K]$). Subsequently, a **second training stage** is conducted to learn $P(Z_q)$. Since computing the posterior $p_\theta(Z_q|x)$ exactly is intractable, we utilize an approximate posterior distribution $q_\phi(Z_q|x)$ instead.

At the encoding process, directly mapping from $x$ to the discrete $Z_q$ is challenging due to the discrete nature of $Z_q$. To overcome this issue, Takida et al. [64] proposed to construct a stochastic encoder by introducing the following two processes:

- **Stochastic Dequantization Process**: The transformation function from $Z_q$ to the auxiliary *continuous* variable, $Z$, denoted as $p_\psi(Z|Z_q)$, where $\psi$ is its parameters.

- **Stochastic Quantization Process**: The transformation from $Z$ to $Z_q$ is given by $\hat{P}_\phi(Z_q|Z) \propto p_\phi(Z|Z_q)P(Z_q)$ obtained via Bayes' theorem, which is represented as the categorical distribution $q(J|\mathbf{e}, \phi, x)$ through the softmax function as in Eq. (2).

We can obtain $\hat{Z}_q$ from a deterministic encoder $f_\phi(x)$, where we expect that $\hat{Z}_q$ is close to $Z_q$. Therefore, we can similarly define the dequantization process of $\hat{Z}_q$ as $Z|\hat{Z}_q \sim p_\psi(Z|\hat{Z}_q)$. By combining this process with the stochastic quantization process, we can establish the following *stochastic encoding* process from $x$ to $Z_q$: $\mathbb{E}_{q_\omega(Z|x)}[\hat{P}_\phi(Z_q|Z)]$, where $\omega := \{\phi, \psi\}$ and $q_\omega(Z|x) := p_\psi(Z|f_\phi(x))$.

According to these facts, we can derive the following evidence lower bound (ELBO) for SQ-VAE:

$$-\mathcal{L}_{\text{SQ}}(x; \theta, \omega, \mathbf{e}) \tag{40}$$
$$:= \underbrace{\mathbb{E}_{q_\omega(Z|x), \hat{P}_\phi(Z_q|Z)}\left[\log \frac{p_\theta(x|Z_q)p_\phi(Z|Z_q)}{q_\omega(Z|x)}\right]}_{=\text{KL}(\mathbf{Q}\|\mathbf{P})} + \mathbb{E}_{q_\omega(Z|x)} H(\hat{P}_\phi(Z_q|Z)) + (\text{Const.}),$$

where $H(\hat{P}_\phi(Z_q|Z))$ is the entropy of $\hat{P}_\phi(Z_q|Z)$.

From the above, the optimization problem of SQ-VAE is minimizing $\mathbb{E}_{p_{\text{data}}(x)}[\mathcal{L}_{\text{SQ}}(x; \theta, \omega, \mathbf{e})]$ w.r.t. $\{\theta, \omega, \mathbf{e}\}$. This approach eliminates the need for heuristic techniques traditionally required, such as stop-gradient, exponential moving average (EMA), and codebook reset [82].

Moreover, the categorical posterior distribution $\hat{P}_\phi(Z_q|Z) = q(J|\mathbf{e}, \phi, x)$ can be approximated using the Gumbel–Softmax relaxation [35, 45], where the Gumbel–Softmax function is defined as, for all $k$ ($1 \le k \le K$),

$$\frac{\exp(-\beta\|f_\phi(x) - e_k\|^2 + G_k)/\tau)}{\sum_{j=1}^K \exp(-\beta\|f_\phi(x) - e_j\|^2 + G_j)/\tau)},$$

---

[1]https://github.com/sony/sqvae/tree/main/vision

Table 1: Experimental settings on MNIST.

| Experimental setup for MNIST experiments | |
|---|---|
| Model | Gaussian stochastically quantized VAE (SQ-VAE) [64] |
| Network archtecture | ConvResNets with three convolutional layers, two transpose convolutional layers, and one ResBlocks. |
| The size of a codebook ($K$) and the dimension of the latent space $d_z$ | $K = \{16, 32, 64, 128\}$; $d_z = 64$ |
| Optimizer | Adam with 0.001 initial learning rate |
| Batch size | 32 |
| Num. of training/validation samples | $[250, 1000, 2000, 4000]$ |
| Num. of epochs | 200 |
| Num. of samples for CMI estimation | 3 |
| Num. of samplings for $U$ | 5 |

Table 2: Experimental settings on CIFAR10.

| Experimental setup for CIFAR10 experiments | |
|---|---|
| Model | Gaussian stochastically quantized VAE (SQ-VAE) [64] |
| Network architecture | ConvResNets with three convolutional layers, two transpose convolutional layers, and one ResBlocks. |
| The size of a codebook ($K$) and the dimension of the latent space $d_z$ | $K = \{16, 32, 64, 128\}$; $d_z = 64$ |
| Optimizer | Adam with 0.001 initial learning rate |
| Batch size | 32 |
| Num. of training/validation samples | $[1000, 5000, 10000, 20000]$ |
| Num. of epochs | 200 |
| Num. of samples for CMI estimation | 3 |
| Num. of samplings for $U$ | 5 |

where $G_k$ is an i.i.d. sample from the Gumbel distribution and $\tau$ is the temperature parameter that is deferent from $\beta$ in Eq. (2). This allows the application of the reparameterization trick from VAEs during backpropagation, enabling efficient gradient computation and model training.

## G.2  Gaussian SQ-VAE

Gaussian SQ-VAE assumes that the dequantization process $p_\psi(Z|Z_q)$ follows a Gaussian distribution. In this paper, we set the following Gaussian distribution: $p_\psi(Z_i|Z_q) = \mathcal{N}(Z_{q,i}, \sigma_\psi^2 \mathbf{I})$, where $\sigma_\psi^2 \in \mathbb{R}_+$. Then, the stochastic decoder and the stochastic dequantization process in SQ-VAE can be written as $p_\theta(x|Z_q) = \mathcal{N}(g_\theta(Z_q), \sigma^2 \mathbf{I})$ and $p_\psi(Z_i|\hat{Z}_q) = \mathcal{N}(\hat{Z}_{q,i}, \sigma_\psi^2 \mathbf{I})$.

## G.3  Details of experimental settings

**Dataset:**  We used the MNIST dataset [41], which is $28 \times 28$ gray scale images with 10 classes. We prepared the subset dataset with $\{1000, 2000, 4000, 8000\}$ samples from the default training dataset (60000 samples). Then, we split it as the training and the validation datasets following the supersample setting as in Section 2.3.

**Model architecture and training procedure:**  We adopted the ConvResNets with the architecture provided by Google DeepMind [2]. We summarize the details of this model in Table 1.

Regarding the training procedure, we adopted the settings in Takida et al. [64] as follows. We used the Adam optimizer with 0.001 initial learning rate. The learning rate was halved every 3 epochs if the validation loss is not improving. We trained the model 200 epochs with 32 mini-batch size. As for the annealing schedule for the temperature parameter of the Gumbel-softmax sampling, we set $\tau = \exp(10^{-5} \cdot t)$ as in Jang et al. [35], where $t$ is the global training step size.

**GPU environment:**  We used NVIDIA GPUs with 32GB memory (NVIDIA DGX-1 with Tesla V100 and DGX-2) in our experiments.

**Mutual information estimation:**  To estimate the mutual information $I(\tilde{\mathbf{J}}; U|\mathbf{e}, \phi, \tilde{X})$ in Eq. (7), we developed a plug-in estimator for it, which is computed using estimators for the probability density of $\tilde{\mathbf{J}}$ and $\tilde{X}$, as well as their joint probability density, employing $k$-nearest-neighbor-based density estimation [43].  The estimation strategy is incorporated into the `sklearn.feature_selection.mutual_info_classif` function [3]. We set $k = 3$ following the default setting of this function and Kraskov et al. [40], Ross [54].

---

[2] https://github.com/deepmind/sonnet/blob/v2/examples/vqvae_example.ipynb

[3] https://scikit-learn.org/stable/modules/generated/sklearn.feature_selection.mutual_info_classif.html

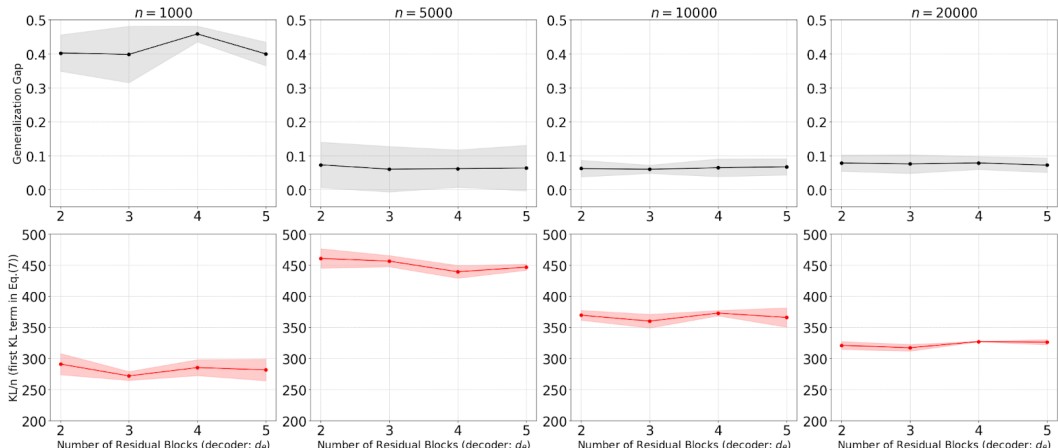

Figure 5: Behavior of the generalization gap and the empirical KL term ($\mathrm{KL}(\mathbf{Q}_{\mathbf{J},U}\|\mathbf{P})/n$) on the CIFAR-10 dataset ($K = 128, d_z = 64$). The top row shows their asymptotic behavior as a function of sample size $n$. The bottom row shows their behavior as the decoder complexity (number of residual blocks) is increased (for $n = 20000$).

## G.4 Additional Experimental Results

Here, we summarize our additional experimental results. These experiments are organized to empirically support the three central claims of our paper, which we present in sequence: (1) the decoder-independent nature of the generalization gap, (2) a detailed analysis of the two KL terms in Theorem 2, and (3) the practical utility of our theoretical framework.

### G.4.1 Validation of Decoder-Independence (Theorems 2, 3, & 4)

A central claim of our paper is that the generalization gap is independent of the decoder $g_\theta$. We validate this claim across various settings.

First, Figure 5 shows the results on the CIFAR-10 dataset, which is more complex than MNIST. These results support our implication: increasing the complexity of $g_\theta$ by adding a ResBlock (introducing approximately 74,000 parameters) has a negligible effect on the generalization gap.

Second, to provide a complete picture for Figure 2 in the main text, we provide its corresponding training losses in Table 3. The table confirms that for larger datasets ($n \geq 1000$), a more expressive decoder (i.e., with more residual blocks) tends to achieve a lower training loss. This observation, when viewed alongside the stable generalization gap in Figure 2, strongly reinforces our central claim: the decoder's capacity to fit the training data is not the primary driver of generalization performance.

Third, we compare the behavior of stochastic (SQ-VAE, Theorem 4) and deterministic (VQ-VAE, Theorems 2 & 3) encoders. The results in Figure 6 show two key findings:

- **SQ-VAE (Stochastic):** As shown in the two left panels, the generalization gap is independent of the decoder complexity (leftmost) and instead depends on the latent dimension $d_z$ (second from left). This is perfectly consistent with Theorem 4, which is independent of the learning algorithm $q(w|S)$ and fully eliminates the decoder's influence.

- **VQ-VAE (Deterministic):** As shown in the two right panels, the gap is also largely independent of the decoder (second from right) and dependent on $d_z$ (rightmost), supporting Theorems 2 & 3. However, we observe a slight tendency for the gap to increase in the moderate complexity range (e.g., 2 to 6 ResBlocks). This does not contradict our theory. Our bounds (which depend on $q(w|S)$) state that the *upper bound* is decoder-independent, implying that while increasing complexity substantially does not worsen the gap, a poorly learned $q(w|S)$ in the moderate range can still affect generalization under that bound.

Overall, these findings suggest that the influence of decoder complexity depends on whether the latent variable mechanism is stochastic or deterministic, which is an important direction for future work.

Table 3: Training loss corresponding to the generalization gap experiments in Figure 2 (top row). As decoder complexity (number of Residual Blocks, RB) increases, the training loss tends to decrease for larger sample sizes ($n \geq 1000$), confirming that a more expressive decoder can better fit the training data.

| $n$ | RB=2 | RB=3 | RB=4 | RB=5 |
|---|---|---|---|---|
| 250 | $6.4851 \pm 0.2642$ | $7.0816 \pm 0.2817$ | $7.6026 \pm 0.2786$ | $7.4940 \pm 0.7172$ |
| 1000 | $3.4664 \pm 0.0293$ | $3.4869 \pm 0.1286$ | $3.3180 \pm 0.0398$ | $3.2609 \pm 0.0905$ |
| 2000 | $2.6391 \pm 0.0177$ | $2.5114 \pm 0.0130$ | $2.4645 \pm 0.0778$ | $2.3915 \pm 0.1214$ |
| 4000 | $2.1102 \pm 0.0478$ | $1.9466 \pm 0.0152$ | $1.9223 \pm 0.0115$ | $1.9001 \pm 0.0475$ |

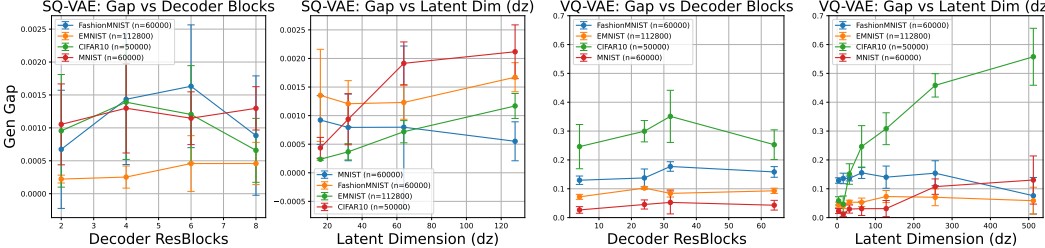

Figure 6: Behavior of the generalization gap when increasing the number of residual blocks of the decoder network and the latent dimension $d_z$ in SQ-VAE (stochastic, left two panels) and VQ-VAE (deterministic, right two panels) models.

### G.4.2 Analysis of the KL Divergence Terms (Theorem 2)

Theorem 2 presents a bound comprising two KL terms. We now empirically analyze the behavior of both terms.

Figure 7 shows the behavior of these terms on the MNIST dataset. As shown in the top row (left and middle panels), the first KL term ($\text{KL}(\mathbf{Q}_{\mathbf{J},U} \| \mathbf{P})/n$) does not decrease monotonically with $n$, consistent with our theoretical claim in Lemma 3. In contrast, the generalization gap (top left) and the second KL term (CMI term, top right) both decrease steadily as $n$ increases. This suggests that the second KL term, not the first, correctly captures the generalization behavior. The bottom row also shows that both KL terms increase with the codebook size $K$, confirming our theoretical predictions.

To make this relationship explicit, we plot the correlation between the generalization gap and each KL term in Figure 3. The results on MNIST are clear: the second KL term (right panel) exhibits a consistent positive correlation ($r \approx 0.46\text{-}0.60$) with the generalization gap across all decoder complexities. Conversely, the first KL term (left panel) shows a negative correlation, as its value does not decrease with $n$ while the generalization gap does.

We further validate this finding on the more complex CIFAR-10 dataset in Figure 8. The trends observed in MNIST are not only confirmed but are even more pronounced. The asymptotic behavior (left three panels) again shows that both the generalization gap and the second KL term decrease with $n$, while the first KL term does not. Most importantly, the correlation plots (right two panels) provide definitive evidence. The second KL term exhibits an **extremely strong and consistent positive correlation ($r > 0.92$) with the generalization gap**. In stark contrast, the first KL term shows a strong negative correlation ($r < -0.58$).

This provides robust empirical evidence that the second term in Eq. (7) (the CMI term) is the component that correctly characterizes generalization behavior.

### G.4.3 Practical Utility of the Data-Dependent Prior

In addition to validating our theoretical bounds, we also investigated the practical utility of our framework by implementing a data-dependent prior following the approach of Sefidgaran et al. [57].

Sefidgaran et al. [57] proposed the *Lossless Category-Dependent Variational Information Bottleneck (CDVIB)*, which is directly motivated by their theoretical results that bound the generalization error of

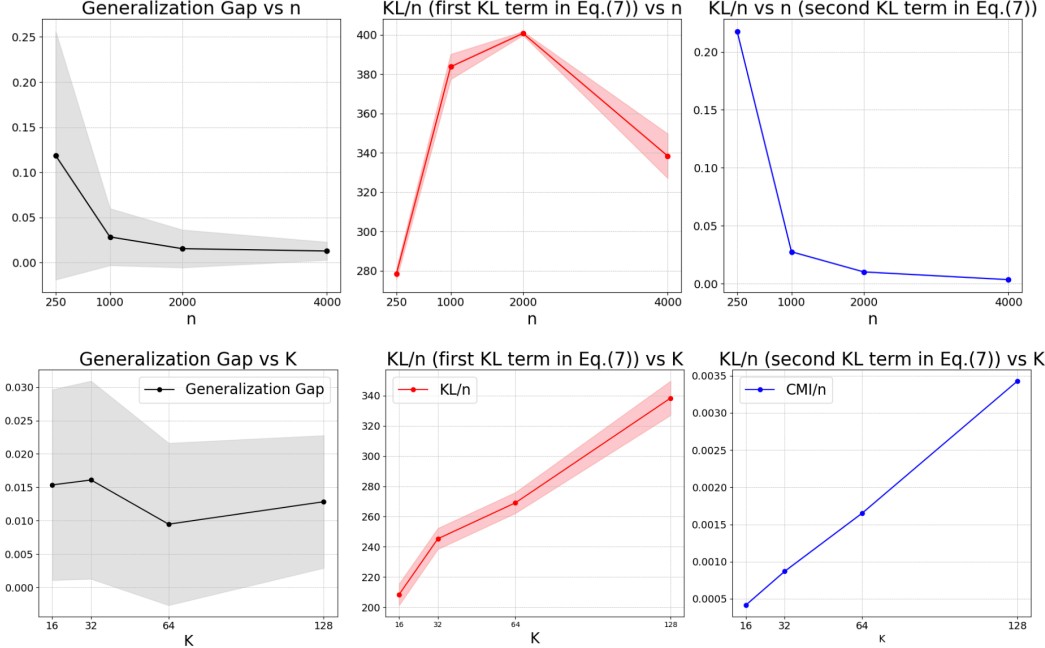

Figure 7: Behavior of the generalization gap and the two KL terms from Eq. (7) on the MNIST dataset ($K = 128, d_z = 64$). **(Top row)** Asymptotic behavior as a function of sample size $n$. **(Bottom row)** Behavior as a function of codebook size $K$ (for $n = 2000$).

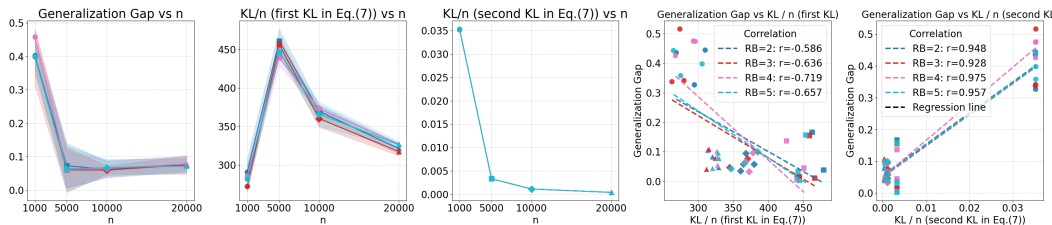

Figure 8: The behavior of the generalization gap and the two KL terms from Eq. (7) on the CIFAR dataset ($K = 128, d_z = 64$). The three leftmost panels show the asymptotic behavior of the generalization gap, the first KL term, and the second KL term as a function of sample size $n$. The two rightmost panels show scatter plots correlating the generalization gap with the first KL term (fourth panel) and the second KL term (fifth panel). In these plots, the color indicates the number of decoder Residual Blocks (RB=2, 3, 4, or 5) and the marker shape indicates the sample size $n$. (Circle for $n = 1000$, Square for $n = 5000$, Diamond for $n = 10000$, and Triangle for $n = 20000$).

encoder–decoder representation learning models. Their analysis demonstrates that the generalization error depends solely on the encoder and latent variables, rather than on the decoder. Consequently, unlike the standard VIB that employs a data-independent prior (e.g., $\mathcal{N}(0, I)$), their bound suggests that *data-dependent priors*—which capture the structure and "simplicity" of the encoder—can tighten theoretical guarantees and improve generalization.

Building upon this insight, the Lossless CDVIB framework introduces a data-dependent Gaussian prior. To implement such a prior, the mean and variance of each prior component are updated at every training iteration $t$ using an exponential moving average of the corresponding batch statistics. This moving average enables the prior to gradually align with the encoder's latent representation, ensuring that the KL regularization term consistently tracks the geometry of the encoder. This adaptive alignment mitigates the mismatch between the encoder's latent distribution and the fixed isotropic prior used in standard VIB. Furthermore, since the "ghost" dataset assumed in the theoretical analysis is unavailable during training, the moving average empirically mimics this expectation by aggregating

Table 4: Reconstruction error comparison between the baseline SQ-VAE (without a data-dependent prior) and our proposed method (with a data-dependent prior) on test dataset. Our method demonstrates consistently lower test loss across all benchmark datasets, validating the practical benefits of our theoretical framework.

| Dataset | SQVAE (baseline) | Proposed method |
|---|---|---|
| CIFAR10 | $10.75 \pm 0.10$ | $10.68 \pm 0.04$ |
| Fashion-MNIST | $1.37 \pm 0.02$ | $1.32 \pm 0.05$ |
| MNIST | $3.23 \pm 0.04$ | $2.99 \pm 0.04$ |

statistics across past mini-batches. In this sense, the moving prior reproduces the averaging effect of the ghost dataset, providing a practical realization of the theoretical setup.

Motivated by these concepts, we introduce a similar data-dependent prior into the ELBO objective in Eq. (40). Specifically, we replace the entropy regularization term in Eq. (40) as follows:

$$\frac{1}{N} \sum_{n=1}^{N} \mathbb{E}_{x_n} \Big[ H(\hat{P}_\phi(Z_q|Z)) \Big] \longrightarrow (1-\beta) \frac{1}{N} \sum_{n=1}^{N} \mathbb{E}_{x_n} \Big[ H(\hat{P}_\phi(Z_q|Z)) \Big] + \beta \, \mathrm{KL}_{\mathrm{CDVIB}}, \quad (41)$$

where $\mathrm{KL}_{\mathrm{CDVIB}}$ is defined as follows. Recall that the entropy term is expressed as

$$\frac{1}{N} \sum_{n=1}^{N} \mathbb{E}_{x_n} \Big[ H(\hat{P}_\phi(Z_q|Z)) \Big] = \frac{1}{N} \sum_{n=1}^{N} \sum_{k=1}^{K} q_{n,k} \, \log q_{n,k},$$

where $q_{n,k}$ denotes the simplified form of $\mathbb{E}\hat{P}_\phi(Z_q|Z)$ for the data point $x_n$. We then define the proposed regularizer as

$$\mathrm{KL}_{\mathrm{CDVIB}} = \frac{1}{N} \sum_{n=1}^{N} \sum_{k=1}^{K} q_{n,k} \, \log \frac{q_{n,k}}{\pi_k},$$

where the denominator $\pi_k$ represents the moving average of the empirical statistics:

$$\hat{p}_k = \frac{1}{N} \sum_{n=1}^{N} q_{n,k},$$

and the data-dependent prior $\pi$ is updated as

$$\pi \leftarrow (1-\alpha)\,\pi + \alpha\,\hat{p}, \qquad \pi \leftarrow \frac{\pi}{\sum_{j=1}^{K} \pi_j}, \qquad \alpha \in (0,1).$$

In practice, the empirical statistics are computed over each mini-batch. We employ a mixture of data-independent and data-dependent priors as the regularization term in Eq. (41). Empirically, we observe that this mixture stabilizes training, while setting $\beta$ too close to 1 often leads to suboptimal performance.

We evaluated the test reconstruction loss in terms of MSE following the experimental protocol of Takida et al. [64]. For all experiments, we fixed $\alpha = 0.9$ and $\beta = 0.5$. The numerical results are reported in Table 4. Here, the MSE represents the total pixel-wise reconstruction loss per image, rather than the per-pixel average. To ensure statistical robustness, we repeated each experiment with ten different random seeds, and report the mean and variance of the MSE across these 10 independent trials. We observed that incorporating the data-dependent prior consistently improves MSE performance across all settings.

