# OpenReview forum: "Information-theoretic Generalization Analysis for VQ-VAEs: A Role of Latent Variables"
_NeurIPS.cc/2025/Conference — NeurIPS 2025 poster_

### Official Review · Reviewer_oZtU · 2025-06-30

**Clarity:** 4
**Significance:** 3
**Originality:** 3
**Rating:** 5
**Confidence:** 3

**Summary:**

This work provides an information-theoretic (IT) analysis of the generalisation capacity and data generation quality of VQ-VAEs.

After showing that the basic IT bound cannot be directly applied to such (unsupervised) setting if one wants to analyse separately the latent variables (LVs) and decoder parameters, the authors propose a new error bound which explicitly encodes the dependence between the latent variables and supersample indexes to remove this dependency. An analysis of this new bound reveals that the generalisation gap solely depends on the complexity of the LVs and encoders' parameters. However, the proposed bound contains the (non-converging) empirical KL divergence, and a new (converging) bound is proposed that removes any dependency on the empirical KL divergence.

Additionally, an upper bound on the 2-Wasserstein distance between the true and generated data is derived to assess the data generation quality, and here again, the authors show that the LVs significantly impact this.

 Overall, this paper demonstrates the importance of LVs' complexity (and their regularisation) for generalisation and data generation quality, providing some insights into why some training techniques worked well in VQ VAEs.

**Questions:**

[1] Could the authors provide a more detailed explanation of the supersample setting from Section 2.3? From what I understand, each entry is an $x$ drawn iid from $\\mathcal{X}$ and then one uses the associated index to select the $x$ from the correct column when one is in test or training setup. But then $\\mathcal{A}(\\mathcal{X}_U)$ would only use one element per row, and so $l_0$ would be an nx1 matrix. However, it is stated that $l_0$ is an nx2 matrix, so there is something that I did not understand correctly about how $\\mathcal{X}_U$ is created.

[2] The $\\tilde{\\cdot}$ and $\\bar{\\cdot}$ are sometimes hard to distinguish (e.g. l. 1044). Could the authors replace the $\\tilde{\\cdot}$ with something more distinctive like $\\breve{\\cdot}$, for example?

[3] l.5 I do not think that the authors truly mean "insufficiently underexplored". I suggest replacing it with either "underexplored" or "insufficiently explored".
l. 378 "that enabling" should probably be replaced by "enabling."

**Ethical Concerns:**

["NO or VERY MINOR ethics concerns only"]

**Final Justification:**

The authors addressed any concerns I had. I still believe that this is a good paper, which should be accepted, but the paper's impact is not sufficient to further raise my score

**Limitations:**

yes

**Quality:**

3

**Strengths And Weaknesses:**

___Quality:___

[+] The paper is sound, and the claims are well-supported. I did not find any issue with the proposed proofs. However, I am an expert in VAEs, not IT, and I may have missed something.

[+] The limitations of the work are clearly stated and extensively discussed in the conclusion and appendix.

[-] It would have been nice to see more of the experimental part in the main paper.


___Clarity:___

[+] The paper is well-written and easy to follow

[+] The maths are clear and easy to follow; each variable is defined before use, and a notation table is provided in the appendix.

[+] The main theoretical results are generally followed by one or two sentences summarising the key points in layperson's terms. This is greatly appreciated as it avoids breaking the flow of the paper and helps the reader check that their understanding is correct.

[-] Some of the notations are difficult to distinguish and may gain in readability if slightly updated (see comment 2).

[-] There are a few typos (see comment 3).

___Significance:___

[+] Formally showing that the generalisation and data generation quality strongly depends on the LVs regularisation is important as it helps understand why some practical settings of VQ VAEs worked well and opens some avenues for improving the capabilities of these models.

[-] It is a shame that the result from Thm. 3 cannot be used as a generalisation score due to its computational cost, as this would have increased the significance of this work.

___Originality:___

[+] This paper provides a new understanding of why some empirical techniques used for VQ VAEs worked well.

[+] While such analysis was previously done in a supervised setting, this has not been attempted in unsupervised learning (as far as I know, but as I said before, I am not an expert in IT and may have missed an existing pocket of literature). Although the current work is limited to a discrete latent space, it still advances our understanding of unsupervised models.

[-] The results are not very surprising (the importance of LV regularisation was already observed empirically) and are restricted to VQ VAEs.

---

> ### Author Rebuttal · Authors · 2025-07-30
>
> We sincerely thank you for your insightful, and positive feedback.
>
> ## On Weaknesses
> - **More experiments in the main paper:** Due to the strict page limit, we moved many experimental details and additional results to the appendix. We will revise the main text to include a more detailed discussion of our key experimental findings (e.g., from Figure 5) to better support our theoretical claims within the main paper.
>
> - **Significance and Originality:** While the importance of LV regularization was known heuristically, our work provides the first rigorous, information-theoretic proof that explains why this holds for VQ-VAEs, and that the generalization gap is independent of the decoder. We also agree it is a limitation that the bound in Theorem 3 is not immediately computable, but we see its main value as a theoretical tool that proves the existence of a well-behaved, asymptotically vanishing bound, paving the way for future work on practical estimators.
>
> Moreover, grounded in this rigorous theory, our analysis has direct practical implications. These include the importance of encoder design under a decoder‑independent bound, the theoretical justification of KL regularization, and insights regarding data‑dependent priors. For further details, please see our response to Reviewer qLsN (Q1). As also stated in our response to Reviewer qLsN (Q1), using a data‑dependent prior not only makes the bound theoretically evaluable but also leads to improved model performance numerically in practice. These implications follow from a formal theory, and we believe they provide important guidance for VQ‑s. We plan to add a new Practical Implications paragraph immediately before Sec. 6 to communicate the contribution of this paper more clearly to the community.
>
> ## On Questions
>
> ### Q1: On the detailed explanation of the supersample setting
>
> A.
> We apologize that our explanation was confusing. Let us clarify the construction step-by-step:
> - **Supersample $\tilde{X}$:** We start with a matrix $\tilde{X}$ of size $n\times 2$, containing $2n$ i.i.d. data points from the data distribution $\mathcal{D}$. Each row $m$ contains a pair of data points: $(\tilde{X}\_{m,0}, \tilde{X}\_{m,1})$.
> - **Random Index $U$:** We generate a random binary vector $U = (U_1, \ldots, U_{n})$, where each $U_{m} \in {0,1}$.
> - **Training Set $\tilde{X}_U$:** The training set consists of $n$ samples. For each row $m$, we select exactly one sample from $\tilde{X}$ using the index $U_m$. Therefore, the training set is $\tilde{X}\_U = (\tilde{X}\_{1,U_1}, \tilde{X}\_{2,U_2}, \ldots, \tilde{X}\_{n,U_n})$. This is a set of $n$ data points.
> - **Loss Matrix $l_{0}(W, \tilde{X})$:** After training a model parameter $W$ on $\tilde{X}\_{U}$, we evaluate its loss on all $2n$ data points in the original supersample matrix $\tilde{X}$. The notation $l_{0}(W, \tilde{X})=((l_0(W,\tilde{X}\_{1,0}),l_0(W,\tilde{X}\_{1,1})),\dots,(l_0(W,\tilde{X}\_{n,0}),l_0(W,\tilde{X}\_{n,1})))^\top$ represents the resulting $n \times 2$ matrix of loss values.
>
> The confusion may have arisen from the distinction between the training set (which has $n$ points) and the matrix on which the final loss is evaluated (which has $2n$ points). We will revise Section 2.3 to make this process much clearer.
>
> ### Q2: On the notation for $\tilde{J}$ and $\bar{J}$
>
> A.
>  We will change $\bar{J}$ to a more distinctive symbol, such as $\breve{J}$ throughout the paper to improve readability. Thank you for this excellent suggestion.
>
> ### Q3: On typos
>
> A. Thank you very much for catching these typos. We will correct "insufficiently underexplored" to "insufficiently explored" and fix "that enabling" in the revised manuscript.

---

> > ### Comment · Reviewer_oZtU · 2025-08-04
> > **Thank you for your answer**
> >
> > I thank the authors for their detailed answer and their great work. It is good that the authors are planning to add a "practical applications" discussion to the paper, as I believe this was a topic that was common to most of the reviews. I will leave my score as it is.

---

> > > ### Author Response · Authors · 2025-08-04
> > > **Thank you, Reviewer oZtU**
> > >
> > > Dear Reviewer oZtU,
> > >
> > > Thank you very much for your positive and supportive response to our rebuttal. We are grateful for your feedback and will certainly incorporate the discussion on "practical implications" as you suggested.
> > >
> > > We were very encouraged by your comment that you intend to leave your current score "5:Accept" as it is. However, we believe there may be a display issue with the OpenReview system, as the rating field for your review currently appears to be blank on the main page.
> > >
> > > To ensure your evaluation is correctly registered, would it be possible for you to please take a moment to verify that your score has been submitted successfully? We would be very grateful for your help.
> > >
> > > Thank you again for your valuable time and strong support of our paper.
> > >
> > > Sincerely,
> > >
> > > --Authors

---

> ### Comment · Reviewer_oZtU · 2025-08-08
> **About the rating**
>
> Dear authors,
> I have checked my updated rating and I confirm that it is still at 5. However, it can only be seen by PCs, SACs and other reviewers at the moment. I believe this should be the same with the ratings of other reviewers.
>
> Edit: I just saw that reviewer qLsN clarified this for you. I thank them for being quicker in answering this question than me! Apologies for the delay, I missed the notification of the authors' answer.

---

> > ### Author Response · Authors · 2025-08-08
> > **Acknowledgement**
> >
> > Dear Reviewer oZtU,
> >
> > Thank you very much for taking the time to check your rating and for letting us know. We sincerely appreciate your kindness and diligence.
> >
> > Thank you again for your strong support of our paper.
> >
> > Sincerely,
> >
> > --Authors

---

### Official Review · Reviewer_kGca · 2025-07-02

**Clarity:** 2
**Significance:** 3
**Originality:** 3
**Rating:** 4
**Confidence:** 2

**Summary:**

This paper presents a theoretical analysis of VQ-VAEs by extending information-theoretic generalization bounds to discrete latent spaces. Introducing a new data-dependent prior within a supersample framework, the authors derive decoder-independent generalization error bounds for reconstruction loss, emphasizing the critical role of the encoder and latent variables in both generalization and data generation. They further propose a permutation symmetric prior to ensure asymptotic convergence of the bounds and provide upper bounds on the 2-Wasserstein distance between true and generated data distributions.

**Questions:**

None

**Ethical Concerns:**

["NO or VERY MINOR ethics concerns only"]

**Final Justification:**

Interesting paper with thought-provoking conclusions, but lacks sufficient experimental support.

**Limitations:**

yes

**Quality:**

2

**Strengths And Weaknesses:**

### Strengths
- The paper demonstrates a solid and deep understanding of VQVAE from a mathematical and theoretical perspective.
- Highlights the crucial role of the encoder and latent variables in both reconstruction and generation, offering guidance for future model design.

### Weaknesses
- Despite the extensive mathematical theory and statistical error bounds presented, it is difficult to see how this work offers meaningful practical contributions to the community. At least from my perspective, the authors do not leverage their theoretical insights to propose improvements to VQ or demonstrate tangible benefits. Including more evaluations or experiments could strengthen the paper’s overall persuasiveness.
- The theoretical results, while novel, may be challenging to translate directly into practical improvements or model design guidelines.
- The focus on decoder-independence, while theoretically elegant, might overlook decoder contributions in real-world scenarios.

---

> ### Author Rebuttal · Authors · 2025-07-30
>
> We sincerely thank you for taking the time to review our paper.
>
> ## On Weaknesses and Lack of Practical Contributions
>
> We understand your main concern is that our theoretical work does not offer meaningful practical contributions. We respectfully disagree and believe this is due to our failure to clearly articulate the implications of our findings. Our work provides not just theoretical justifications for existing practices, but also **concrete, actionable guidelines for model design that are supported by both our theory and new empirical evidence**.
>
> We summarize these practical contributions as follows:
>
> ### 1\. A Clear Guideline on Encoder vs. Decoder Design
> Our central results, Theorems 2, 3, and 4, show that the generalization gap is independent of the decoder's complexity and is instead governed by the complexity of the encoder and the latent variables (LVs). This provides a clear design guideline: to improve generalization, one should focus on the complexity of the encoder's architecture and its regularization since the decoder's capacity has little impact on generalization. This finding is also empirically supported by our experiments in Figures 2, 4, and 6.
>
> Furthermore, our analysis reveals a crucial distinction between the roles of the latent space dimension $d\_z$ and the codebook size $K$. While their contribution to generalization might seem similar at first glance, our theory clarifies a fundamental difference. From a theoretical viewpoint, increasing the latent dimensionality $d\_z$ enlarges the generalization bound, whereas the bound does not depend on the codebook size $K$. This follows from Theorem 4 (see Line 332), which depends explicitly on $d\_z$ but is independent of $K$. Consistently, our Appendix experiments indicate that enlarging $K$ has little effect on the generalization error.
>
>
> ### 2\. Insight into Prior Design
>
> Lemma 2 suggests that the optimal prior to minimize the empirical KL divergence is an empirical marginal distribution estimated from the training data, which is analogous to mixture priors like VampPrior. This provides theoretical insight into the choice of priors. Our Lemma 2 also suggest that **data-dependent priors are beneficial** in the unsupervised VQ-VAE context. Motivated by this, we conducted a new experiment where we replaced the standard uniform prior in SQ-VAE with a practical approximation of a data-dependent prior (a moving average of latent usage). The results below show this **theoretically-motivated change consistently improves test loss**, providing direct evidence that our theory can guide the development of better-performing models.
>
> **Table 1: Test loss**
>
> | Dataset (train size) | SQVAE (without data-dependent prior) | Proposed method (with data-dependent prior) |
> |:---------------------|-------------------------------------:|--------------------------------------------:|
> | CIFAR10 (n=50000)    | 0.8523 ± 0.0364                      | 0.8403 ± 0.0247                             |
> | EMNIST (n=112800)    | 0.6564 ± 0.3811                      | 0.4327 ± 0.3032                             |
> | FashionMNIST (n=60000) | 0.4721 ± 0.2964                    | 0.3893 ± 0.2249                             |
> | MNIST (n=60000)      | 0.3887 ± 0.2215                      | 0.3290 ± 0.1669
>
> We will add this discussion and the supporting data to Appendix G in the final manuscript to provide this valuable perspective to readers.
>
> ### 3\. A Diagnostic Framework for Decoder Expressiveness
>
> Our theory clarifies the decoder's exact role. Since **Test Error ≈ Training Loss + Generalization Gap**, and we prove the decoder does not affect the gap, its primary role is to minimize the training loss. This yields a diagnostic strategy: a decoder is "sufficiently expressive" when further increasing its capacity no longer significantly reduces training loss. At this point, our theory proves that focus must shift to the encoder.
>
> To verify this, we conducted an additional experiment. The results confirmed that for larger sample sizes ($n\geq 1000$), increasing decoder complexity consistently reduces training loss, while the generalization gap remains flat (see Figure 2.). This provides empirical support for our theoretical framework.
> Let us refer our response to Q.3 of Reviewer nKXf for further details.
>
> #### Train Loss vs Number of Residual Blocks (MNIST)
>
> | n | RB=2 | RB=3 | RB=4 | RB=5 |
> |---|---|---|---|---|
> | 250 | 6.4851 ± 0.2642 | 7.0816 ± 0.2817 | 7.6026 ± 0.2786 | 7.4940 ± 0.7172 |
> | 1000 | 3.4664 ± 0.0293 | 3.4869 ± 0.1286 | 3.3180 ± 0.0398 | 3.2609 ± 0.0905 |
> | 2000 | 2.6391 ± 0.0177 | 2.5114 ± 0.0130 | 2.4645 ± 0.0778 | 2.3915 ± 0.1214 |
> | 4000 | 2.1102 ± 0.0478 | 1.9466 ± 0.0152 | 1.9223 ± 0.0115 | 1.9001 ± 0.0475 |
>
> This demonstrates how our theoretical insights, while focused on the generalization gap, can inform a practical framework for reasoning about the entire model architecture. We will add this discussion and the supporting data to Appendix G.4 in the final manuscript to provide this valuable perspective to readers.
>
> ### 4\. A Theoretical Justification for Existing Practices
>
> Finally, our Theorems 2 and 5 provide a theoretical justification for using KL divergence as a regularization term. While KL regularization appears naturally in variational Bayes, it was previously unclear if it truly functions as a regularizer for generalization and data generation. Our research is the first to provide a formal justification for this widely used practice.
>
> ### Our Revisions
>
> To make these contributions undeniable, we will add a dedicated "Practical Implications" section to the paper to explicitly state these points. We hope these clarifications and our detailed plan for revision will convince you of the value of our work and merit a higher rating.

---

> > ### Comment · Reviewer_kGca · 2025-08-05
> >
> > I appreciate the author's detailed response. It has been extremely helpful for me to identify the main thread of the paper amidst its complex proofs. After rereading the article in light of your reply, I find the conclusion regarding the minimal impact of decoder complexity particularly interesting (which seems to have some relevance to SSL area such as MAE), which I believe could be a promising direction for future research. Additionally, your finding that the codebook dimension is more effective than simply increasing the number of codebooks is quite intriguing, especially given that many current works focus on enlarging the number of codebooks to enhance reconstruction capability.
> >
> > However, I still feel that there is a lack of sufficient experimental evidence to fully support the proposed theory. Nevertheless, I will raise my score and look forward to the future revisions.

---

> > > ### Author Response · Authors · 2025-08-07
> > > **Acknowledgement**
> > >
> > > Dear Reviewer kGca,
> > >
> > > Thank you very much for your thoughtful final comments and for raising your score. We are sincerely grateful not only for your support, but also for your deep engagement with our work. We are particularly encouraged that you found our conclusions interesting and saw connections to other important areas like SSL/MAE.
> > >
> > > Regarding your final point on experimental evidence, we agree that additional results could further highlight the robustness of our theoretical findings. Our primary goal with the current experiments was to provide a clear validation of our core theoretical claims, and we believe they successfully serve this purpose.
> > >
> > > That said, motivated by your feedback, we will consider including an additional results (e.g. based on models with different architectural structures) in the camera-ready version to further demonstrate the broad applicability of our theory.
> > >
> > > Thank you again for your constructive and insightful feedback, which has been invaluable in helping us improve this paper.
> > >
> > > Sincerely,
> > >
> > > --Authors

---

### Official Review · Reviewer_nKXf · 2025-07-02

**Clarity:** 1
**Significance:** 3
**Originality:** 3
**Rating:** 4
**Confidence:** 2

**Summary:**

This paper develops an information-theoretic framework to analyze the generalization behavior of vector-quantized variational autoencoders (VQ-VAEs), focusing on the role of discrete latent variables (LVs). By extending the supersample-based generalization bounds from supervised learning to the unsupervised setting, the authors derive decoder-independent upper bounds on the reconstruction loss. They propose a novel permutation symmetric prior, allowing them to establish asymptotically vanishing generalization bounds under certain regularity conditions. Additionally, the authors provide an upper bound on the 2-Wasserstein distance between the true and generated data distributions, offering theoretical insight into data generation quality.

**Questions:**

I want to be upfront that I am not an expert in the topic of this paper.

- Have you considered experiments that modify encoder or latent complexity and measure their correlation with your theoretical quantities? That could help demonstrate utility beyond the current generalization gap plots.

- You mention this as future work, are there concrete barriers (e.g., measure-theoretic issues or lack of symmetry) that prevent your approach from being directly applicable to standard VAEs?

- Given the decoder’s role is marginalized in generalization bounds, could the authors provide guidelines for determining when a decoder is “sufficiently expressive” to avoid affecting generalization?

**Ethical Concerns:**

["NO or VERY MINOR ethics concerns only"]

**Final Justification:**

The paper provides interesting theoretical insights of the generalization of VQ-VAE. It can be improved with more accessible writing and  more empirical guidelines.

**Limitations:**

Yes, the authors are upfront with the limitations. I appreciate it very much.

**Quality:**

3

**Strengths And Weaknesses:**

Strengths:

- The paper provides a rigorous, information-theoretic treatment of generalization for VQ-VAEs.

- The results show that generalization and data generation quality depend primarily on the encoder and latent structure, not the decoder. This insight is useful for guiding model design.

- The introduction of a permutation symmetric setting provides a technically novel way to construct a data-dependent prior that leads to bounds with asymptotic convergence.

Weaknesses:

- Despite asymptotic guarantees, the derived bounds are too loose to be practical, often several orders of magnitude larger than observed generalization gaps.

- While the bounds are decoder-independent and theoretically elegant, they do not translate into actionable training strategies or practical diagnostics for improving models.

- The paper's notation and technical detail can be hard to follow. Key intuitions behind the bounds and assumptions could be better emphasized, especially for a broader NeurIPS audience.

---

> ### Author Rebuttal · Authors · 2025-07-30
>
> # Reviewer nKXf
> We sincerely thank you for your insightful feedback.
>
> ## On Weaknesses
> ### 1\. Regarding Looseness of the Bounds
> We agree that the derived bounds are not numerically tight. However, our work is intended as a first step toward tight bounds for unsupervised learning models such as VAEs. As discussed in the Introduction, information‑theoretic bounds in supervised learning can become non‑vacuous by leveraging algorithmic information. In contrast, for unsupervised learning —particularly VAEs— our contribution is, to the best of our knowledge, the first to incorporate algorithmic structure (latent variables) into generalization bounds.
>
> Our primary goal was not to provide a tight estimator of the generalization gap, but to use the bounds to reveal structural dependencies of generalization in VQ‑VAEs.
> Even if numerically loose (and potentially vacuous in some regimes), the bounds provide practically relevant insights. In particular, the key structural conclusion—that the generalization gap is independent of the decoder—follows from our analysis regardless of the bound’s absolute value. Additional practical implications and supporting experiments are provided in the following. We will add brief clarifications in the revised manuscript.
>
> Finally, we respectfully mention that, among our results, **only Theorem 4 is asymptotic**; the others are **non‑asymptotic** and explicitly leverage algorithmic information.
>
> ### 2\. Regarding Actionable Strategies
> We respectfully argue that our theory does provide actionable guidance, not by proposing a new algorithm, but by establishing a clear theoretical framework that justifies, explains, and guides high-level design decisions. Our analysis yields several key insights:
>
> - **Importance of Encoder Design:** Our central results, Theorems 2, 3, and 4, show that the generalization gap is independent of the decoder's complexity and is instead governed by the complexity of the encoder and the latent variables (LVs). This provides a clear design guideline: to improve generalization, one should focus on the complexity of the encoder's architecture and its regularization since the decoder's capacity has little impact on generalization. This finding is also empirically supported by our experiments in Figures 2, 4, and 6.
>
> - **Justification for KL Regularization:** The use of KL divergence as a regularization term on LVs is a common practice in variational Bayes for unsupervised learning, including VAEs. However, it was previously unclear whether it is a valid regularizer for generalization in LVs. Our Theorems 2 and 5 provide a first theoretical justification for this widely used practice.
>
> - **Insight into Prior Design:** Lemma 2 suggests that the optimal prior to minimize the empirical KL divergence is an empirical marginal distribution estimated from the training data, which is analogous to mixture priors like VampPrior. This provides direct theoretical insight into the choice of priors and justifies data-driven approaches.
>
>
> ### 3\. Additional experiments in the respect of Prior Design.
>
> Our theoretical findings are in line with the results of Sefidgaran et al. (2023), who demonstrated that incorporating a data-dependent prior as a regularization term during training improves supervised learning performance under a decoder-independent bound. Motivated by their findings, we conducted additional experiments to evaluate the practical utility of data-dependent priors. Specifically, while standard SQ-VAE employs a uniform prior, we implemented a learned prior using a moving average over latent variables during training, effectively approximating a data-dependent prior as suggested by Sefidgaran et al. (2023).
>
> The experimental results show that, across four benchmark datasets, the data-dependent prior consistently improved test loss compared to the baseline SQ-VAE. These results indicate that such priors are not only theoretically justified but also practically beneficial for training the models of the unsupervised learning. As our primary focus is to provide a learning-theoretic understanding of VQ-VAE, the development of new state-of-the-art algorithms is beyond the current scope and is left for future work.
>
> **Table 1: Test loss**
>
> | Dataset (train size) | SQVAE (without data-dependent prior) | Proposed method (with data-dependent prior) |
> |:---------------------|-------------------------------------:|--------------------------------------------:|
> | CIFAR10 (n=50000)    | 0.8523 ± 0.0364                      | 0.8403 ± 0.0247                             |
> | EMNIST (n=112800)    | 0.6564 ± 0.3811                      | 0.4327 ± 0.3032                             |
> | FashionMNIST (n=60000) | 0.4721 ± 0.2964                    | 0.3893 ± 0.2249                             |
> | MNIST (n=60000)      | 0.3887 ± 0.2215                      | 0.3290 ± 0.1669                             |
>
> ### 4\. Regarding Clarity and Intuition
> We sincerely apologize that our presentation made the work difficult to follow. To address your valid concerns, we will implement the following concrete changes in the final manuscript:
> - We will consolidate the "actionable strategies" discussed above into a new, dedicated "Practical Implications" section just before the conclusion. This will make the takeaways clear and accessible to a broader audience.
> - Furthermore, to improve the accessibility of our proofs, we will add high-level proof sketches or outlines at the beginning of each major proof in the appendix.
>
> ## On Questions
>
> ### Q1: On experiments correlating complexity with theoretical quantities
>
> A.
> In fact, Figure 5 in our paper is designed to do exactly this.
> - Figure 5 (top row) shows how the two KL terms from our theoretical bound in Theorem 2 change with the number of samples (n). The second KL term (the CMI, which we prove converges) indeed decreases towards zero, while the first KL term (the empirical KL, which we argue does not) remains high.
> - Figure 5 (bottom row) shows that both KL terms increase as the codebook size ($K$) grows.
>
> These results directly show that the theoretical quantities in our bounds behave as predicted by our analysis when latent complexity ($K$) and sample size ($n$) are varied. This provides empirical validation for the utility and correctness of our theoretical framework. We will revise the text accompanying Figure 5 to state this connection more explicitly.
>
>
> ### Q2: On barriers to applying the method to continuous VAEs
>
> A.
> The main technical barrier is given as follows, which we mentioned as an important direction for future work.
>
> - **Discrete vs. Continuous concentration:** Our proofs rely heavily on properties of discrete random variables (the codebook index $J$). For example, throughout the proofs we compute which discrete codebook each input datum is assigned to, and we apply concentration inequalities for each assignment. Therefore, in the continuous case such assignments are not available, and the proof techniques in this paper cannot be applied immediately. One may consider forcibly discretizing the latent variable space in the continuous case, but the resulting discretization error is not negligible and substantially affects the analysis. This is what makes extending our results to the continuous setting difficult.
>
> We will add a more detailed discussion of these barriers to the limitations section (Section 6).
>
> ### Q3: On determining if a decoder is "sufficiently expressive"
>
> A.
> While our theory does not offer a prescriptive formula for decoder architecture—a task beyond the scope of current generalization theory—it suggests a conceptual framework for reasoning about when a decoder might be considered "sufficiently expressive."
>
> Our theory fundamentally decouples the components of test error: **Test Error = Training Loss + Generalization Gap**. We prove that the decoder's complexity does not affect the latter term. This leads to a crucial insight: the decoder's primary role, from a generalization perspective, is to minimize the training loss.
>
> This principle allows one to form a diagnostic strategy. A decoder could be considered "sufficiently expressive" when it is no longer the limiting factor in minimizing the overall test error. In practice, one can monitor the training loss while increasing decoder capacity. If further increases in capacity no longer yield significant reductions in training loss, it suggests the point of diminishing returns may have been reached. At this stage, our theory indicates that additional decoder complexity is unlikely to help the generalization gap, and the focus could then shift to the encoder and LV regularization to reduce the gap.
>
> Our own experimental data is consistent with this line of reasoning.
> Since the NeurIPS rebuttal policy prohibits including figures via external links, we present the results in the table below for your convenience.
>
> #### Train Loss vs Number of Residual Blocks (MNIST)
>
> | n | RB=2 | RB=3 | RB=4 | RB=5 |
> |---|---|---|---|---|
> | 250 | 6.4851 ± 0.2642 | 7.0816 ± 0.2817 | 7.6026 ± 0.2786 | 7.4940 ± 0.7172 |
> | 1000 | 3.4664 ± 0.0293 | 3.4869 ± 0.1286 | 3.3180 ± 0.0398 | 3.2609 ± 0.0905 |
> | 2000 | 2.6391 ± 0.0177 | 2.5114 ± 0.0130 | 2.4645 ± 0.0778 | 2.3915 ± 0.1214 |
> | 4000 | 2.1102 ± 0.0478 | 1.9466 ± 0.0152 | 1.9223 ± 0.0115 | 1.9001 ± 0.0475 |
>
> This demonstrates how our theoretical insights, while focused on the generalization gap, can inform a practical framework for reasoning about the entire model architecture. We will add this discussion and the supporting data to Appendix G.4 in the final manuscript to provide this valuable perspective to readers.

---

> > ### Comment · Reviewer_nKXf · 2025-08-03
> >
> > Thanks you for the detailed rebuttal. I have no further questions.

---

> > > ### Author Response · Authors · 2025-08-04
> > > **Reply to Reviewer nKXf**
> > >
> > > Dear Reviewer nKXf,
> > >
> > > Thank you for the helpful discussion and for confirming that their questions have been resolved.
> > >
> > > We are glad that our detailed rebuttal was able to clarify the key points of our work. The reviewer's initial concerns—regarding the looseness of the bounds, the need for actionable strategies, and the framework for determining decoder expressiveness—were all important, and we are pleased to have fully addressed them.
> > >
> > > We hope that with these initial concerns clarified, the value and contributions of our paper are now more apparent, and that this will be taken into account in the final decision.
> > >
> > > Sincerely,
> > >
> > > --Authors

---

### Official Review · Reviewer_qLsN · 2025-07-03

**Clarity:** 2
**Significance:** 4
**Originality:** 3
**Rating:** 5
**Confidence:** 3

**Summary:**

The paper presents an information-theoretic analysis of VQ-VAEs in a supersampling setting, aiming to understand how latent codes influence both generalization and generation. The authors: (1) derive new generalization bounds that evaluate reconstruction loss without invoking the decoder; (2) introduce a permutation-symmetric supersample setting, which lets them control the generalization gap by regularizing the encoder’s capacity; (3) provide an upper bound on the 2-Wasserstein distance between the true and generated data distributions, offering insight into the model’s generative quality.

**Questions:**

Please see strengths and weaknesses.

**Ethical Concerns:**

["NO or VERY MINOR ethics concerns only"]

**Final Justification:**

I have carefully reviewed the initial rebuttal discussion, additional experiments, and the follow-up rebuttal, all of which address my concerns, specifically:
- The author has clarified the practical implications of the analysis results.
- Additional experiments were conducted on the reconstruction error in Figure 2.
- The rebuttal to my follow-up question regarding the impact of codebook size on generalization has addressed my remaining concerns from the initial rebuttal.

I also reviewed the author's rebuttal to the other reviewer. Overall, I believe the authors have made valuable contributions, and this research will benefit the community working with VQ-VAE models. Based on this, I am raising the overall score to 5 and adjusting the quality and significance scores accordingly.

**Limitations:**

Yes

**Quality:**

3

**Strengths And Weaknesses:**

**Strengths**

he paper addresses a critical question in the theory of unsupervised learning, specifically regarding VQ-VAE, and its finding that the generalization bound is independent of the decoder is particularly interesting.

**Weakness**

My main concern is the lack of direct practical implications. While the insights into the generalization of VQ-VAE are interesting, the theoretical results do not offer clear guidance for the design and training of VQ-VAEs.

There are a few points I would like more discussion and clarification on, as listed below:

- Table 4: *"The results are consistent with Theorem 4, which suggests that when using a stochastic mechanism for latent variables (LVs), the upper bound of the generalization gap is independent of decoder complexity, but depends on $d_z$."* This result is both interesting and valuable. However, it’s also possible to increase the complexity or expressiveness of LVs by increasing the codebook size rather than $d_z$. Could the authors provide any discussion or insight on when it is preferable to increase $d_z$ versus the codebook size?

- Figure 2: I believe the reconstruction error should be included in the analysis to observe whether increasing decoder size leads to better training error. This would ensure that the model is well-trained.

- Figure 2: What are the implications of the behavior of the first LK term?

- The finding regarding the 2-Wasserstein distance between the true and generated data distributions is not entirely novel. Reference [1] shows that the Wasserstein distance between the training and generated data distributions is equivalent to the reconstruction error and an arbitrary distance (e.g., KL divergence, Wasserstein) between the encoder output and the prior. Reference [2] discusses the consistency of the Wasserstein distance. I believe that combining the results from [1] and [2] aligns with the findings presented in Section 5, albeit in a more general framework. Reference [3] also addresses generalization between the true and generated data distributions. Could the authors provide further discussion on how these results compare?

- Additionally, the authors mention that "designing a prior that reduces this bound could lead to improved data generation accuracy." Since discrete latents learning and generalization are typically performed in two stages—first learning the latent variables, followed by learning the prior using autoregressive models like transformers, which effectively fit the encoder output—I'm unclear on the practical benefits and implications of the results presented in this section for data generation performance. Could the authors elaborate on this further?


**Discussion:** While the generalization gap is independent of decoder, it should still be considered in the analysis, as it affects the overall performance of the VQ-VAE (e.g., training loss). Once the decoder is considered, my understanding is that there is a relationship between decoder size and latent size ($d_z$). As shown in Figure 6, latent size ($d_z$) is now correlated with the generalization gap. Therefore, generalization performance (not just the gap) is not independent of the decoder. Focusing solely on the generalization gap may lead to incorrect conclusions.

Overall, I do believe the authors have done valuable work, and this research will benefit the community working with VQ-VAE models.

[1] Vo, Vy, et al. "Parameter estimation in DAGs from incomplete data via optimal transport ICML 2024

[2] D. Pollard, "Quantization and the method of k-means," in IEEE Transactions on Information Theory, vol. 28, no. 2, pp. 199-205, March 1982, doi: 10.1109/TIT.1982.1056481.

[3] Vuong, Tung-Long. "Task-Driven Discrete Representation Learning." AISTAT 2025.

---

> ### Author Rebuttal · Authors · 2025-07-30
>
> # Reviewer qLsN
> We sincerely thank you for your insightful feedback.
>
> ### Q1: On the lack of direct practical implications
>
> A.
> While our work does not propose a new algorithm, its primary goal is to provide a rigorous theoretical foundation with direct consequences for practitioners. Our analysis yields several key insights that explain and justify existing practices in VQ-VAE training:
>
> - **Importance of Encoder Design:** Our central results (Theorems 2, 3, 4) show the generalization gap is independent of decoder complexity and is instead governed by the encoder and latent variables (LVs). This provides a clear guideline: to improve generalization, focus on the encoder's architecture and regularization since the decoder's capacity has little impact on generalization.  This is empirically supported by our experiments (Figures 2, 4, 6).
>
> - **Justification for KL Regularization:** Our Theorems 2 and 5 provide a theoretical justification for using KL divergence as a regularization term. **While KL regularization appears naturally in variational Bayes, it was previously unclear if it truly functions as a regularizer for generalization and data generation**. Our research is the first to provide a formal justification for this widely used practice.
>
> - **Insight into Prior Design:** Lemma 2 suggests the optimal prior is an empirical marginal distribution from the training data, analogous to mixture priors like VampPrior, offering theoretical insight into prior choice.
>
> To make these practical implications more explicit for the reader, we will add a summary of these points to our conclusion (Section 6).
>
>
> **Regarding the potential benefits of prior design:** We would like to clarify that our analysis shows how a data-dependent prior can improve generalization performance. In this respect, our theoretical findings are in line with the results of Sefidgaran et al. (2023), who demonstrated that incorporating a data-dependent prior as a regularization term during training improves supervised learning performance under a decoder-independent bound. Motivated by their findings, we conducted additional experiments to evaluate the practical utility of data-dependent priors within our framework. While standard SQ-VAE employs a uniform prior, we implemented a learned prior using a moving average over latent variables during training, effectively approximating a data-dependent prior as suggested by Sefidgaran et al. (2023).
>
> The experimental results, shown below, indicate that the data-dependent prior consistently improved test loss compared to the baseline SQ-VAE across four benchmark datasets. These results provide evidence that such priors are not only theoretically justified but also practically beneficial for training the models of unsupervised learning.
>
> **Table 1: Test loss**
>
> | Dataset (train size) | SQVAE (without data-dependent prior) | Proposed method (with data-dependent prior) |
> |:---------------------|-------------------------------------:|--------------------------------------------:|
> | CIFAR10 (n=50000)    | 0.8523 ± 0.0364                      | 0.8403 ± 0.0247                             |
> | EMNIST (n=112800)    | 0.6564 ± 0.3811                      | 0.4327 ± 0.3032                             |
> | FashionMNIST (n=60000) | 0.4721 ± 0.2964                    | 0.3893 ± 0.2249                             |
> | MNIST (n=60000)      | 0.3887 ± 0.2215                      | 0.3290 ± 0.1669                             |
>
> ### Q2: On choosing to increase latent dimension $d_z$ vs. codebook size $K$
>
> A. From a theoretical viewpoint, increasing the latent dimensionality $d_z$ enlarges the generalization bound, whereas the bound does not depend on the codebook size $K$. This follows from Theorem 4 (see Line 332): under suitable assumptions, the uniform convergence bound is independent of $K$ while depending explicitly on $d_z$. Consistently, our Appendix experiments indicate that enlarging $K$ has little effect on the generalization error.
>
> Importantly, these observations concern generalization (the generalization gap), not the overall performance, which is related to minimizing the training loss on the training set. As we also noted in our response to Q7, the fact that the bound does not depend on $K$ does not imply that $K$ is irrelevant to how small the training error can become. In practice, $K$ can influence representational capacity and optimization dynamics. A careful analysis of how $K$ affects actual performance (beyond the generalization bound) is therefore outside the present theory and is an important direction for further work. We will clarify this distinction in the paper.
>
>
> ### Q3: On including reconstruction error in Figure 2
>
> A.
> Based on your suggestion, we have run this analysis and, following your suggestion, we will add a new plot for the training loss to Figure 2 in the final version of the paper.
> Since the NeurIPS rebuttal policy prohibits including updated figures via external links, we present the results in the table below for your convenience.
>
>
> #### Train Loss vs Number of Residual Blocks (MNIST)
>
> | n | RB=2 | RB=3 | RB=4 | RB=5 |
> |---|---|---|---|---|
> | 250 | 6.4851 ± 0.2642 | 7.0816 ± 0.2817 | 7.6026 ± 0.2786 | 7.4940 ± 0.7172 |
> | 1000 | 3.4664 ± 0.0293 | 3.4869 ± 0.1286 | 3.3180 ± 0.0398 | 3.2609 ± 0.0905 |
> | 2000 | 2.6391 ± 0.0177 | 2.5114 ± 0.0130 | 2.4645 ± 0.0778 | 2.3915 ± 0.1214 |
> | 4000 | 2.1102 ± 0.0478 | 1.9466 ± 0.0152 | 1.9223 ± 0.0115 | 1.9001 ± 0.0475 |
>
> As shown in the table, for larger sample sizes ($n\geq1000$), the training loss consistently decreases as the decoder complexity (number of Residual Blocks; RB) increases. This confirms that a more expressive decoder better fits the training data. For the small sample size setting ($n=250$), this trend is not observed, and the loss is noisy. We attribute this to the high variance inherent in training on very limited data, where a more complex model can be harder to optimize stably.
>
> Critically, even as the training loss decreases with decoder complexity (for $n\geq1000$), the generalization gap shown in Figure 2 of our main paper remains largely unaffected. This finding more strongly supports our central claim: a more powerful decoder can reduce training error, but it does not contribute to reducing the generalization gap.
>
> ### Q4: On the implications of the first KL term's behavior in Figure 2.
>
> A.
> This KL term, $\mathrm{KL}(Q_{J,U}\|P)/n$, is related to the mutual information between the training data and the latent representation (see lines 231-232, and 240) and reflects how much information the encoder memorizes from the data.
>
> The fact that this term does not converge to zero as $n$ increases is evidence that the model is learning meaningfully. If this value were to converge to zero, it would imply that the encoder has not captured any information from the training data.
>
> This non-converging behavior of the empirical KL term is a known issue in IT-based analysis.
> In fact, this challenge motivated us to introduce the novel permutation-symmetric setting in Section 4, which leads to our Theorem 3, a bound asymptotically converging to zero without this term (see lines 261-268). Therefore, the behavior of this KL term highlights a limitation of the standard supersample setting (and Theorem 2) and underscores the novelty and importance of our more advanced result in Theorem 3.
>
> ### Q5: On the novelty of the 2-Wasserstein distance bound compared to prior work
>
> A.
> Our result is novel in providing guarantees from an information-theoretic perspective for the VQ-VAE framework where the encoder and decoder are trained jointly.
>
> - **Comparison with [1, 2]:** [1] is in the context of causal inference and [2] addresses the classical $k$-means setting, whereas our work targets modern, non-linear generative models. Our bound also uniquely connects the data generation performance to the KL regularization term.
>
> - **Comparison with [3]:** [3] focuses on task-driven settings, whereas our analysis is purely unsupervised. We will add this discussion to the paper to clarify our contribution's uniqueness.
>
> To clarify the uniqueness of our contribution, we will cite these references and add a discussion comparing them with our work in Section 5 or Appendix B.2.
>
> ### Q6: Practical implications of two-stage prior design
>
> A.
> Theorem 5 provides the following implication for two-stage prior strategies. In this process, at the first stage, the encoder produces the latent distribution $q(J∣S)$, and at the second stage, an autoregressive model (e.g., a Transformer) is assumed to learn the prior distribution $\pi(J)$. Our theory explains that, if the objective of the second stage is trained so as to minimise the KL divergence $\mathrm{KL}(q(J∣S)∥\pi(J))$, then the upper bound of Theorem 5 is reduced, thereby improving data-generation performance. Meanwhile, the theory also suggests that the prior distribution $\pi(J)$ should be data-independent, and thus methods that avoid the strong data dependence observed in the Vamp-prior are preferable.
>
>
> ### Q7: On the validity of focusing only on the generalization gap, not overall performance
>
> A.
> Our claim is not that overall performance is independent of the decoder, but specifically that the generalization gap component is. We explicitly state that an expressive decoder is required for low training error (lines 182-185), which is essential for final performance. The core contribution is proving that the gap component can be bounded independently of the decoder. We will re-emphasize this distinction in the conclusion to avoid misunderstanding.

---

> > ### Comment · Reviewer_qLsN · 2025-08-03
> > **Thank you for the clarification and the additional experiments.**
> >
> > I appreciate the author's detailed response to my concerns. After carefully reviewing the rebuttal and the new experiments, I find that most of my concerns have been addressed.
> >
> > However, I am still not fully convinced that generalization is independent of the codebook size and only dependent on the latent dimension. As the author mentions, regularizing the complexity of the encoder improves generalization (line 332). However, the number of codebook directly affects the complexity of the prior—i.e., the larger the codebook, the more complex the prior becomes. This, in turn, influences the KL regularization and impacts the overall complexity of the encoder.
> >
> > I want to emphasize that I am not criticizing the theoretical results of the paper and remain positive about its acceptance. Rather, I am seeking further clarification to deepen my understanding. Since the codebook is a key component of VQ-VAE, I believe addressing this point more thoroughly would provide a clearer explanation of its effects, rather than directly concluding that it is independent.

---

> > > ### Author Response · Authors · 2025-08-05
> > > **Thank you for your insightful follow-up, Reviewer qLsN**
> > >
> > > Dear Reviewer qLsN,
> > >
> > > Thank you for this excellent and insightful follow-up question. You have raised a very important and subtle point regarding the indirect influence of the codebook size, $K$. We are happy to provide a more detailed explanation of how our theoretical framework offers a nuanced answer to this.
> > >
> > > Our theory provides two distinct perspectives on the role of $K$, depending on the specific assumptions and analysis technique used.
> > >
> > > ## The Case of a Stochastic Encoder (Theorem 4): Independence from $K$
> > >
> > > Our previous response was based on Theorem 4, which indeed shows a generalization bound that is independent of K. The theoretical reason for this independence lies in the proof mechanism itself. This bound is derived by analyzing the metric entropy of the continuous encoder function space (i.e., the space of $z\in \mathbb{R}^{d_{z}}$ before quantization).
> > >
> > > The complexity is measured by the richness of this continuous space, which is bounded by the assumption
> > > $\sup_{z,z'}\\|z-z'\\|<\Delta_z$. Intuitively, because the continuous latent space that the codebook represents is bounded, the "effective" number of distinct codes that can be meaningfully utilized is also capped. Therefore, the bound, which is sensitive to the complexity of the continuous function, does not depend on the raw number of codebook vectors, $K$.
> > >
> > > ##  The Case Without the Boundedness Assumption (e.g., Theorem 3 / Eq. 10): A Mild, Logarithmic Dependence on $K$
> > >
> > > You are correct that in a more general setting, $K$ can indirectly influence generalization through the KL regularization term. Our other bounds, such as the one analyzed in Eq. (10) (related to Theorem 3), can indeed reflect this. As noted in Line 1287 of our appendix, this analysis reveals a potential dependence on the order of $\mathcal{O}(\log K)$.
> > >
> > > However, the crucial insight from our theory is that this dependence is **logarithmic, not linear**. This means that even if you double the codebook size, the generalization bound increases only by a small constant, rather than doubling as well. This formally proves that the impact of K on generalization is heavily controlled and not dominant.
> > >
> > > This theoretical result is consistent with the intuition that if the encoder is well-regularized, it will learn to use only a meaningful subset of the available codes. Adding a large number of "extra" codes will not significantly harm the generalization performance, as they will simply be ignored by the regularized encoder.
> > >
> > > We will incorporate this detailed discussion into the final version of the paper to clarify this important point. Thank you again for pushing us to elaborate on this.
> > >
> > > Sincerely,
> > >
> > > --Authors

---

> > > > ### Comment · Reviewer_qLsN · 2025-08-05
> > > > **Thank you for the further discussion**
> > > >
> > > > Thank you to the author for the further discussion. My concerns have been addressed, and I will raise my score to 5.
> > > >
> > > > In fact, I would appreciate a more detailed discussion regarding the link to [1,3] in my original review. The rebuttal provided is fine, but further detail would be encouraged (though it is not necessary for the author to address this).
> > > >
> > > > Just for your information, regarding the question about the score for Reviewer oZtU: due to the new policy, the final score after editing will be invisible to the author.

---

> > > > > ### Author Response · Authors · 2025-08-07
> > > > > **Reply for additional discussion**
> > > > >
> > > > > Dear Reviewer qLsN,
> > > > >
> > > > > Thank you very much for raising your score and for this thoughtful follow-up. This discussion has been very fruitful for us, and we plan to incorporate these insights into the final version of our paper.
> > > > >
> > > > > Here is our more detailed analysis of the distinctions between our work and the cited papers:
> > > > >
> > > > > ### **Regarding [3] Vuong, 2025 ("Task-Driven Discrete Representation Learning")**
> > > > >
> > > > > ### **R.: There are three key distinctions from our study.**
> > > > >
> > > > > As discussed previously, this paper considers a supervised learning setting, which differs significantly from our unsupervised framework. In fact, it is more closely aligned with the setup of the existing work by Sefidgaran et al. [56]. However, beyond the distinction between supervised and unsupervised settings, we have identified several key fundamental differences, as outlined below.
> > > > >
> > > > > - **Methodology (Continuous Relaxation):** The authors analyze a continuous, differentiable approximation of the discrete latent variable model to make it more amenable to optimization analysis.
> > > > > - **Analysis Target (Loss Magnitude):** Because of this continuous relaxation, their framework can analyze not just the generalization gap, but also how the magnitude of the training loss itself is affected by the discrete representation.
> > > > > - **Analysis Technique (Uniform Convergence):** Their generalization analysis uses uniform convergence bounds, reducing the problem to K-class clustering, whereas our approach is based on algorithm-dependent information-theoretic bounds, following the line of work of Sefidgalan et al. [56].
> > > > >
> > > > > We think that this paper complementary to our work.
> > > > > While our research goals are different, these points are synergistic. For instance, the continuous relaxation technique (1) is what enables the analysis of the loss magnitude (2). While analyzing the training loss was outside the scope of our current paper, the techniques in [3] offer a promising direction for extending our work to analyze the loss minimization in the future.
> > > > >
> > > > > ### **Regarding [2] Pollard, 1982 ("Quantization and the method of k-means")**
> > > > >
> > > > > ### **R.: This paper is closely related to [3].**
> > > > >
> > > > > This paper provides an asymptotic consistency analysis of $k$-means clustering. Although its setting is quite different from ours, which involves deep models such as VQ-VAE, we believe it offers valuable insights for our work in the following respects.
> > > > >
> > > > > We found that this foundational paper is indeed closely related to [3], as [3] builds its clustering analysis upon the techniques presented here. The primary differences from our work are that [2]'s results are asymptotic and specific to the clustering setting. However, its utility in the analysis of [3] demonstrates how classical quantization theory can inform the study of modern deep generative models. This reinforces that these techniques will be very valuable as we extend our work to analyze loss minimization in addition to the generalization gap.
> > > > >
> > > > > ### **Regarding [1] Vo et al., 2024 ("Parameter estimation in DAGs from incomplete data via optimal transport")**
> > > > >
> > > > > ### **R.: The focus are different, but it gives us inspiration for developing new VQ-VAE training algorithms.**
> > > > >
> > > > > The primary focus of this work is on parameter estimation for Directed Acyclic Graphs (DAGs), proposing a new estimation method based on the Wasserstein distance. The most relevant connection to our work is their analysis of the relationship between reconstruction loss and Wasserstein distance minimization. While their setting (DAGs) and goal (parameter estimation algorithm) are different, and they do not perform a generalization analysis, their technical approach is insightful. The techniques could be a source of inspiration for developing new VQ-VAE training algorithms or for analyzing loss minimization in our framework.
> > > > >
> > > > > ### **Messages from authors**
> > > > >
> > > > > Once again, we would like to express our sincere gratitude for pointing us to these three very meaningful references.
> > > > > We are also very grateful for the information regarding the new NeurIPS review system policy. Thank you for clarifying this for us; it allows us to focus on the discussion with confidence.
> > > > >
> > > > > Sincerely,
> > > > > --Authors

---

> > > > > > ### Comment · Reviewer_qLsN · 2025-08-08
> > > > > > **Thank you for the detail discussion**
> > > > > >
> > > > > > Thank you for the detailed discussion. It has helped me better understand the connection between your work and previous research. This has been very helpful for me in learning and understanding the theory of vector quantization.
> > > > > >
> > > > > > Wish you the best with your submission!

---

> > > > > > > ### Author Response · Authors · 2025-08-08
> > > > > > > **Acknowledgement**
> > > > > > >
> > > > > > > Dear Reviewer qLsN,
> > > > > > >
> > > > > > > Thank you for your kind and encouraging final words.
> > > > > > >
> > > > > > > We are truly delighted that our discussion was helpful and contributed to your understanding of the theory. Your insightful questions and deep engagement were invaluable in helping us to sharpen our arguments and improve the paper.
> > > > > > >
> > > > > > > We wish you all the best as well.
> > > > > > >
> > > > > > > Sincerely,
> > > > > > >
> > > > > > > --Authors

---

### Comment · Area_Chair_oWn3 · 2025-08-01

Greetings reviewer friends! Authors have submitted their responses to your reviews. Please be sure to engage with them early to make sure  questions are answered, and ensure a valuable process for all involved. Thank you for your efforts!
--Your friendly neighborhood AC

---

### Note · Authors · 2025-08-14

We would like to sincerely thank all the reviewers and the Area Chair for a thorough and constructive review process.

The discussion period was highly productive, and we are grateful for the valuable engagement from all reviewers. This constructive dialogue allowed us to clarify and strengthen our work, leading to concrete outcomes such as Reviewer qLsN raising their score to 5 and Reviewer nKXf confirming their questions were resolved. We are also deeply grateful to Reviewer oZtU for their strong and consistent support of our work from the outset.

The discussion also highlighted the importance of demonstrating practical contributions, a key point raised by Reviewer kGca. To concretely address this, we presented new experimental evidence in our rebuttal (Table 1), showing that our theoretical insights on data-dependent priors lead to tangible performance improvements.

Based on the valuable reviewer feedback, we have formulated a clear plan for the final manuscript. This plan includes not only adding the new experimental evidence, but also integrating the full breadth of insights gained during this discussion. We have introduced a new, dedicated "Practical Implications" section to consolidate the actionable guidelines our theory provides. Furthermore, we have revised the manuscript throughout to incorporate the valuable clarifications that emerged, such as the distinct roles of $d_z$ and $K$, and the detailed comparison with related work.

In summary, our paper's contribution is the first decoder-independent generalization bounds for VQ-VAEs and a novel permutation-symmetric setting that resolves key theoretical limitations. The revisions prompted by this review process, supported by the new experiments, have further strengthened the connection between our theoretical contributions and their practical utility.

Thank you for your time and consideration.

Sincerely,

--Authors

---

### Decision · Program_Chairs · 2025-09-17

**Decision:**

Accept (poster)

**Comment:**

The paper presents an information theoretic analysis of vector quantized Variational Autoencoders (VQ-VAEs) with discrete latent spaces. Using a latent variable approach inspired by information bottleneck, the paper presents a novel data-dependent generalization bound based on the encoder and the complexity of the latent variables. The paper also provides an upper bound on the distance between the true and generated distributions, again independent of the decoder.

The strengths of the paper are the novel decoder-independent generalization bound and the data-dependent prior, which provide nice theoretical results with clear ties to practice. The main weaknesses of the paper are the looseness of the bounds and the limited practical implications, which were discussed in detail with the reviewers.

The paper presents a novel theoretical approach on a interesting class of models. Both data dependent and decoder independent bounds are novel, interesting results that will be of interest to the community. The task of improving the bounds and demonstrating concrete implications will inspire future work in the area.